# Exposing Weaknesses of Large Reasoning Models through Graph Algorithm Problems

**Qifan Zhang**[*]   **Jianhao Ruan**[*]   **Aochuan Chen**   **Kang Zeng**   **Nuo Chen**[†]
**Jing Tang**   **Jia Li**[†]

The Hong Kong University of Science and Technology (Guangzhou)

`{qzhang297, jruan189, achen149, kzeng228}@connect.hkust-gz.edu.cn`
`chennuo26@gmail.com, {jingtang, jialee}@ust.hk`

## Abstract

Large Reasoning Models (LRMs) have advanced rapidly, yet existing benchmarks on mathematics, code, and common-sense reasoning remain limited: they lack long-context evaluation, offer insufficient challenge, and provide answers that are difficult to verify programmatically. We introduce GRALGOBENCH, a benchmark designed to evaluate LRMs through graph algorithm problems. Such problems are particularly well-suited for probing reasoning abilities: they demand long-context reasoning, allow fine-grained control of difficulty levels, and enable standardized programmatic evaluation. Across nine tasks, our systematic experiments reveal two major weaknesses of current LRMs. **First**, accuracy deteriorates sharply with longer context inputs—falling below 50% once graphs exceed 120 nodes—driven by frequent execution errors, weak memory, and redundant reasoning. **Second**, LRMs suffer from an *over-thinking* phenomenon, primarily driven by extensive yet largely ineffective self-verification, which inflates reasoning traces without improving correctness. By exposing these limitations, GRALGOBENCH establishes graph algorithm problems as a rigorous, multidimensional, and practically relevant testbed for advancing the study of reasoning in LRMs. Code is available at `https://github.com/Bklight999/GrAlgoBench`.

## 1 Introduction

Large language models (LLMs) have evolved rapidly over the past few years, becoming powerful general-purpose AI tools with remarkable achievements in natural language processing and complex reasoning. Recently, the emergence of Large Reasoning Models (LRMs), such as OpenAI-O1 (OpenAI, 2025) and DeepSeek-R1 (Guo et al., 2025), has further pushed the frontier of LLM development. By leveraging long chains of thought enriched with human-like cognitive strategies—including self-verification, strategy-shift, and backtracking—these models show significant improvements on challenging reasoning tasks (Qin et al., 2024; Min et al., 2024; Feng et al., 2025; Dong et al., 2025). Accordingly, evaluating the reasoning capabilities of LRMs and probing their limitations, such as *over-thinking* and *under-thinking*, has become a timely research focus (Zheng et al., 2025; Chen et al., 2024d; Sun et al., 2025; Yang et al., 2025b;c).

Current benchmarks for LRMs, while valuable, exhibit several critical deficiencies when centered on domains like *mathematical reasoning* (Chen et al., 2024b; Petrov et al., 2025; He et al., 2024; Huang et al., 2024; Gulati et al., 2024; Glazer et al., 2024; Li et al., 2025b; Huang et al., 2025), *code generation* (Li et al., 2022; Dai et al., 2024; Yang et al., 2025b; Yu et al., 2024; Wei et al., 2025; He et al., 2025b; Zheng et al., 2025), and *common-sense reasoning* (White et al., 2024; Suzgun et al., 2022; Lin et al., 2025). First, they **lack long-context input evaluation**: existing benchmarks predominantly use short problem texts, which cannot be easily scaled to assess LRMs' reasoning capabilities over extended contexts. As modern LRMs increasingly support longer inputs, rigorous evaluation of their performance on lengthy, complex reasoning tasks becomes essential. Second,

---

[*] Equal Contribution
[†] Corresponding author

Table 1: Comparison of our work with prior benchmarks on graph algorithm problems.

| Work | Graph Size (by nodes) | Reasoning Taxonomy | LRMs' Evaluation | Real-world Data |
|---|---|---|---|---|
| NLGraph (Wang et al., 2023) | 5-35 | ✗ | ✗ | ✗ |
| GPT4Graph (Guo et al., 2023) | 10-20 | ✗ | ✗ | ✗ |
| GraphQA (Fatemi et al., 2023) | 5-20 | ✗ | ✗ | ✗ |
| LLM4DyG (Zhang et al., 2024b) | 5-20 | ✗ | ✗ | ✗ |
| GraphInstruct (Luo et al., 2024) | 5-35 | Node, edge, graph level | ✗ | ✗ |
| GraphArena (Tang et al., 2024) | 4-50 | P and NP | ✗ | ✓ |
| VisionGraph (Li et al., 2024) | 5-35 | ✗ | ✗ | ✗ |
| GraCore (Yuan et al., 2024) | 8-30 | Graph reasoning and graph understanding | ✗ | ✓ |
| GraphOmni (Xu et al., 2025) | 5-30 | Node, edge, graph level | ✗ | ✗ |
| **Ours** | 8-160 | Enumeration, Exploration, Intuition | ✓ | ✓ |

they exhibit **inadequate challenge levels**: Current benchmarks are no longer sufficiently difficult for LRMs (e.g., GPT-5 achieving $94.3\%$ on AIME-2025), yet creating more challenging variants requires non-trivial human effort in problem redesign. Third, their answers are **not standardized and therefore hard to verify programmatically**: In mathematical problems, a single solution can be expressed as $\frac{1}{3}$, $0.33\overline{3}$, or $3^{-1}$, and in code generation, a model may pass functional tests while harboring latent logic flaws that are difficult to catch without exhaustive test suites.

These collective limitations motivate a pivotal research question: *Can we gather a family of reasoning tasks that address the concerns above to form a more rigorous benchmarking suite for LRMs?*

In this work, we posit that *graph algorithm problems* (Wang et al., 2025b; Peng et al., 2025; Zhang et al., 2024a; Chen et al., 2024a; Luo et al., 2024; Zhang et al., 2025) represent a superior and highly promising choice. They furnish several distinctive advantages: (1) **Effective Long-Context Input Reasoning**: Describing a graph typically requires listing nodes and edges, producing lengthy inputs whose solutions cannot be directly extracted but must be derived through multi-step reasoning. These properties make graph algorithm problems naturally suited for evaluating long-context reasoning. For example, OpenAI employed the *GraphWalks* (OpenAI, 2025) dataset to benchmark the long-context capabilities of GPT-5. (2) **Scalable Difficulty Level Control**: Task complexity can be precisely modulated by scaling graph parameters like node size, often leading to quadratic or even exponential growth in difficulty. As evidenced in Table 3, model accuracy deteriorates sharply as graph size surpasses 120 nodes. (3) **Standardized and Programmatic Evaluation**: Outputs typically consist of integers or explicit graph elements (e.g., nodes, edges, or paths) that has no alternative representations, which permits standardized and programmatic verification.

Furthermore, graph algorithm problems are both highly extensible and resistant to data contamination, as minor structural perturbations can generate a vast number of new, valid instances at scale. Crucially, graph algorithms are foundational to numerous real-world applications, including social networks (Ceccarello et al., 2024; Oettershagen et al., 2024), transportation systems (Ahmadian et al., 2024), and web mining (Chen et al., 2024c; Miyauchi et al., 2024; Pang et al., 2024). This practical relevance ensures that such benchmarks are not only rigorous but also grounded by real-world utility.

While several benchmarks have been proposed for evaluating LLMs on graph algorithm problems (Wang et al., 2023; Zhang et al., 2024b; Tang et al., 2024; Yuan et al., 2024), they suffer from three critical limitations (see Table 1). First, prior works focus mainly on non-reasoning LLMs, leaving the behaviors of O1-like LRMs largely unexplored. Second, their graph sizes are capped at 50 nodes, leaving the advantage of scalable difficulty level control unexploited. Third, their task categorizations are often based on ad-hoc criteria such as local vs. global, graph understanding vs. graph reasoning, or P vs. NP, rather than grounded in algorithmic design principles that better capture distinct and practically relevant reasoning paradigms.

To address these gaps, we propose **GRALGOBENCH**, a new benchmark for evaluating LRMs on graph algorithm problems. Figure 1 presents an overview of our benchmark, which consists of:

❶ **Systematic Dataset Construction:** We categorize problems into three categories—*Enumeration*, *Exploration*, and *Intuition*—each containing subproblems with different difficulty levels to probe distinct reasoning capacities. Our dataset comprises 2,700 graphs sampled directly from diverse real-world networks, ranging from 8 to 160 nodes across six scales, which increases diversity and mitigates data contamination risks.

❷ **Comprehensive Evaluation:** We evaluate not only non-reasoning models but also reasoning models (e.g., O1-like LRMs). Multiple metrics are adopted, including Pass@k, Cons@k, Z-score, and outcome efficiency, enabling multi-faceted analysis.

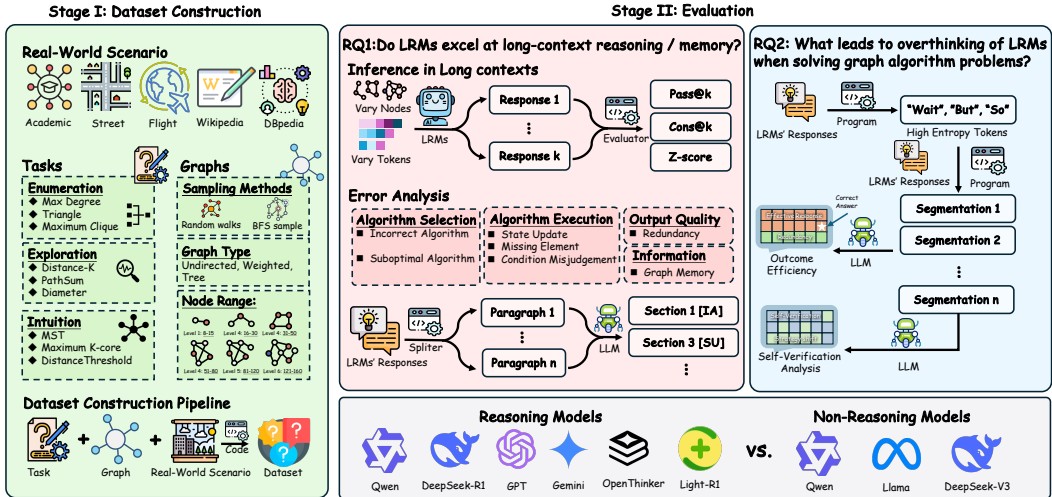

Figure 1: Benchmark overview.

❸ **Novel Research Questions:** We systematically investigate: (1) Do LRMs excel at long-context reasoning (Section 3.1)? (2) What leads to over-thinking of LRMs when solving graph algorithm problems (Section 3.2)?

Our experiments reveal two key findings. **First**, by evaluating LRMs on graphs with varying node sizes as well as on fixed graphs with increasingly verbose textual descriptions, we observe a consistent performance decline as context length grows—for instance, most models achieve less than 50% accuracy once the graph size surpasses 120 nodes. This demonstrates that LRMs are particularly weak at long-context reasoning. Through detailed error analysis, we identify three primary causes: frequent step-by-step execution errors, poor memory capability, and redundant reasoning patterns. **Second**, by analyzing how efficiently models reach the correct answer together with the frequency and effectiveness of self-verification, we find that LRMs engage in extensive yet largely ineffective self-verification, which emerges as the main driver of *over-thinking*.

## 2 BENCHMARK CONSTRUCTION

Figure 1 illustrates the composition and structure of our dataset. Section 2.1 introduces the reasoning taxonomy, followed by task design in Section 2.2 and graph collection in Section 2.3.

### 2.1 GRAPH REASONING TAXONOMY

Graph algorithm problems encompass diverse reasoning paradigms, providing a comprehensive framework to evaluate LRMs' reasoning capabilities across multiple dimensions. Unlike prior approaches that categorize problems based on difficulty levels or by local versus global scopes (Luo et al., 2024; Tang et al., 2024; Yuan et al., 2024; Xu et al., 2025), we propose a taxonomy grounded in the algorithmic nature of reasoning, closely aligned with the major algorithmic design families in *Introduction to Algorithms (CLRS)* (Cormen et al., 2022). Specifically, graph algorithm problems can be systematically divided into three fundamental classes: `Enumeration`, `Exploration`, and `Intuition`. Each class corresponds directly to a canonical algorithmic design paradigm in CLRS:

- **Enumeration (brute-force algorithms)**: Problems in this category align with the brute-force paradigm in CLRS. They require LRMs to systematically generate all possible solutions or elements within a set. The canonical example in graph algorithms is brute-force enumeration of subgraphs or paths. Beyond graph problems, enumeration manifests in domains like mathematics, where LRMs need to enumerate all permutations in a combinatorial problem or all cases in a probability calculation.

- **Exploration (search algorithms)**: This category corresponds to the search paradigm in CLRS, requiring traversal of a state space with potential backtracking. LRMs must explore multiple paths and recover from dead-ends. Typical graph examples include depth-first search and breadth-first search. Beyond graphs, it appears in tasks such as Sudoku or N-Queens, where candidate moves are tried, conflicts detected, and backtracking applied.

- **Intuition (greedy algorithms)**: This category corresponds to the greedy paradigm in CLRS, where locally optimal choices are made to efficiently approximate or eventually reach a global optimum. It tests LRMs' ability to exploit subtle problem-specific signals for efficient decisions. Representative graph examples include Dijkstra's and Kruskal's algorithms; outside graphs, a classic case is Huffman coding.

Importantly, all three paradigms collectively test complementary reasoning abilities of LRMs: enumeration probes *systematic coverage of a search space*, exploration evaluates *multi-path search and backtracking*, and intuition assesses *intuitive and efficient decision making*. Beyond these, all graph problems inherently test **memory** (remembering the graph structure), **logical reasoning** (deriving consequences from premises, e.g., inferring the existence of cycles), and **structural reasoning** (reasoning about relationships among nodes and edges). Owing to this diversity of reasoning paradigms and general cognitive requirements, graph algorithm problems constitute a uniquely challenging and informative benchmark for evaluating the reasoning capabilities of LRMs.

## 2.2 TASK DESIGN

With the problem taxonomy established, the next challenge lies in **classifying graph algorithm problems into these categories**. It is important to emphasize that classification cannot be based solely on a single optimal algorithm for solving the problem, as a given problem can be approached using multiple algorithmic strategies. Moreover, we do not know which algorithmic approach an LRM is more likely to employ when tackling a specific problem. For example, shortest path problems can be solved using brute-force methods (an `Enumeration` approach), DFS (an `Exploration` approach), or Dijkstra's algorithm (an `Intuition` approach).

To address this, we collect a variety of graph algorithm problems and propose an **LLM-as-judge** approach for determining the classification of each problem. Specifically, for each problem—comprising a graph description and a task description expressed in natural language—we first generate 300 Erdős-Rényi (ER) graphs (Erdős et al., 1960) to replace the original graph description, creating a diverse set of problem instances. We then prompt LRMs with these instances, obtain their responses, and use Qwen-2.5-72B (Team, 2024; Yang et al., 2024) as a judge to assign a reasoning category (detailed prompts are provided in Appendix K.1). To validate these labels, three postgraduate students with competitive programming experience manually review a subset of instances and reach almost identical judgments to Qwen-2.5-72B. Finally, we select **9** graph tasks in which the algorithms used by LRMs for reasoning are relatively unambiguous, thereby facilitating more straightforward classification. We present in Appendix H.1 the distribution of algorithms used by representative LRMs across different tasks, illustrating how each task aligns with our proposed reasoning taxonomy.

We now introduce the tasks within each reasoning taxonomy, along with the definition of each task. Based on the optimal time complexity of each task under the corresponding algorithm of the category, we classify the tasks into three levels: easy, medium, and hard. More specified task definitions and the time complexity of the optimal algorithm for each task are given in Appendix F.1.

**Enumeration:** (1) `Maximum Degree node (easy)`: Identify the node with the maximum degree in a given undirected graph and output its degree. (2) `Maximum Weight Triangle (medium)`: Given an undirected graph where each node is assigned a weight, find a triangle (a set of three nodes connected to each other) that has the maximum sum of node weights and output its weight sum. (3) `Maximum Clique Problem (MCP, hard)`: Given an undirected graph, identify a subgraph in which all nodes are pairwise connected, and the subgraph contains the maximum number of nodes. Then, output the size of the subgraph.

**Exploration:** (1) `PathSum (easy)`: Given an undirected tree where each edge is assigned a weight, output the number of paths from the root to the leaves for which the sum of the edge weights exceeds a given threshold $\tau$. (2) `Distance-$k$ (medium)`: Given an undirected graph and a specific node $v$, find all other nodes that are within $k$-hops from $v$ and output the number of qualified

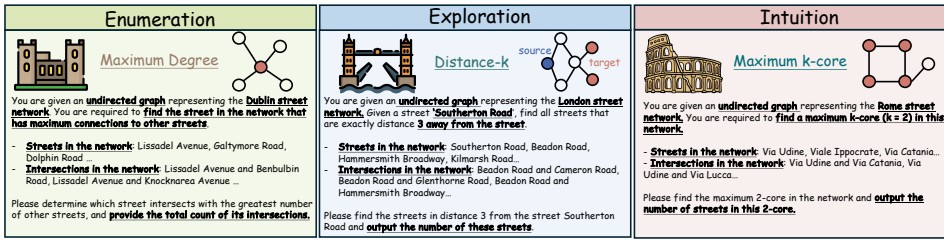

Figure 2: Illustrative problem description.

nodes. (3) `Diameter (hard)`: Identify the longest shortest path between any two nodes in a given undirected graph and output the length of this path.

**Intuition:** (1) `Maximum k-core (MKC, easy)`: Find a subgraph in an undirected graph where each node has a degree of at least $k$ and the subgraph contains the maximum number of nodes. Then, output the size of the subgraph. (2) `Minimum Spanning Tree (MST, medium)`: Given a connected, undirected graph with edge weights, find a spanning tree that connects all nodes with the minimum possible total edge weight and output the total edge weight. (3) `Distance Threshold (hard)`: Given an undirected graph with positive edge weights, find the node that can reach the smallest number of other nodes via shortest paths of length at most a threshold, and output that node.

## 2.3 GRAPH COLLECTION

To measure the semantic understanding capabilities of LRMs and further avoid data contamination, we construct graphs and graph problems using nodes and edges with real-world semantics. We collect graph problems from the following five scenarios: (1) **DBLP (Ley, 2002)**: An academic collaboration network consisting of over 1.3 million nodes (authors) and 5 million edges (collaborations). This graph is used for the PathSum task. (2) **Street Network (Boeing, 2025)**: A collection of city transportation networks worldwide, where nodes represent streets and edges represent intersections between two streets. We use the transportation networks of `London` (containing 36,281 nodes and 83,662 edges, for Distance-$k$), `Sydney` (containing 30,291 nodes and 69,126 edges, for Minimum Spanning Tree), `Dublin` (containing 3,962 nodes and 7,000 edges, for Maximum Degree node), and `Rome` (containing 14,719 nodes and 28,087 edges, for Maximum $k$-core). (3) **OpenFlight (Open-Flights)**: An airport network consisting of 3,390 nodes (airports) and 19,166 edges (flight routes)

Table 2: Average graph statistics across groups. $V$: Avg. number of nodes; $E$: Avg. number of edges; $D$: Avg. density; $\lambda$: Avg. edge connectivity.

| Group | $V$ | $E$ | $D$ | $\lambda$ |
| --- | --- | --- | --- | --- |
| Level-1 | 11.55 | 18.52 | 0.31 | 1.23 |
| Level-2 | 22.49 | 44.51 | 0.19 | 1.02 |
| Level-3 | 39.60 | 90.09 | 0.12 | 0.87 |
| Level-4 | 61.69 | 157.60 | 0.08 | 0.83 |
| Level-5 | 97.83 | 268.50 | 0.06 | 0.82 |
| Level-6 | 138.95 | 407.94 | 0.04 | 0.81 |

weighted by geographical distance. This graph is assigned to the Distance Threshold task. (4) **Wikipedia (Yin et al., 2017)**: A web graph of Wikipedia hyperlinks collected in September 2011, containing 1,791,489 nodes (representing articles) and 28,511,807 edges (representing hyperlinks between articles). This graph is used for the Maximum Weight Triangle and Maximum Clique Problem tasks. (5) **DBpedia (Bizer et al., 2009)**: A knowledge graph subset with over 1 million nodes (entities) and 7 million edges (relationships). This graph is assigned to Diameter task.

For each task, we employ random walk or BFS sampling methods to extract subgraphs of varying sizes from the corresponding real-world graphs. These subgraph sizes are divided into six categories: level-1 (8–15 nodes), level-2 (16–30 nodes), level-3 (31–50 nodes), level-4 (51-80 nodes), level-5 (81-120 nodes), and level-6 (121-160 nodes). In total, the dataset comprises 2,700 problem instances, with each size category containing 50 samples. Examples of problem descriptions for selected tasks are presented in Figure 2, while detailed statistics are provided in Table 2.

## 3 EVALUATION

In this section, we evaluate the capabilities of LRMs in solving graph algorithm problems and analyze the limitations in their reasoning. Our evaluation focuses on two research questions: **RQ1:** Do LRMs

Table 3: We evaluate models on six graph scales (Level-1 to Level-6). For each size, performance is measured on three task taxonomies: Enumeration, Exploration, and Intuition. The evaluation metrics are cons@k (c@k) and pass@k (p@k). We also report per-scale averages across tasks. Results for Level-1 and Level-2 are presented in the Appendix H.2. For each model scale, the best results are highlighted in **bold**, while the second-best results are underlined.

| Models | Level-3 (31-50 nodes) | | | | | | | | Level-4 (51-80 nodes) | | | | | | | |
|---|---|---|---|---|---|---|---|---|---|---|---|---|---|---|---|---|
| | Enumeration | | Exploration | | Intuition | | Avg. | | Enumeration | | Exploration | | Intuition | | Avg. | |
| | c@k | p@k | c@k | p@k | c@k | p@k | c@k | p@k | c@k | p@k | c@k | p@k | c@k | p@k | c@k | p@k |
| Skywork-OR1-7B-Preview | 0.13 | 0.58 | 0.10 | 0.35 | 0.00 | 0.03 | 0.08 | 0.32 | 0.06 | 0.44 | 0.06 | 0.30 | 0.01 | 0.06 | 0.04 | 0.27 |
| Light-R1-7B-DS | 0.11 | 0.53 | 0.07 | 0.35 | 0.01 | 0.08 | 0.06 | 0.32 | 0.08 | 0.34 | 0.07 | 0.30 | 0.00 | 0.03 | 0.05 | 0.22 |
| Distill-Qwen-7B | 0.13 | 0.59 | 0.05 | 0.39 | 0.01 | 0.13 | 0.06 | 0.37 | 0.12 | 0.41 | 0.07 | 0.39 | 0.00 | 0.07 | 0.06 | 0.29 |
| Qwen2.5-7B | 0.11 | 0.39 | 0.10 | 0.21 | 0.00 | 0.03 | 0.07 | 0.21 | 0.09 | 0.36 | 0.11 | 0.29 | 0.03 | 0.07 | 0.08 | 0.24 |
| OpenThinker-7B | 0.19 | 0.61 | 0.07 | 0.43 | 0.01 | 0.13 | 0.09 | 0.39 | 0.03 | 0.41 | 0.04 | 0.32 | 0.01 | 0.03 | 0.03 | 0.25 |
| Qwen3-8B-no-thinking | 0.17 | 0.51 | 0.11 | 0.49 | 0.01 | 0.10 | 0.10 | 0.37 | 0.12 | 0.48 | 0.09 | 0.34 | 0.00 | 0.03 | 0.07 | 0.28 |
| Qwen3-8B | 0.79 | 0.99 | 0.69 | 0.93 | 0.40 | 0.69 | **0.63** | **0.87** | 0.44 | 0.83 | 0.28 | 0.63 | 0.09 | 0.26 | **0.27** | **0.58** |
| GPT-OSS-20B | 0.59 | 0.78 | 0.50 | 0.67 | 0.39 | 0.59 | 0.49 | 0.68 | 0.43 | 0.54 | 0.40 | 0.50 | 0.31 | 0.43 | 0.38 | 0.49 |
| Light-R1-32B | 0.71 | 0.97 | 0.64 | 0.94 | 0.35 | 0.59 | 0.57 | 0.83 | 0.43 | 0.87 | 0.38 | 0.76 | 0.06 | 0.31 | 0.29 | 0.65 |
| Skywork-OR1-32B | 0.87 | 0.99 | 0.77 | 0.99 | 0.51 | 0.75 | **0.72** | **0.91** | 0.54 | 0.87 | 0.51 | 0.84 | 0.12 | 0.42 | 0.39 | 0.71 |
| Distill-Qwen-32B | 0.63 | 0.89 | 0.46 | 0.81 | 0.21 | 0.51 | 0.43 | 0.74 | 0.42 | 0.80 | 0.23 | 0.60 | 0.03 | 0.31 | 0.23 | 0.57 |
| Qwen2.5-32B | 0.25 | 0.61 | 0.10 | 0.43 | 0.04 | 0.19 | 0.13 | 0.41 | 0.16 | 0.49 | 0.06 | 0.31 | 0.04 | 0.13 | 0.09 | 0.31 |
| OpenThinker-32B | 0.75 | 0.97 | 0.56 | 0.91 | 0.30 | 0.57 | 0.54 | 0.82 | 0.44 | 0.87 | 0.31 | 0.74 | 0.11 | 0.30 | 0.29 | 0.64 |
| QWQ-32B | 0.69 | 0.88 | 0.69 | 0.93 | 0.53 | 0.73 | 0.69 | 0.88 | 0.68 | 0.96 | 0.53 | 0.84 | 0.30 | 0.58 | 0.50 | 0.79 |
| Qwen3-32B | 0.79 | 0.99 | 0.69 | 0.93 | 0.40 | 0.69 | 0.63 | 0.87 | 0.52 | 0.89 | 0.51 | 0.87 | 0.26 | 0.59 | 0.43 | 0.78 |
| Qwen3-32B-no-thinking | 0.33 | 0.69 | 0.35 | 0.81 | 0.02 | 0.21 | 0.23 | 0.57 | 0.20 | 0.63 | 0.16 | 0.57 | 0.02 | 0.16 | 0.13 | 0.45 |
| Llama-3.3-70B | 0.10 | 0.36 | 0.13 | 0.39 | 0.01 | 0.04 | 0.08 | 0.26 | 0.04 | 0.27 | 0.00 | 0.21 | 0.00 | 0.01 | 0.01 | 0.16 |
| GPT-OSS-120B | 0.76 | 0.82 | 0.78 | 0.88 | 0.68 | 0.92 | 0.74 | 0.87 | 0.56 | 0.74 | 0.56 | 0.76 | 0.52 | 0.75 | 0.55 | 0.75 |
| Qwen3-235B-A22B-Thinking | 0.98 | 1.00 | 0.96 | 1.00 | 0.96 | 0.99 | **0.97** | **1.00** | 0.88 | 0.98 | 0.89 | 0.99 | 0.81 | 0.92 | **0.86** | **0.96** |
| Qwen3-235B-A22B-Instruct | 0.91 | 0.98 | 0.87 | 0.97 | 0.80 | 0.97 | 0.86 | 0.97 | 0.77 | 0.97 | 0.79 | 0.98 | 0.66 | 0.89 | 0.74 | 0.95 |

| Models | Level-5 (81-120 nodes) | | | | | | | | Level-6 (121-160 nodes) | | | | | | | |
|---|---|---|---|---|---|---|---|---|---|---|---|---|---|---|---|---|
| | Enumeration | | Exploration | | Intuition | | Avg. | | Enumeration | | Exploration | | Intuition | | Avg. | |
| | c@k | p@k | c@k | p@k | c@k | p@k | c@k | p@k | c@k | p@k | c@k | p@k | c@k | p@k | c@k | p@k |
| GPT-OSS-20B | 0.51 | 0.68 | 0.18 | 0.33 | 0.16 | 0.27 | 0.28 | 0.43 | 0.28 | 0.48 | 0.10 | 0.14 | 0.04 | 0.09 | 0.14 | 0.24 |
| Light-R1-32B | 0.80 | 0.80 | 0.45 | 0.45 | 0.01 | 0.07 | **0.42** | 0.44 | 0.22 | 0.68 | 0.05 | 0.23 | 0.00 | 0.01 | 0.09 | 0.31 |
| Skywork-OR1-32B | 0.67 | 0.67 | 0.51 | 0.51 | 0.00 | 0.00 | 0.39 | 0.39 | 0.10 | 0.47 | 0.03 | 0.23 | 0.00 | 0.00 | 0.04 | 0.24 |
| Distill-Qwen-32B | 0.45 | 0.83 | 0.09 | 0.37 | 0.00 | 0.03 | 0.18 | 0.41 | 0.26 | 0.77 | 0.07 | 0.31 | 0.00 | 0.03 | 0.11 | 0.37 |
| Qwen2.5-32B | 0.15 | 0.53 | 0.05 | 0.22 | 0.02 | 0.11 | 0.07 | 0.29 | 0.21 | 0.48 | 0.03 | 0.19 | 0.01 | 0.01 | 0.08 | 0.23 |
| OpenThinker-32B | 0.37 | 0.76 | 0.14 | 0.45 | 0.02 | 0.21 | 0.18 | 0.47 | 0.27 | 0.68 | 0.07 | 0.33 | 0.00 | 0.02 | 0.11 | 0.34 |
| QWQ-32B | 0.61 | 0.95 | 0.25 | 0.63 | 0.09 | 0.25 | 0.32 | **0.61** | 0.47 | 0.87 | 0.13 | 0.40 | 0.09 | 0.21 | **0.23** | **0.49** |
| Qwen3-32B | 0.51 | 0.92 | 0.21 | 0.59 | 0.09 | 0.25 | 0.27 | 0.59 | 0.42 | 0.80 | 0.09 | 0.43 | 0.09 | 0.19 | 0.20 | 0.47 |
| Qwen3-32B-no-thinking | 0.15 | 0.61 | 0.07 | 0.28 | 0.01 | 0.07 | 0.08 | 0.32 | 0.13 | 0.49 | 0.07 | 0.23 | 0.01 | 0.07 | 0.07 | 0.27 |
| Llama-3.3-70B | 0.07 | 0.21 | 0.01 | 0.11 | 0.00 | 0.02 | 0.03 | 0.11 | 0.02 | 0.22 | 0.00 | 0.07 | 0.00 | 0.00 | 0.01 | 0.10 |
| GPT-OSS-120B | 0.51 | 0.67 | 0.41 | 0.51 | 0.22 | 0.28 | 0.38 | 0.49 | 0.37 | 0.54 | 0.19 | 0.33 | 0.11 | 0.16 | 0.22 | 0.34 |
| Qwen3-235B-A22B-Thinking | 0.86 | 0.96 | 0.52 | 0.82 | 0.37 | 0.49 | **0.58** | **0.76** | 0.80 | 0.93 | 0.28 | 0.52 | 0.26 | 0.36 | **0.45** | 0.50 |
| Qwen3-235B-A22B-Instruct | 0.76 | 0.92 | 0.41 | 0.68 | 0.26 | 0.38 | 0.48 | 0.66 | 0.51 | 0.79 | 0.28 | 0.59 | 0.21 | 0.26 | 0.33 | **0.55** |
| GPT5-mini | 0.66 | 0.72 | 0.53 | 0.63 | 0.02 | 0.03 | 0.40 | 0.46 | 0.77 | 0.86 | 0.36 | 0.47 | 0.10 | 0.18 | 0.41 | **0.50** |
| Deepseek-V3 | 0.56 | 0.71 | 0.34 | 0.54 | 0.00 | 0.00 | 0.30 | 0.42 | 0.40 | 0.54 | 0.22 | 0.32 | 0.00 | 0.00 | 0.21 | 0.29 |
| Deepseek-R1 | 0.82 | 0.82 | 0.69 | 0.69 | 0.15 | 0.15 | **0.55** | 0.55 | 0.79 | 0.79 | 0.50 | 0.50 | 0.05 | 0.05 | **0.45** | 0.45 |
| Gemini-2.5-pro | 0.73 | 0.90 | 0.46 | 0.63 | 0.17 | 0.30 | 0.45 | **0.61** | 0.63 | 0.86 | 0.37 | 0.54 | 0.08 | 0.1 | 0.36 | **0.50** |

excel at long-context reasoning? **RQ2:** What leads to over-thinking of LRMs when solving graph algorithm problems? Details of the evaluation setup are provided in Appendix E.

## 3.1 Exploring LRMs' Long-Context Reasoning Ability (RQ1)

With the rapid extension of context windows in modern LRMs, a central question is whether these models can truly perform *understanding and reasoning* over long contexts. Graph algorithm problems provide an ideal testbed: they allow scalable control over input length, and their solutions cannot be trivially extracted from the problem statement. To systematically **explore LRMs' long-context reasoning ability**, we design several experiments. **First**, we vary the number of graph nodes, and alternatively fix the graph structure while extending textual descriptions of nodes, to observe performance changes as input length increases. **Second**, we collect and analyze erroneous responses to identify the underlying causes of degraded performance in long-context reasoning.

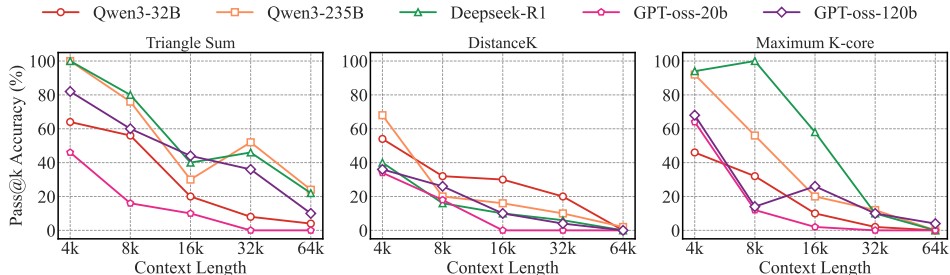

Figure 3: Models pass@k performance across different context length.

**Evaluation by Varying Problem Length.** We examine model performance under different input lengths through two complementary settings. **First**, we increase problem length by scaling the number of nodes and edges within the graphs, and evaluate the resulting performance across varying graph sizes; average accuracy for each reasoning taxonomy is reported in Table 3. **Second**, to decouple the effect of increased task difficulty from larger graph structures, we fix the graph topology and vary token length by modifying the textual descriptions of nodes. Specifically, we select three graph tasks—TRIANGLE SUM, DISTANCEK, and MAXIMUM K-CORE—and construct 50 graphs with 80 nodes and 200 edges each. By adjusting node name length, we generate five datasets with average token lengths of 4k, 8k, 16k, 32k, and 64k. We then evaluate five representative models—Qwen3-32B, Qwen3-235B-Thinking, GPT-oss-20B, GPT-oss-120B, and DeepSeek-R1—under these conditions, with results summarized in Figure 3.

**Error Analysis.** We examine the types of errors that LRMs commonly make when solving graph algorithm problems. Our methodology begins with establishing a taxonomy of **error categories** (see Appendix H.3), followed by the development of an automated **error detection pipeline**. Specifically, to support annotation and improve labeling accuracy, we reorganize LRM responses into coherent **paragraphs**. Specifically, we collect LRMs' responses with incorrect final answers, segment them by "\n\n," and index each paragraph as "$\langle i \rangle$," where $i$ is the segment ID. We then use qwen-2.5-72B to merge paragraphs into **sections** by summarizing each response into tuples of (*start-index*, *end-index*, *summary*), which define the regrouped structure. Next, O3-mini annotates each section with one or more error categories, from which we compute the error-type distribution. In addition, three postgraduate students with competitive programming experience independently review a subset of annotations and confirm their consistency with the LLM-based labels. Detailed prompts and the full transformation process are provided in Appendix K.1 and K.2. Figure 4 presents the error distribution for Qwen3-32B and its non-reasoning counterpart across three taxonomies, with results for other models and detailed case studies in Appendices H.3 and K.3.

① **LRMs cannot effectively handle long-context reasoning.** Our two complementary experiments consistently reveal the limitations of LRMs in processing extended text inputs. As shown in Table 5, model accuracy decreases sharply as graph size grows. For example, Qwen3-32B achieves 87.0% pass@k accuracy on graphs with 31–50 nodes, but drops to 59.0% with 81–120 nodes and further to 47.0% beyond 121–160 nodes. Similar drops are observed across all evaluated models. While this decline could be attributed to increased structural complexity, additional experiments disentangle structural difficulty from textual length. By fixing the graph structure while gradually lengthening node descriptions, we observe a comparable pattern: on the TRIANGLE SUM task, DeepSeek-R1 achieves 100.0% at 4k tokens but only 22.0% at 64k tokens, while GPT-oss-120B falls from 68.0% to 10.0%. Taken together, these results point to a common bottleneck: as context length increases—whether from more graph nodes or longer textual descriptions—LRMs fail to maintain stable reasoning traces, resulting in significant performance degradation.

② **LRMs' long-context reasoning is constrained by three key bottlenecks: coarse step-by-step execution, weak memory capability, and excessive redundancy. First**, LRMs often fail in algorithm execution even after selecting the correct strategy, as Algorithm Execution Errors (AEE) far exceed Algorithm Selection Errors (ASE)—for example, as shown in Figure 4, Qwen3-32B records AEE rates of 35.3% and 48.6% versus ASE rates of only 5.5% and 5.7% in *Exploration* and *Intuition*, respectively. The most frequent issues are State Update Errors and Omissions, indicating that models grasp high-level plans but falter on procedural details. **Second**, LRMs struggle to maintain an accurate representation of graph structure, leading to Graph Memorization Errors (GME): while non-reasoning

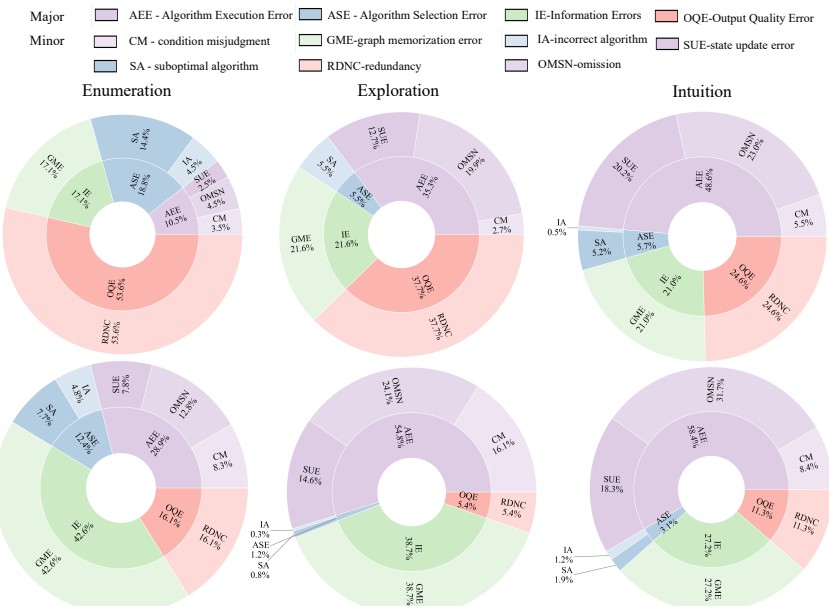

Figure 4: Error type distributions across reasoning taxonomies for Qwen3-32B *(top)* and its non-reasoning variant Qwen3-32B-no-thinking *(bottom)*.

variants reach error rates up to $42.6\%$, reasoning models such as `Qwen3-32B` still exhibit more than $21\%$ GME in *Exploration* and *Intuition*. **Finally**, reasoning models frequently introduce excessive redundancy—for instance, `Qwen3-32B` shows $37.7\%$ redundancy errors in *Exploration*—where verbose and repetitive steps inflate reasoning traces without improving accuracy, and sometimes even cause additional confusion. Importantly, such redundancy reflects a form of *over-thinking*, which we will further analyze in the next section.

③ **Both reasoning and non-reasoning models achieve the best performance on *Enumeration* tasks, followed by *Exploration*, and the worst on *Intuition*.** This observation can be explained from two perspectives. First, within each node-size level, almost all models follow the pattern *Enumeration > Exploration > Intuition*. For instance, as shown at Table 3, at Level-3, GPT-OSS-20B obtains $cons@k$ scores of $0.59$, $0.50$, and $0.39$ on *Enumeration*, *Exploration*, and *Intuition*, respectively. Second, considering how large a graph each model can handle, we find that some remain more robust on *Enumeration*. For example, at Level-6, GPT-5-mini reaches $0.77$ cons@k accuracy on *Enumeration*, but drops below $0.4$ on *Exploration* and to only about $0.1$ on *Intuition*. This indicates that LRMs are relatively competent at systematic, enumeration-based reasoning but struggle substantially with intuition-driven reasoning. To further support this finding, we provide a normalized analysis using the Z-score metric in Appendix H.2.

## 3.2 EXPLORING UNDERLYING MECHANISMS OF LRMS' OVER-THINKING (RQ2)

In error analysis, we observe that reasoning models often produce substantial redundancy when solving graph problems, with responses containing unnecessary information. This phenomenon is closely related to the *over-thinking* behavior characteristic of LRMs, where the model inefficiently expends resources by engaging in excessive verification or redundant checks even after reaching a final answer (Lu et al., 2025; Chen et al., 2024d). Such over-thinking not only generates superfluous tokens, but also leads to wasted computational resources and longer response times. In this section, we investigate LRMs' over-thinking phenomenon in graph algorithm problem solving from two perspectives: *post-answer generation* and *self-verification*.

**Response Partitioning via High-Entropy Tokens.** To facilitate subsequent evaluation, we partition LRM-generated responses into segments. Our partitioning approach is guided by token-level generation entropy, a metric that quantifies the model's uncertainty at each step of the generation process (See Appendix F.3 for more details). High entropy values often coincide with moments of self-verification or strategy-shifting, which may indicate over-thinking (Wang et al., 2025a). By

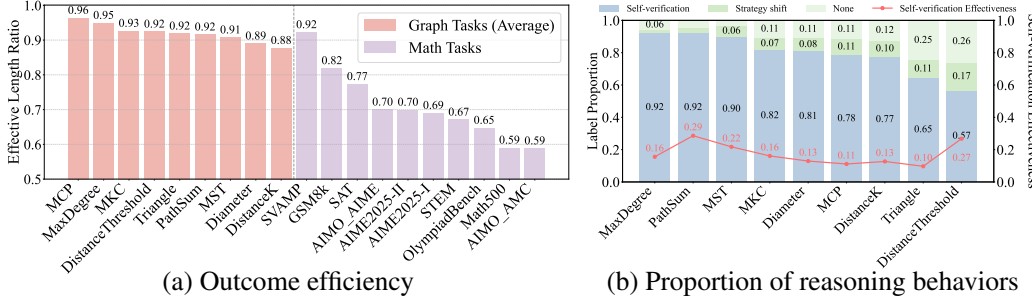

(a) Outcome efficiency  (b) Proportion of reasoning behaviors

Figure 5: Overthinking analysis of QWQ-32B (See Appendix H.5 for more results).

computing this token-level entropy across reasoning traces and visualizing it with a word cloud (Figure 19 in Appendix H.4), we find that tokens such as *"wait"*, *"but"*, and *"so"* consistently exhibit high entropy. This observation aligns with prior studies (Li et al., 2025a; Lu et al., 2025), which have identified these tokens as typical triggers of over-thinking. Therefore, we adopt these tokens as **special partition markers**. Formally, let a response be a token sequence $T = (w_1, w_2, \ldots, w_m)$. If special tokens appear at positions $i_1, i_2, \ldots, i_n$, then $T$ is partitioned into $n + 1$ consecutive **segments**:

$$T \;\Rightarrow\; (w_1, \ldots, w_{i_1-1}), \; (w_{i_1+1}, \ldots, w_{i_2-1}), \; \ldots, \; (w_{i_n+1}, \ldots, w_m). \tag{1}$$

**Post-Answer Generation.** One possible source of over-thinking is **post-answer generation**, i.e., the model keeps producing tokens even after outputting a correct answer (Sui et al., 2025; Chen et al., 2024d; 2025). To quantify this behavior, we adopt the **outcome efficiency** metric (Chen et al., 2024d), the fraction of tokens generated up to the first correct answer relative to the full response length:

$$\zeta_O = \frac{1}{N} \sum_{i=1}^{N} \frac{\hat{T}_i}{T_i}, \tag{2}$$

where $N$ is the number of responses containing the correct answer, $T_i$ the total response length, and $\hat{T}_i$ the tokens up to the first correct answer. We compute $\hat{T}_i$ by prompting Qwen2.5-72B with the original problem, the segmented LRM response, and the ground-truth answer, asking it to locate the first correct segment (Prompts can be seen in Appendix K.1). Outcome efficiency of QWQ-32B on graph and math problems is shown in Figure 5 (a).

**Self-Verification.** Another relevant behavior is **self-verification**, where LRMs attempt to re-check earlier reasoning or conclusions. To analyze this process, we again use Qwen2.5-72B: for each segment $i$, the model classifies it (given all prior segments) as *self-verification*, *strategy-shift*, or *other*. If labeled as self-verification, the segment is further judged as *effective* when it correctly identifies prior errors. Detailed prompts are listed in Appendix K.1, and Figure 5(b) summarizes the distribution of segment types and their effectiveness.

④ **Frequent yet ineffective self-verification is a primary driver of over-thinking in LRMs when solving graph algorithm problems.** As shown in Figure 5, outcome efficiency (a) remains consistently high on graph tasks—above 0.88 in all cases—and is higher than on math tasks. This indicates that LRMs seldom continue generating tokens long after reaching the correct answer, suggesting that post-answer generation is not the main source of over-thinking in graph reasoning. In contrast, panel (b) shows that self-verification occupies a substantial fraction of reasoning traces across different tasks, yet its effectiveness is strikingly low: in most cases fewer than 30% of self-verification attempts successfully detect prior errors. Instead, the majority of these segments simply restate earlier steps or add irrelevant content, inflating response length without improving correctness. These observations together point to frequent but ineffective self-verification as the main mechanism underlying the over-thinking phenomenon in LRMs when solving graph algorithm problems.

## 3.3 DISCUSSION

We further discuss two aspects: (1) the potential benefits of having LRMs generate code and invoke external tools to solve graph algorithm problems, and (2) the extent to which the insights derived from our benchmark generalize beyond the specific tasks and settings considered in this work.

**Solving Graph Algorithms Problems by External Tools.** To better understand the impact of tool usage on our benchmark, we conduct an additional experiment where models are explicitly

Table 4: Model performance (pass@k / cons@k (%)) with and without code-based tool usage.

| Task | Metric | Level-5 (81–120 nodes) | | | Level-6 (121–160 nodes) | | |
|---|---|---|---|---|---|---|---|
| | | Qwen3-32B | Gemini-2.5-pro | GPT5-mini | Qwen3-32B | Gemini-2.5-pro | GPT5-mini |
| Triangle | Base | 76.0 / 66.6 | 90.0 / 80.0 | 56.7 / 50.0 | 60.0 / 42.2 | 86.7 / 83.3 | 86.7 / 70.0 |
| | Code | **96.0 / 93.3** | **94.0 / 92.0** | **99.0 / 99.0** | **92.0 / 88.0** | **90.0 / 88.0** | **99.0 / 99.0** |
| DistanceK | Base | 40.0 / 20.0 | 43.3 / 30.0 | 60.0 / 53.3 | 36.0 / 10.0 | 43.3 / 33.3 | 26.7 / 13.3 |
| | Code | **98.0 / 96.0** | **100.0 / 100.0** | **100.0 / 100.0** | **96.7 / 96.7** | **100.0 / 100.0** | **100.0 / 100.0** |
| MST | Base | 0 / 0 | 16.6 / 3.3 | 3.3 / 0 | 0 / 0 | 0 / 0 | 0 / 0 |
| | Code | **97.3 / 94.7** | **98.7 / 98.0** | **100.0 / 100.0** | **88.0 / 84.0** | **92.0 / 90.0** | **100.0 / 100.0** |

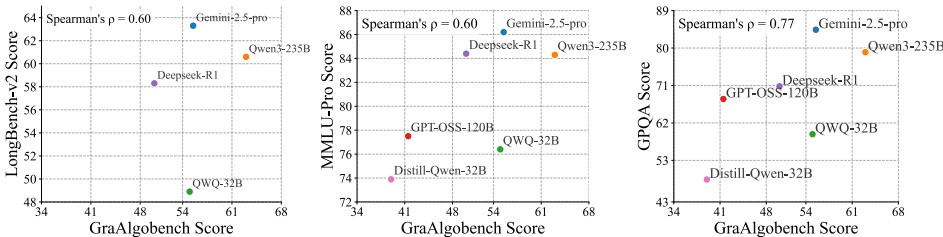

Figure 6: Model-wise correlation with external benchmarks.

instructed to solve problems by writing code and then using a code-execution tool (examples are given in appendix G). Concretely, we compare three LRMs of different strengths (Qwen3-32B, Gemini-2.5-pro, GPT5-mini) on three representative tasks (Triangle, DistanceK, MST), at graph size level-5 and level-6. We report pass@k and cons@k with and without tool calls (See Table 4). Across models and tasks, enabling code and tool usage consistently and substantially improves performance, especially on the most challenging instances. However, delegating the hardest parts of the problem—such as graph memorization and step-by-step algorithmic execution—to external tools no longer directly evaluates the model's intrinsic reasoning ability. For this reason, we do not adopt the code-generation setting as our primary evaluation protocol.

**Conclusion Generalization.** To verify that our findings generalize beyond graph-specific scenarios, we conduct two additional experiments: (1) we measure model-wise correlations between GraAlgoBench and established benchmarks spanning long-context and general reasoning—LongBench-v2 (Bai et al., 2025), GPQA (Rein et al., 2024), and MMLU-Pro (Wang et al., 2024) (Figure 6); and (2) we vary the graph serialization format (edge lists, adjacency lists, Markdown tables, and numeric node IDs; examples in Appendix G) and compare performance across formats (Table 6 in Appendix I). According to standard guidelines for interpreting correlation coefficients (Mukaka, 2012), values in the range 0.50–0.79 indicate a moderate to strong association. The observed correlations between GraAlgoBench and GPQA ($\rho = 0.77$) and between GraAlgoBench and LongBench-v2 ($\rho = 0.60$) therefore suggest that the *complex reasoning skills* assessed by our benchmark align closely with those underlying high-level general reasoning and long-context understanding. In addition, across graph sizes, models achieve *similar performance under all four representation formats*, indicating that performance is *largely independent of input format* and primarily reflects intrinsic reasoning ability rather than superficial properties of the graph encoding.

## 4 CONCLUSION

We introduced GRALGOBENCH, a benchmark that leverages graph algorithm problems to probe reasoning in LRMs with effective long-context evaluation, fine-grained difficulty control, and programmatic assessment. Our results highlight two central limitations: accuracy drops substantially as context length grows, and over-thinking emerges from frequent yet ineffective self-verification. By uncovering these challenges, GRALGOBENCH establishes graph algorithms as a practical and rigorous foundation for advancing the robustness and efficiency of future reasoning models.

ACKNOWLEDGEMENTS

This work is supported by NSFC Grant No.62572418.

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

## A   Ethic Statement

This work focuses on benchmarking large reasoning models through graph algorithm problems. The study does not involve human subjects, personally identifiable information, or sensitive data. All experiments are conducted using publicly available models and synthetic datasets. We have taken care to ensure that the research does not promote harmful applications and is intended solely for advancing understanding of reasoning in AI systems.

## B   Reproductivity Statement

We provide detailed task definitions, dataset generation procedures, and evaluation protocols to ensure full reproducibility. All experiments are conducted with publicly available models and standardized settings, as described in Section E. The complete codebase, including dataset generators and evaluation scripts, is released at the project repository to facilitate replication and future extensions.

## C   The Use of Large Language Models (LLMs)

During the preparation of this work, we made use of LLMs in several stages of writing and experimentation:

- **Language Refinement:** LLMs (such as OpenAI's ChatGPT) were employed to improve the readability and academic style of the manuscript, helping to enhance clarity and consistency.

- **Evaluation Assistance:** We employed LLMs as judges to help with evaluation. These models supported the assessment process and complemented our programmatic evaluation pipeline.

- **Programming Support:** The Cursor environment, which incorporates AI assistance, was used for generating, modifying, and refactoring segments of the codebase. This contributed to faster implementation and better code quality.

## D   Related Works

**Benchmarking LLMs on Graph Algorithm Problems.** Graph algorithm problems demand a deep understanding of structural information and long-range multi-step reasoning, making them particularly challenging for LLMs. For this reason, they have been widely used in prior studies to evaluate LLM capabilities. NLGraph (Wang et al., 2023), GPT4Graph (Guo et al., 2023), and GraphQA (Fatemi et al., 2023) are among the earliest works, assessing LLMs on graph algorithm problems by encoding tasks with LLMs alone or with a GNN–LLM combination. Building on this line of research, LLM4DyG (Zhang et al., 2024b) examines tasks on dynamic graphs, while GraphInstruct (Luo et al., 2024) provides a broader benchmark with 21 tasks ranging from node-level to graph-level. GraphArena (Tang et al., 2024) evaluates LLMs on real-world graphs with more fine-grained metrics, and VisionGraph (Li et al., 2024) explores their capabilities on image graphs. Most recently, GraCore (Yuan et al., 2024) evaluates LLMs from the perspectives of graph reasoning and graph understanding, and GraphOmni (Xu et al., 2025) investigates their performance across diverse graph types, serialization formats, and prompting schemes. However, none of these works evaluate the performance of O1-like large reasoning models (LRMs) or analyze the challenges LRMs face when solving graph algorithm problems, which motivates our study.

**Evaluating Large Reasoning Models.** The emergence of O1-like models, equipped with long chains of thought and advanced reasoning strategies such as self-verification, strategy-shift, and backtracking, has substantially advanced the reasoning capabilities of LLMs. This progress has motivated extensive efforts to evaluate LRMs across diverse domains (Petrov et al., 2025; He et al., 2024; Huang et al., 2024; Gulati et al., 2024; Glazer et al., 2024; Li et al., 2025b; Huang et al., 2025; Chen et al., 2024d; Wang et al., 2023; Li et al., 2022; Dai et al., 2024; Yang et al., 2025b; Yu et al., 2024; Wei et al., 2025; He et al., 2025b; Zheng et al., 2025; White et al., 2024; Suzgun et al., 2022; Lin et al., 2025). Among these, mathematics has become the predominant benchmarking arena, yielding many new insights into the strengths and weaknesses of LRMs. For instance, Putnam-AXIOM (Gulati

et al., 2024), FrontierMath (Glazer et al., 2024), and OlympiadBench (Sun et al., 2025) introduce increasingly challenging problems to probe the reasoning limits of models, while other works (Chen et al., 2024d; Wang et al., 2023) analyze phenomena such as *over-thinking* and *under-thinking* and explore potential mitigations. However, no prior work has systematically evaluated LRMs on *graph algorithm problems*, which demand structural reasoning and memory—skills missing from existing benchmarks. To close this gap, we present **GRALGOBENCH**, the first graph-based benchmark for rigorous and application-relevant LRMs evaluation.

## E  EXPERIMENTAL SETUP

**Models**: We evaluate a broad range of reasoning models as well as non-reasoning models in order to ensure comprehensive coverage. For open-source models, we include the Qwen3 series (Yang et al., 2025a), QwQ (Team, 2025b), Qwen-2.5 series (Team, 2024), OpenThinker2 series (Team, 2025a), Skywork series (He et al., 2025a), Light-R1 series (Wen et al., 2025), Llama-3.3 series (Dubey et al., 2024), GPT-OSS series (Agarwal et al., 2025), Deepseek-V3 (Liu et al., 2024), and Deepseek-R1 series (Guo et al., 2025). For closed-source models, we evaluate GPT5-mini (OpenAI, 2025) and Gemini-2.5-pro (Comanici et al., 2025).

**Evaluation Metrics.** Our evaluation framework is designed in a systematic manner. For each problem, we collect 8 responses from every comparison model. For a few models (i.e., DeepSeek-V3/R1, GPT-5-mini, Gemini-2.5-pro), we reduce the number of responses to 4 due to computational limitations and budget constraints. We adopt two evaluation metrics: *Pass@k* and *Cons@k*. For the *Pass@k* metric, a sample is regarded as correct if at least one of the $k$ responses generated by the LRM matches the ground-truth answer. For the *Cons@k* metric, we apply majority voting to derive a consensus answer for each problem, and then compute the average accuracy across the dataset. To ensure standardized evaluation, each problem is paired with a fixed reference answer, and correctness is assessed through exact string matching. We enforce that LRMs output their final answers in the \boxed{} format. For decoding hyperparameters, we follow prior work (Guo et al., 2025; Team, 2025b), setting temperature, top_p, min_p, and max_token to 0.6, 0.95, 0, and 32768, respectively.

## F  DEFINITION AND EXPLANATION

### F.1  TASK DEFINITION

(1) Maximum Degree node (easy): Given an undirected graph $\mathcal{G} = (\mathcal{V}, \mathcal{E})$, the task is to identify the node $v \in \mathcal{V}$ with the maximum degree and output its degree $\deg(v) = \max_{u \in \mathcal{V}} \deg(u)$.

**Optimal complexity:** $O(|V| + |E|)$ using a single scan of graph.

(2) Maximum Weight Triangle (medium): Given an undirected graph $\mathcal{G} = (\mathcal{V}, \mathcal{E})$ with a positive node weight function $w : \mathcal{V} \to \mathbb{R}^+$, the task is to find a triangle $\{v_1, v_2, v_3\}$ that maximizes the sum of their weights $\sum_{i=1}^{3} w(v_i)$, and output this maximum sum.

**Optimal complexity:** $O(|V|^3)$ by brute-force enumeration of all triples of vertices.

(3) Maximum Clique Problem (hard): Given an undirected graph $\mathcal{G} = (\mathcal{V}, \mathcal{E})$, the task is to find a subset of nodes $\mathcal{V}' \subseteq \mathcal{V}$ that forms a clique (i.e., every pair of nodes in $\mathcal{V}'$ is connected by an edge) with the maximum possible size $|\mathcal{V}'|$. The output is this size.

**Optimal complexity:** NP-hard. The best known exact exponential-time algorithms (e.g., branch-and-bound, branch-and-reduce) run in about $O(1.1996^{|V|})$.

(4) PathSum (easy): Given an undirected tree $\mathcal{T} = (\mathcal{V}, \mathcal{E})$ with a root $r$ and edge weights $w : \mathcal{E} \to \mathbb{R}$, the task is to count the number of paths from the root $r$ to a leaf node $l$ such that the sum of edge weights $\sum_{(u,v) \in \text{path}(r,l)} w(u, v)$ exceeds a given threshold $\tau$.

**Optimal complexity:** $O(|V|)$ using depth-first search (DFS).

(5) Distance-$k$ (medium): Given an undirected graph $\mathcal{G} = (\mathcal{V}, \mathcal{E})$ and a specific node $v \in \mathcal{V}$, the task is to find the number of nodes $u \in \mathcal{V}$ such that their shortest path distance $d(u, v) \leq k$.

**Optimal complexity:** $O(|V| + |E|)$ using BFS.

(6) `Diameter (hard)`: Given an unweighted undirected graph $\mathcal{G} = (\mathcal{V}, \mathcal{E})$, the task is to find the longest shortest path between any two nodes $u, v \in \mathcal{V}$ and output its length, i.e., $\max_{u,v \in \mathcal{V}} d(u,v)$.

**Optimal complexity:** $O(|V|(|V| + |E|))$ using BFS (or DFS) from every node.

(7) `Maximum k-core (easy)`: Given an undirected graph $\mathcal{G} = (\mathcal{V}, \mathcal{E})$ and an integer $k$, the task is to find a subgraph $\mathcal{G}' = (\mathcal{V}', \mathcal{E}')$ where for every node $v \in \mathcal{V}'$, its degree within the subgraph $\deg_{\mathcal{G}'}(v) \geq k$. The goal is to find the subgraph with the maximum number of nodes $|\mathcal{V}'|$ and output this size.

**Optimal complexity:** $O(|V| + |E|)$ using the peeling algorithm.

(8) `Minimum Spanning Tree (medium)`: Given a connected, undirected graph $\mathcal{G} = (\mathcal{V}, \mathcal{E})$ with positive edge weights $w : \mathcal{E} \to \mathbb{R}^+$, the task is to find a spanning tree (a subset of edges $\mathcal{E}' \subseteq \mathcal{E}$ that connects all nodes) with the minimum possible total edge weight $\sum_{(u,v) \in \mathcal{E}'} w(u,v)$. The output is this minimum total weight.

**Optimal complexity:** Kruskal's algorithm: $O(|E| \log |V|)$.

(9) `Distance Threshold (hard)`: Given an undirected graph $\mathcal{G} = (\mathcal{V}, \mathcal{E})$ with positive edge weights, the task is to find the node $v^* = \arg\min_{v \in \mathcal{V}} |\{u \in \mathcal{V} \setminus \{v\} \mid d(u,v) \leq \tau\}|$, where $\tau$ is a given threshold, and output the node $v^*$.

**Optimal complexity:** worst case $O(|V||E| \log |V|)$ using Dijkstra's algorithm from each node (truncated at distance $\tau$).

## F.2 ERROR TYPE CLASSIFICATION

We categorize model errors into four major classes:

1. *Algorithm Selection Error.* These errors capture whether LRMs truly understand how to select an appropriate algorithm for a given problem.

    (I) **Incorrect algorithm:** The chosen algorithm cannot produce the correct solution for the problem.

    (II) **Suboptimal algorithm:** The algorithm is intuition (not guaranteed to produce the optimal solution) or computationally inefficient.

2. *Algorithm Execution Error.* These errors occur when LRMs make mistakes during the execution of an algorithm.

    (I) **State update error:** Incorrectly updating intermediate results, either by missing necessary information or storing extraneous data.

    (II) **Missing elements:** Failure to include essential steps, such as neglecting to traverse all nodes or edges.

    (III) **Condition misjudgment:** Errors in conditional checks, e.g., incorrect `if`-statements or loop conditions.

3. *Output Quality Error.* In these cases, the model produces the correct answer but with low-quality output. We primarily focus on:

    (I) **Redundancy:** Unnecessary repetition or superfluous information in the solution.

4. *Information Error.* These errors arise when the model fails to capture or recall critical problem-specific information. We focus on:

    (I) **Graph memorization error:** Misunderstanding or incorrectly recalling the graph structure, such as introducing extra nodes/edges or omitting existing ones.

F.3 DEFINITION OF ENTROPY

Formally, we define the entropy $H_t$ at generation step $t$ as the Shannon entropy of the model's predicted probability distribution over its vocabulary:

$$H_t = -\sum_{j=1}^{V} p_{t,j} \log p_{t,j} \tag{3}$$

Here, $p_t = (p_{t,1}, \ldots, p_{t,V})$ represents the probability distribution over the entire vocabulary $V$. This distribution is produced by the language model, denoted as $\pi_\theta$, conditioned on the input query $q$ and the preceding token sequence $o_{<t}$. Specifically, $p_t$ is computed by applying a temperature-scaled Softmax function to the model's raw output logits $z_t$:

$$p_t = \pi_\theta(\cdot \mid q, o_{<t}) = \text{Softmax}\left(\frac{z_t}{T}\right) \tag{4}$$

## G ADDITIONAL TASKS DESCRIPTION

---

**Problem Description G.1: Diameter**

You are required to calculate the diameter of an undirected knowledge graph. The diameter of a graph is the greatest shortest-path distance between any two nodes in the graph.

**Problem to Solve**

- Entities in this knowledge graph: Seine-et-Marne, Vendrest, Paris, Clickteam, Soignolles-en-Brie, Saint-Fiacre Seine-et-Marne, France, Didier Julia
- The relationships between these entities are as follows:
    - Seine-et-Marne is connected to Vendrest via the relationship department.
    - Seine-et-Marne is connected to Saint-Fiacre Seine-et-Marne via the relationship department.
    - Seine-et-Marne is connected to Didier Julia via the relationship region.
    - Seine-et-Marne is connected to Soignolles-en-Brie via the relationship department.
    - Seine-et-Marne is connected to France via the relationship country.
    - Vendrest is connected to France via the relationship country.
    - Paris is connected to France via the relationship capital.
    - Paris is connected to Clickteam via the relationship locationCity.
    - Paris is connected to Didier Julia via the relationship birthPlace.
    - Soignolles-en-Brie is connected to France via the relationship country.
    - Saint-Fiacre Seine-et-Marne is connected to France via the relationship country.
    - France is connected to Didier Julia via the relationship region.

Please determine the diameter of this network and output the diameter in the following format: `\boxed{n}`

---

**Problem Description G.2: Distance Threshold**

You are given an undirected weighted graph representing the airport network, where nodes represent airports (codes: DLY, TAH, HIR, IPA, VLI, BNE, NAN, AKL, LNE, FTA, AWD, EAE, LNB, AUY) and edges represent direct flights between airports. The weight of each edge is the distance between two airports.

- The list of airports: DLY, TAH, HIR, IPA, VLI, BNE, NAN, AKL, LNE, FTA, AWD, EAE, LNB, AUY
- The direct flights (edges) are: `[DLY, IPA, 34], [DLY, VLI, 139], [TAH, AUY, 105], [TAH, AWD, 46], [TAH, FTA, 105], [TAH, IPA, 64], [TAH, VLI, 217], [HIR, VLI, 1282], [HIR, BNE, 2126], [HIR, NAN, 2093], [IPA, VLI, 167], [VLI, NAN, 966], [VLI, AKL,`

---

```
        2238], [VLI, BNE, 1893], [VLI, EAE, 67], [VLI, LNB, 125],
        [VLI, LNE, 204], [BNE, AKL, 2295], [BNE, NAN, 2710], [NAN,
        AKL, 2156],[FTA, AWD, 72]
```
- The distance threshold is 284.

For each airport, the distance to another airport is defined as the sum of the weights (distances) along the shortest path connecting them.

Your task is: Return the airport code with the smallest number of other airports that can be reached with a shortest path distance no more than the threshold. If there are multiple such airports, return the one with the lexicographically largest code.

Present your answer in the following format: airport_code . The airport_code is the airport code.

---

**Problem Description G.3: Maximum Clique Problem**

You are required to solve the Maximum Clique Problem for an undirected wikipedia network. In this network, nodes represent wikipedia articles and edges represent hyperlinks between articles. Your objective is to find the largest subset of nodes such that every pair of vertices in this subset is connected by an edge.

- Articles in the network: Eugene Domingo, Philippines, Talk show, David Cook (singer), Live television, BB Gandanghari, Cool Center, Katya Santos, David Archuleta, Sadako, Gladys Guevarra, List of Philippine television shows

- Hyperlinks between these articles: Eugene Domingo and Cool Center, Eugene Domingo and Philippines, Philippines and Cool Center, Philippines and Gladys Guevarra, Philippines and BB Gandanghari, Philippines and List of Philippine television shows, Philippines and David Archuleta, Talk show and Cool Center, David Cook (singer) and Cool Center, David Cook (singer) and David Archuleta, Live television and Cool Center, BB Gandanghari and Cool Center, Cool Center and List of Philippine television shows, Cool Center and Sadako, Cool Center and David Archuleta, Cool Center and Gladys Guevarra, Cool Center and Katya Santos.

Identify the clique with the maximum number of articles in this network. Present your answer in the following format: `\boxed{k}`. k is the number of articles of this clique.

---

**Problem Description G.4: Minimum Spanning Tree**

You are required to solve the Minimum Spanning Tree Problem for an undirected street network. In this network, nodes represent streets (e.g., street IDs) and edges represent intersections between streets. The weight of each edge is the distance between two streets.

- Streets in the network: Mulgray Avenue, Vanessa Avenue, Eames Avenue, Gabrielle Avenue, Justine Avenue, Coronation Road, Turon Avenue, Jasper Road, Louise Avenue, Glanmire Road, Hilda Road, Seven Hills Road

- Intersections between these streets: Mulgray Avenue and Coronation Road (weight: 6), Mulgray Avenue and Jasper Road (weight: 3), Vanessa Avenue and Jasper Road (weight: 1), Eames Avenue and Hilda Road (weight: 7), Eames Avenue and Jasper Road (weight: 10), Gabrielle Avenue and Justine Avenue (weight: 6), Gabrielle Avenue and Turon Avenue (weight: 7), Justine Avenue and Jasper Road (weight: 1), Justine Avenue and Turon Avenue (weight: 6), Coronation Road and Jasper Road (weight: 9), Turon Avenue and Jasper Road (weight: 4), Jasper Road and Glanmire Road (weight: 5), Jasper Road and Hilda Road (weight: 1), Jasper Road and Louise Avenue (weight: 5), Jasper Road and Seven Hills Road (weight: 2), Hilda Road and Seven Hills Road (weight: 1).

Identify the minimum spanning tree of this network. The minimum spanning tree is a subset of edges in a connected, weighted graph that connects all the vertices together with the smallest possible total edge weight and without any cycles.

Present your answer in the following format: `\boxed{n}`, where n is the sum of the weights of the edges in the minimum spanning tree.

**Problem Description G.5: PathSum**

You are given a binary tree representing a co-authorship network. Each node is an author, and each edge represents a co-authorship relationship, with the edge's weight indicating the number of papers co-authored by the two authors. Find all paths from the root author to any leaf author such that the sum of the edge weights (i.e., the total number of co-authored papers along the path) is greater than the given value.

- Authors in the network: Tarkan Tan, Ruud H. Teunter, Hui-Ming Wee, Samuel Yáñez Artus, Po-Chung Yang, Asoke Kumar Bhunia, Anne Barros, Yu-Chung Tsao, Samiran Chattopadhyay, A. K. Bhunia, Michel Roussignol, Mahmood Shafiee, Shib Sankar Sana, Gwo-Ji Sheen

- Co-authorship relationships (with number of co-authored papers): Tarkan Tan and Ruud H. Teunter (co-authored 2 papers), Ruud H. Teunter and Hui-Ming Wee (co-authored 4 papers), Ruud H. Teunter and Samuel Yáñez Artus (co-authored 9 papers), Hui-Ming Wee and Po-Chung Yang (co-authored 4 papers), Hui-Ming Wee and Asoke Kumar Bhunia (co-authored 5 papers), Samuel Yáñez Artus and Anne Barros (co-authored 6 papers), Po-Chung Yang and Yu-Chung Tsao (co-authored 4 papers), Asoke Kumar Bhunia and Samiran Chattopadhyay (co-authored 6 papers), Asoke Kumar Bhunia and A. K. Bhunia (co-authored 1 papers), Anne Barros and Michel Roussignol (co-authored 9 papers), Anne Barros and Mahmood Shafiee (co-authored 5 papers), Yu-Chung Tsao and Shib Sankar Sana (co-authored 10 papers), Yu-Chung Tsao and Gwo-Ji Sheen (co-authored 3 papers).

- The root of the tree is Tarkan Tan.

- The target value is 19.

Present your answer in the following format: `\boxed{n}`. n is the number of qualifying paths.

**Problem Description G.6: Maximum Weight Triangle**

You are required to solve the Maximum Triangle Sum Problem for an undirected wikipedia network. In this network, nodes represent wikipedia articles and edges represent hyperlinks between articles. Each node is assigned a weight. Your objective is to find the triangle with the maximum sum of weights of its three nodes.

- Articles in the network: Montebelluna (weight: 1), Massimo Mascioletti (weight: 8), Gianluca Faliva (weight: 4), Eppelheim (weight: 7), United States (weight: 6), Alberto Rebecca (weight: 10), October 15 (weight: 6), Rugby union (weight: 1), Italy (weight: 9), List of football clubs in Italy (weight: 9), Manuel Dallan (weight: 8), Brad Johnstone (weight: 1)

- Hyperlinks between these articles: Montebelluna and List of football clubs in Italy, Montebelluna and Eppelheim, Montebelluna and Alberto Rebecca, Montebelluna and Italy, Montebelluna and Manuel Dallan, Massimo Mascioletti and Brad Johnstone, Massimo Mascioletti and Gianluca Faliva, Gianluca Faliva and Italy, Gianluca Faliva and Brad Johnstone, Gianluca Faliva and Manuel Dallan, Gianluca Faliva and Rugby union, Eppelheim and Italy, United States and Italy, United States and October 15, Alberto Rebecca and Italy, October 15 and Italy, October 15 and Manuel Dallan, Rugby union and Manuel Dallan, Italy and Manuel Dallan.

Identify the triangle with the maximum sum of weights of its three nodes in this network. Present your answer in the following format: `\boxed{n}`. n is the maximum sum of weights of its three nodes.

**Problem Description G.7: Minimum Spanning Tree (Adjacency List)**

You are required to solve the Minimum Spanning Tree Problem for an undirected street network. In this network, nodes represent streets (e.g., street IDs) and edges represent intersections between streets. The weight of each edge is the distance between two streets.

- Streets in the network: Maplin Road, Golden Plover Close, Widgeon Close, Downs Court Road, Greyfields Close, Pheasant Close, Coolfin Road, Burrard Road, Freemasons Road, Ethel Road, Partridge Knoll

- Adjacency list representation (each street followed by its connected streets): Maplin Road (weight: 5): Freemasons Road (weight: 9), Golden Plover Close (weight: 6), Pheasant Close (weight: 8), Widgeon Close (weight: 7)

  Golden Plover Close (weight: 4): Maplin Road (weight: 5), Widgeon Close (weight: 7)

  Widgeon Close (weight: 7): Maplin Road (weight: 5), Golden Plover Close (weight: 4)

  Downs Court Road (weight: 3): Partridge Knoll (weight: 10)

  Greyfields Close (weight: 6): Partridge Knoll (weight: 10)

  Pheasant Close (weight: 8): Maplin Road (weight: 5), Partridge Knoll (weight: 10)

  Coolfin Road (weight: 2): Freemasons Road (weight: 9)

  Burrard Road (weight: 4): Freemasons Road (weight: 9)

  Freemasons Road (weight: 9): Maplin Road (weight: 5), Coolfin Road (weight: 2), Burrard Road (weight: 4), Ethel Road (weight: 3) Ethel Road (weight: 3): Freemasons Road (weight: 9)

  Partridge Knoll (weight: 10): Downs Court Road (weight: 3), Greyfields Close (weight: 6), Pheasant Close (weight: 8)

Identify the minimum spanning tree of this network. The minimum spanning tree is a subset of edges in a connected, weighted graph that connects all the vertices together with the smallest possible total edge weight and without any cycles.

Present your answer in the following format: \boxed{n}, where n is the sum of the weights of the edges in the minimum spanning tree.

---

**Problem Description G.8: Minimum Spanning Tree (Markdown Table)**

You are required to solve the Minimum Spanning Tree Problem for an undirected street network. In this network, nodes represent streets (e.g., street IDs) and edges represent intersections between streets. The weight of each edge is the distance between two streets.

| Street 1            | Street 2          | Weight |
|---------------------|-------------------|--------|
| Heritage Court      | Darcey Road       | 3      |
| Jordana Place       | Darcey Road       | 6      |
| Neville Court       | Darcey Road       | 7      |
| Silky Oak Place     | Darcey Road       | 3      |
| Candlebush Crescent | Darcey Road       | 1      |
| Candlebush Crescent | Henley Close      | 4      |
| Candlebush Crescent | Lemonwood Place   | 10     |
| Candlebush Crescent | Melaleuca Close   | 9      |
| Darcey Road         | Castlewood Drive  | 10     |
| Darcey Road         | Crane Road        | 4      |
| Darcey Road         | Henley Close      | 5      |
| Darcey Road         | Jarrah Place      | 10     |
| Crane Road          | Castlewood Drive  | 8      |

Find the minimum spanning tree of this street network and output the sum of the weights of the edges in the MST. Present your answer in the following format: '$n$', where 'n' is the sum of the weights of the edges in the MST.

## Problem Description G.9: Minimum Spanning Tree (Numeric ID)

You are given an undirected weighted graph with 12 nodes labeled from 0 to 11.

- Edges (format: [Node1, Node2, weight]): [0, 5, 3], [1, 5, 6], [2, 5, 7], [3, 5, 3], [4, 5, 1], [4, 6, 4], [4, 7, 10], [4, 8, 9], [5, 11, 10], [5, 10, 4], [5, 6, 5], [5, 9, 10], [10, 11, 8]

Find the minimum spanning tree of this graph and output the sum of the weights of the edges in the MST. Present your answer in the following format: '$\boxed{n}$', where n is the sum of the weights of the edges in the minimum spanning tree.

## Problem Description G.10: Minimum Spanning Tree (Code Version)

You are required to solve the Maximum Triangle Sum Problem for an undirected wikipedia network. In this network, nodes represent wikipedia articles and edges represent hyperlinks between articles. Each node is assigned a weight. Your objective is to find the triangle with the maximum sum of weights of its three nodes.

- Streets in the network: Mulgray Avenue, Vanessa Avenue, Eames Avenue, Gabrielle Avenue, Justine Avenue, Coronation Road, Turon Avenue, Jasper Road, Louise Avenue, Glanmire Road, Hilda Road, Seven Hills Road

- Intersections between these streets: Mulgray Avenue and Coronation Road (weight: 6), Mulgray Avenue and Jasper Road (weight: 3), Vanessa Avenue and Jasper Road (weight: 1), Eames Avenue and Hilda Road (weight: 7), Eames Avenue and Jasper Road (weight: 10), Gabrielle Avenue and Justine Avenue (weight: 6), Gabrielle Avenue and Turon Avenue (weight: 7), Justine Avenue and Jasper Road (weight: 1), Justine Avenue and Turon Avenue (weight: 6), Coronation Road and Jasper Road (weight: 9), Turon Avenue and Jasper Road (weight: 4), Jasper Road and Glanmire Road (weight: 5), Jasper Road and Hilda Road (weight: 1), Jasper Road and Louise Avenue (weight: 5), Jasper Road and Seven Hills Road (weight: 2), Hilda Road and Seven Hills Road (weight: 1).

Identify the minimum spanning tree of this network. The minimum spanning tree is a subset of edges in a connected, weighted graph that connects all the vertices together with the smallest possible total edge weight and without any cycles.

Solve this problem using Python code only. Wrap your Python code in the following format:

```python
# Your code here

print("\\boxed{n}")
```

Here, 'n' is the sum of the weights of the edges in the minimum spanning tree.

# H ADDITIONAL EXERIMENTS RESULTS

## H.1 ALGORITHM RATIO CALCULATION

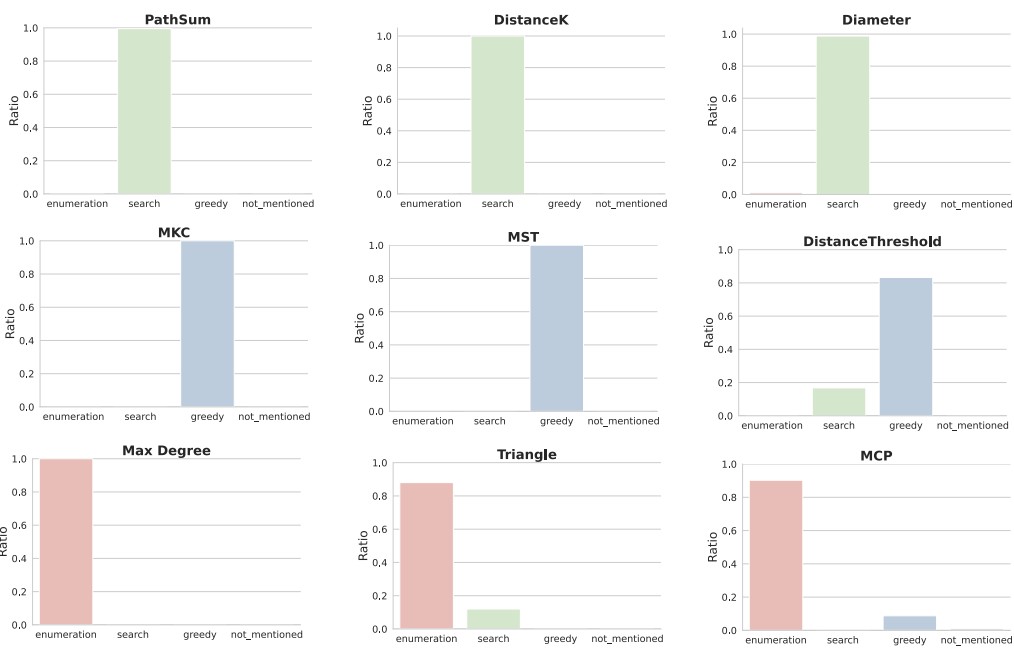

Figure 7: Algorithm ratio of Qwen3-32B.

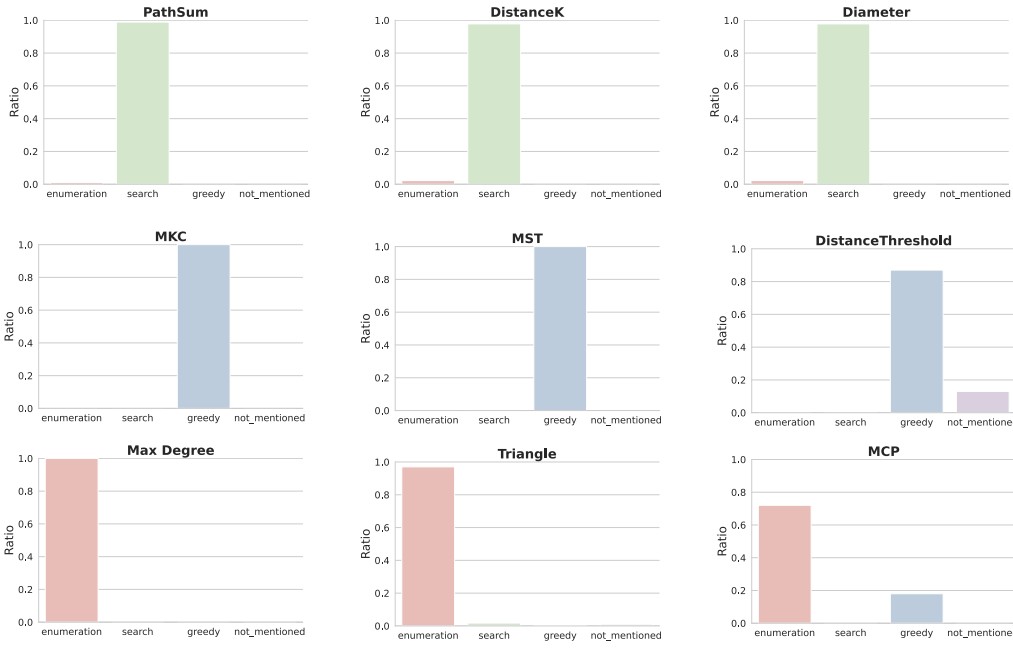

Figure 8: Algorithm ratio of Gemini-2.5-Pro.

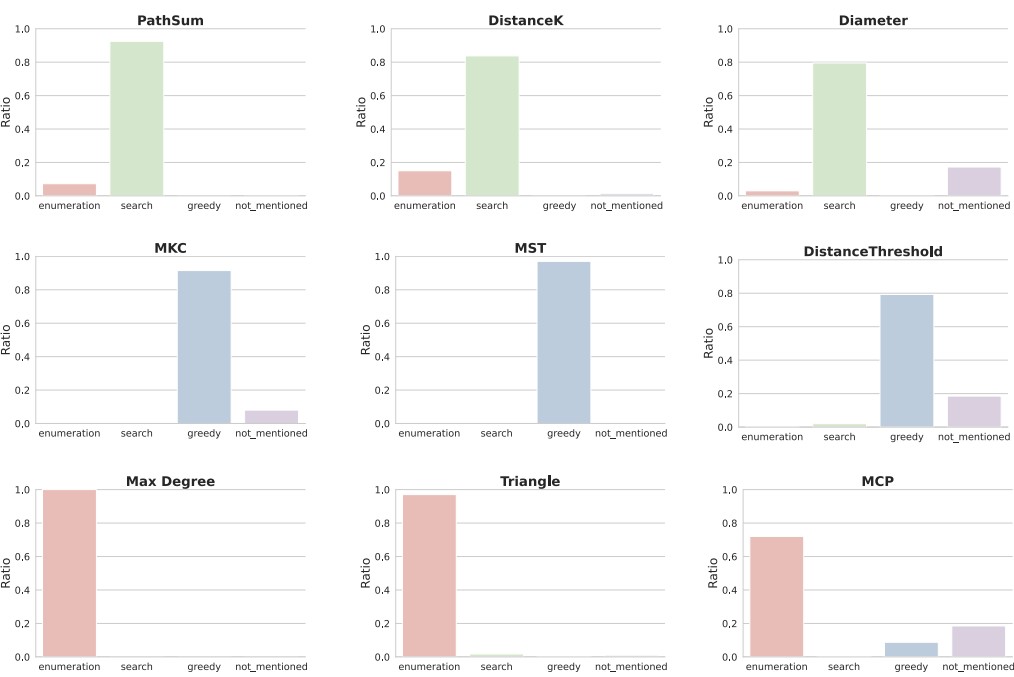

Figure 9: Algorithm ratio of Llama-3.3-70B.

## H.2 PERFORMANCE ACROSS GRAPH ALGORITHM CATEGORIES (RQ1)

Table 5: Evaluation results of Level-1 and Level-2 groups.

| Models | Level-1 (8-15 nodes) | | | | | | | | Level-2 (16-30 nodes) | | | | | | | |
|---|---|---|---|---|---|---|---|---|---|---|---|---|---|---|---|---|
| | Enumeration | | Exploration | | Intuition | | Avg. | | Enumeration | | Exploration | | Intuition | | Avg. | |
| | c@k | p@k | c@k | p@k | c@k | p@k | c@k | p@k | c@k | p@k | c@k | p@k | c@k | p@k | c@k | p@k |
| Skywork-OR1-7B-Preview | 0.50 | 0.91 | 0.63 | 0.93 | 0.31 | 0.67 | 0.48 | 0.84 | 0.20 | 0.67 | 0.20 | 0.67 | 0.08 | 0.39 | 0.16 | 0.58 |
| Light-R1-7B-DS | 0.37 | 0.79 | 0.41 | 0.82 | 0.24 | 0.61 | 0.34 | 0.74 | 0.07 | 0.48 | 0.09 | 0.49 | 0.04 | 0.23 | 0.07 | 0.40 |
| Distill-Qwen-7B | 0.43 | 0.88 | 0.47 | 0.87 | 0.30 | 0.68 | 0.40 | 0.81 | 0.17 | 0.68 | 0.14 | 0.63 | 0.02 | 0.32 | 0.11 | 0.54 |
| Qwen2.5-7B | 0.21 | 0.47 | 0.29 | 0.65 | 0.06 | 0.20 | 0.19 | 0.44 | 0.12 | 0.45 | 0.13 | 0.29 | 0.04 | 0.13 | 0.10 | 0.29 |
| OpenThinker-7B | 0.57 | 0.88 | 0.54 | 0.91 | 0.41 | 0.79 | 0.51 | 0.86 | 0.23 | 0.83 | 0.17 | 0.61 | 0.10 | 0.37 | 0.17 | 0.60 |
| Qwen3-8B-no-thinking | 0.57 | 0.85 | 0.70 | 0.96 | 0.30 | 0.65 | 0.52 | 0.82 | 0.22 | 0.62 | 0.35 | 0.75 | 0.03 | 0.29 | 0.20 | 0.55 |
| Qwen3-8B | 0.89 | 0.99 | 0.77 | 0.99 | 0.96 | 1.00 | **0.87** | **0.99** | 0.63 | 0.93 | 0.63 | 0.93 | 0.66 | 0.93 | **0.64** | **0.93** |
| Light-R1-32B | 0.97 | 1.00 | 0.97 | 1.00 | 0.95 | 0.99 | 0.96 | **1.00** | 0.85 | 0.99 | 0.89 | 0.97 | 0.71 | 0.93 | 0.82 | 0.96 |
| Skywork-OR1-32B | 0.91 | 0.99 | 1.00 | 1.00 | 0.89 | 1.00 | 0.94 | **1.00** | 0.93 | 0.99 | 0.97 | 0.99 | 0.87 | 0.98 | **0.92** | **0.99** |
| Distill-Qwen-32B | 0.93 | 0.99 | 0.96 | 0.99 | 0.96 | 1.00 | 0.95 | 0.99 | 0.75 | 0.98 | 0.73 | 0.95 | 0.62 | 0.89 | 0.70 | 0.94 |
| Qwen2.5-32B | 0.25 | 0.73 | 0.42 | 0.85 | 0.36 | 0.61 | 0.34 | 0.73 | 0.11 | 0.56 | 0.18 | 0.52 | 0.16 | 0.39 | 0.15 | 0.49 |
| OpenThinker-32B | 0.97 | 0.99 | 0.97 | 1.00 | 0.95 | 1.00 | 0.96 | **1.00** | 0.85 | 0.98 | 0.85 | 0.97 | 0.62 | 0.94 | 0.77 | 0.96 |
| QWQ-32B | 0.98 | 1.00 | 1.00 | 1.00 | 0.99 | 1.00 | **0.99** | **1.00** | 0.93 | 0.99 | 0.92 | 0.99 | 0.91 | 0.98 | **0.92** | **0.99** |
| Qwen3-32B | 0.97 | 0.99 | 0.98 | 1.00 | 0.99 | 1.00 | 0.98 | **1.00** | 0.89 | 0.99 | 0.83 | 0.98 | 0.84 | 0.96 | 0.85 | 0.98 |
| Qwen3-32B-no-thinking | 0.77 | 0.97 | 0.84 | 0.97 | 0.66 | 0.87 | 0.76 | 0.94 | 0.51 | 0.84 | 0.59 | 0.89 | 0.19 | 0.51 | 0.43 | 0.75 |

**Normalized Analysis.** To address the imbalance in task difficulty across reasoning taxonomies, we follow prior work (Yuan et al., 2024; Yu et al., 2023) and apply a normalization procedure to reduce the influence of task complexity on model performance. Specifically, for each model and task, we compute a $z$-score of Pass@$k$ and Cons@$k$ accuracy and then linearly rescale the scores to the range $[0, 100]$. Specifically, for each model $i$ and each task $j$, we compute the $z$-score of its Pass@$k$ and Cons@$k$ accuracy as follows:

$$z_{ij} = \frac{x_{ij} - \mu\big(x_{i1}, \ldots, x_{i|M|}\big)}{\sigma\big(x_{i1}, \ldots, x_{i|M|}\big)}, \tag{5}$$

where $x_{ij}$ denotes the Pass@$k$ or Cons@$k$ accuracy of model $i$ on task $j$, while $\mu(\cdot)$ and $\sigma(\cdot)$ denote the mean and standard deviation over all $M$ models on that task.

We then linearly rescale the $z$-scores into the range $[0, 100]$ to obtain a normalized score:

$$s_{ij} = 100 \cdot \frac{z_{ij} - \min(z)}{\max(z) - \min(z)}, \tag{6}$$

where $\min(z)$ and $\max(z)$ are taken over all models on the same task.

Figure 10-Figure 13 present the normalized scores of different models across reasoning taxonomies and graph sizes. The normalized outcomes preserve the same performance hierarchy observed earlier, highlighting persistent weaknesses of LRMs in intuitive reasoning.

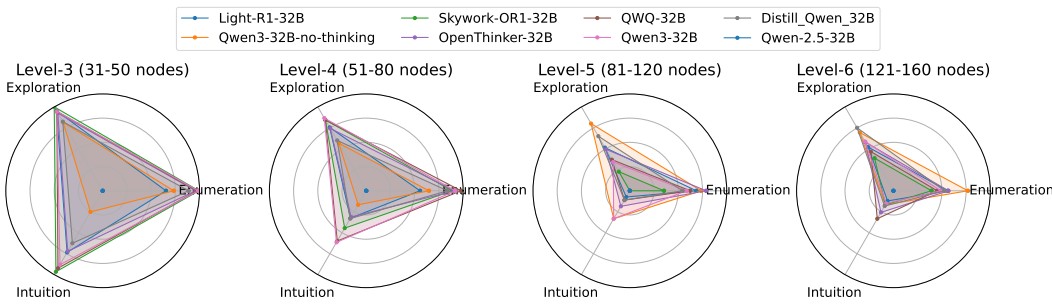

Figure 10: Normalized pass@k scores of different reasoning taxonomies (32B models).

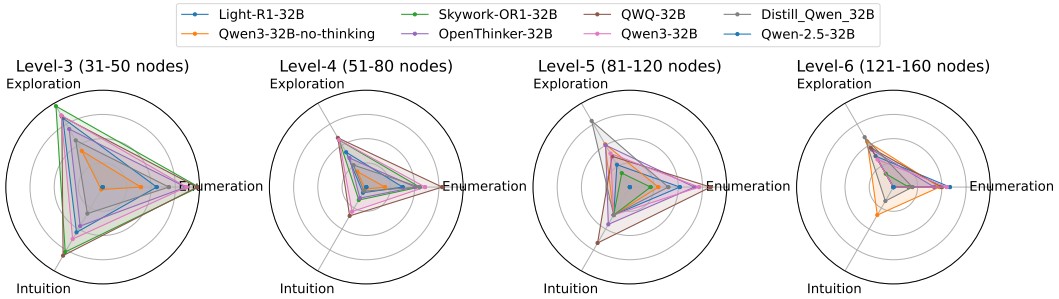

Figure 11: Normalized cons@k z-scores of different reasoning taxonomies (32B models).

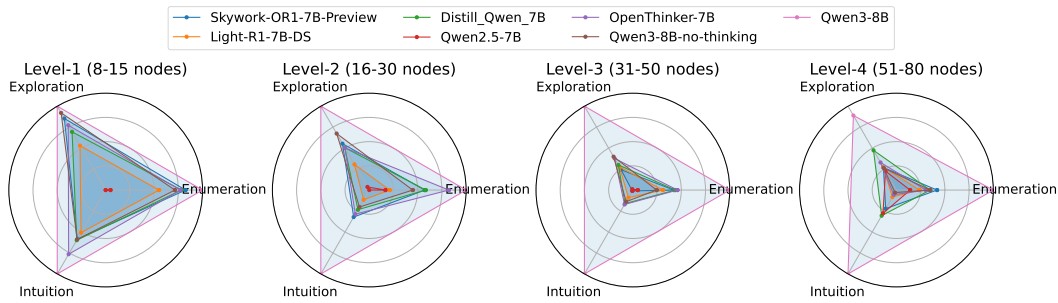

Figure 12: Normalized pass@k z-scores of different reasoning taxonomies (8B models).

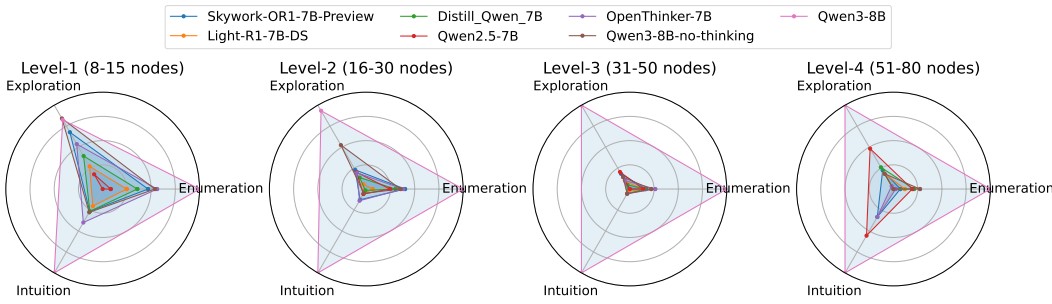

Figure 13: Normalized cons@k Z-scores of different reasoning taxonomies (8B models).

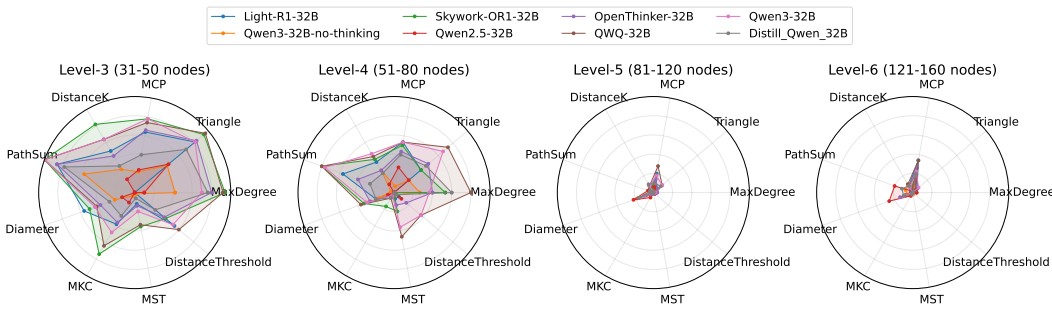

Figure 14: Cons@k accuracy of different reasoning taxonomies (32B models).

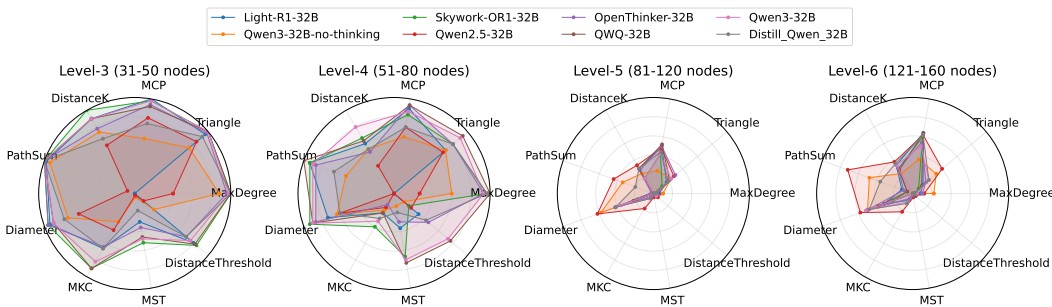

Figure 15: Pass@k accuracy of different reasoning taxonomies (32B models).

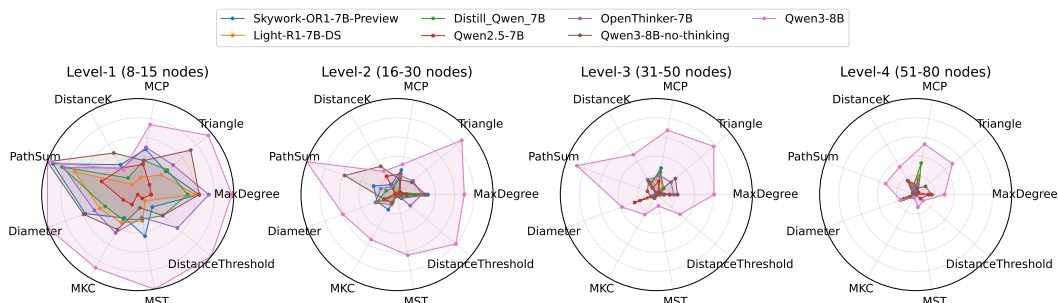

Figure 16: Cons@k accuracy of different reasoning taxonomies (8B models).

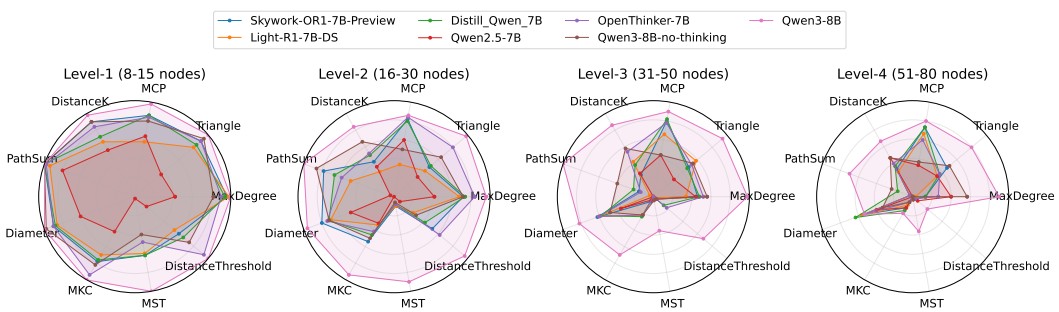

Figure 17: Pass@k accuracy of different reasoning taxonomies (8B models).

## H.3 ERROR ANALYSIS (RQ1)

Figure 18: Error type distributions across reasoning taxonomies for Qwen3-8B *(top)* and its non-reasoning variant Qwen3-8B-no-thinking *(bottom)*.

## H.4 HIGH ENTROPY TOKENS (RQ2)

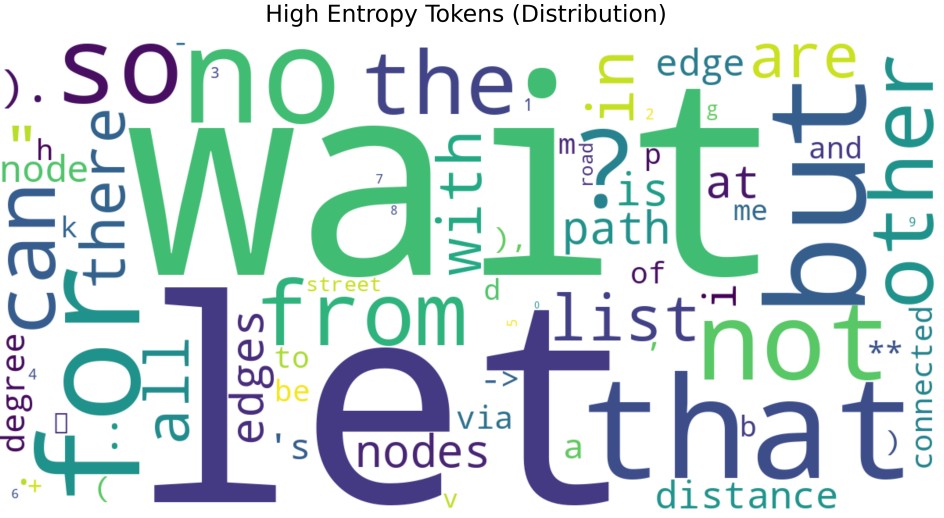

Figure 19: High-entropy tokens in LRMs inference traces.

## H.5 OVERTHINKING ANALYSIS (RQ2)

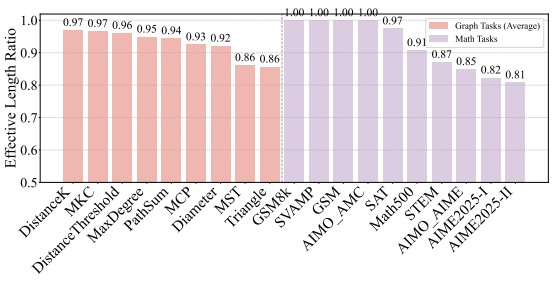
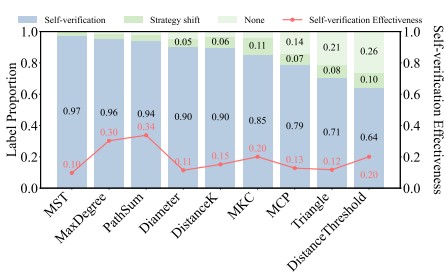

(a) Outcome efficiency

(b) Proportion of reasoning behaviors

Figure 20: Overthinking analysis of Distill-Qwen-32B.

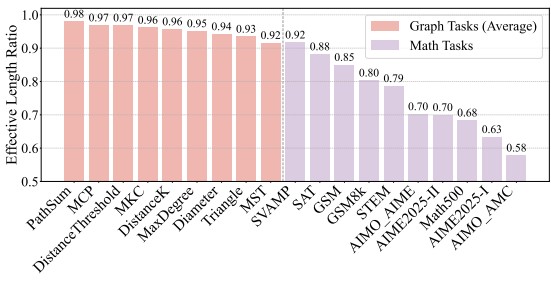
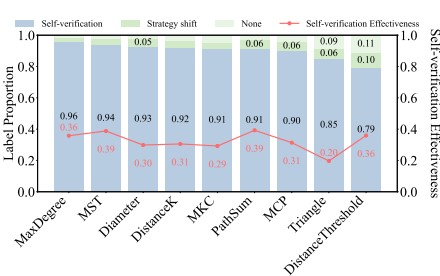

(a) Outcome efficiency

(b) Proportion of reasoning behaviors

Figure 21: Overthinking analysis of Qwen3-32B.

# I DISCUSSION

Table 6: Performance (pass@k / cons@k (%)) of Qwen3-32B under different graph representations.

| Level | Task | Edge list | Adjacency list | Markdown table | Numeric IDs |
|---|---|---|---|---|---|
| Level-1 | DistanceK | 100.0 / 100.0 | 100.0 / 100.0 | 100.0 / 100.0 | 100.0 / 100.0 |
| (8–15 nodes) | MST | 100.0 / 100.0 | 100.0 / 100.0 | 100.0 / 100.0 | 100.0 / 100.0 |
| Level-2 | DistanceK | 100.0 / 83.3 | 100.0 / 96.7 | 96.7 / 86.7 | 100.0 / 90.0 |
| (16–30 nodes) | MST | 96.7 / 83.3 | 100.0 / 86.7 | 93.3 / 83.3 | 96.7 / 83.3 |
| Level-3 | DistanceK | 93.3 / 66.7 | 100.0 / 100.0 | 86.7 / 76.7 | 96.7 / 83.3 |
| (31–50 nodes) | MST | 23.3 / 13.3 | 23.3 / 3.3 | 53.3 / 13.3 | 53.3 / 26.7 |

# J OTHER EXPERIMENTS

Table 7: Level-3 (31–50 nodes) results. Each cell reports pass@k / cons@k.

| Model | Easy | | | Medium | | | Hard | | |
|---|---|---|---|---|---|---|---|---|---|
| | Max Degree | Path Sum | MKC | Triangle | Distance-K | MST | MCP | Diameter | Distance Threshold |
| Distill_Qwen_32B | 1.00 / 0.80 | 1.00 / 0.74 | 0.60 / 0.28 | 0.90 / 0.66 | 0.66 / 0.34 | 0.20 / 0.00 | 0.72 / 0.36 | 0.72 / 0.32 | 0.24 / 0.24 |
| Light-R1-32B | 0.90 / 0.84 | 0.96 / 0.90 | 0.62 / 0.34 | 0.94 / 0.80 | 0.84 / 0.50 | 0.33 / 0.07 | 1.00 / 0.66 | 0.92 / 0.56 | 0.56 / 0.50 |
| Llama-3.3-70B | 0.78 / 0.32 | 0.56 / 0.28 | 0.06 / 0.02 | 0.12 / 0.00 | 0.34 / 0.04 | 0.00 / 0.00 | 0.08 / 0.00 | 0.20 / 0.02 | 0.00 / 0.00 |
| OpenThinker-32B | 0.96 / 0.78 | 0.98 / 0.80 | 0.62 / 0.36 | 0.98 / 0.80 | 0.76 / 0.44 | 0.30 / 0.10 | 0.94 / 0.68 | 0.90 / 0.40 | 0.40 / 0.30 |
| QWQ-32B | 0.98 / 0.96 | 1.00 / 0.98 | 0.90 / 0.64 | 1.00 / 0.94 | 0.88 / 0.64 | 0.70 / 0.50 | 0.92 / 0.76 | 0.90 / 0.48 | 0.80 / 0.60 |
| Qwen2.5-32B | 0.40 / 0.12 | 0.08 / 0.00 | 0.40 / 0.12 | 0.82 / 0.44 | 0.58 / 0.18 | 0.00 / 0.00 | 0.80 / 0.26 | 0.58 / 0.18 | 0.12 / 0.00 |
| Qwen3-32B | 0.98 / 0.76 | 1.00 / 0.98 | 0.78 / 0.46 | 1.00 / 0.80 | 0.88 / 0.66 | 0.63 / 0.30 | 0.98 / 0.76 | 0.90 / 0.48 | 0.64 / 0.56 |
| Qwen3-32B-no-thinking | 0.84 / 0.36 | 0.94 / 0.58 | 0.30 / 0.02 | 0.68 / 0.44 | 0.72 / 0.32 | 0.13 / 0.03 | 0.54 / 0.20 | 0.66 / 0.20 | 0.24 / 0.02 |
| Skywork-OR1-32B | 1.00 / 0.94 | 1.00 / 1.00 | 0.90 / 0.76 | 1.00 / 0.92 | 1.00 / 0.82 | 0.67 / 0.27 | 0.98 / 0.82 | 0.94 / 0.52 | 0.54 / 0.38 |
| gpt_oss_20b | 0.98 / 0.90 | 1.00 / 1.00 | 0.82 / 0.60 | 0.98 / 0.88 | 0.94 / 0.76 | 0.73 / 0.50 | 0.96 / 0.78 | 0.88 / 0.44 | 0.42 / 0.22 |

Table 8: Level-4 (51–80 nodes) results. Each cell reports pass@k / cons@k.

| Model | Easy | | | Medium | | | Hard | | |
|---|---|---|---|---|---|---|---|---|---|
| | Max Degree | Path Sum | MKC | Triangle | Distance-K | MST | MCP | Diameter | Distance Threshold |
| Distill_Qwen_32B | 0.93 / 0.57 | 0.67 / 0.30 | 0.27 / 0.07 | 0.80 / 0.43 | 0.40 / 0.20 | 0.16 / 0.06 | 0.70 / 0.33 | 0.60 / 0.20 | 0.03 / 0.03 |
| Light-R1-32B | 0.60 / 0.50 | 0.90 / 0.53 | 0.23 / 0.03 | 0.70 / 0.37 | 0.57 / 0.33 | 0.30 / 0.12 | 0.87 / 0.50 | 0.67 / 0.23 | 0.07 / 0.07 |
| Llama-3.3-70B | 0.47 / 0.13 | 0.10 / 0.00 | 0.03 / 0.00 | 0.13 / 0.00 | 0.17 / 0.00 | 0.00 / 0.00 | 0.10 / 0.00 | 0.23 / 0.00 | 0.00 / 0.00 |
| OpenThinker-32B | 0.90 / 0.37 | 0.80 / 0.37 | 0.13 / 0.03 | 0.80 / 0.50 | 0.50 / 0.23 | 0.36 / 0.14 | 0.80 / 0.40 | 0.87 / 0.27 | 0.13 / 0.13 |
| QWQ-32B | 1.00 / 0.73 | 1.00 / 0.90 | 0.23 / 0.10 | 0.90 / 0.70 | 0.63 / 0.43 | 0.46 / 0.36 | 0.90 / 0.57 | 0.87 / 0.37 | 0.70 / 0.33 |
| Qwen2.5-32B | 0.27 / 0.00 | 0.00 / 0.00 | 0.17 / 0.03 | 0.57 / 0.20 | 0.30 / 0.07 | 0.02 / 0.00 | 0.70 / 0.20 | 0.57 / 0.07 | 0.20 / 0.10 |
| Qwen3-32B | 0.93 / 0.40 | 0.90 / 0.77 | 0.30 / 0.03 | 0.87 / 0.63 | 0.77 / 0.47 | 0.48 / 0.14 | 0.83 / 0.53 | 0.90 / 0.23 | 0.50 / 0.33 |
| Qwen3-32B-no-thinking | 0.60 / 0.23 | 0.50 / 0.07 | 0.13 / 0.00 | 0.60 / 0.17 | 0.53 / 0.27 | 0.04 / 0.00 | 0.53 / 0.03 | 0.60 / 0.23 | 0.13 / 0.03 |
| Skywork-OR1-32B | 0.93 / 0.53 | 0.93 / 0.83 | 0.37 / 0.17 | 0.80 / 0.37 | 0.67 / 0.37 | 0.52 / 0.40 | 0.80 / 0.50 | 0.93 / 0.30 | 0.00 / 0.00 |
| gpt_oss_20b | 0.97 / 0.83 | 0.80 / 0.67 | 0.37 / 0.17 | 0.70 / 0.50 | 0.57 / 0.47 | 0.34 / 0.20 | 0.67 / 0.43 | 0.63 / 0.37 | 0.63 / 0.50 |

**Findings of group problems by computational complexity** We re-evaluate model performance by grouping tasks based on theoretical time complexity—Easy ($O(n)$), Medium ($O(n^2)$), and Hard ($O(n^3)$/NP-hard)—while controlling for graph size (using Level-3 and Level-4 datasets) to isolate the impact of algorithmic difficulty. Results are shown in Table 7 and 8. This analysis yields three critical insights. First, within the same reasoning taxonomy, performance generally declines as computational complexity increases; for instance, models like QWQ-32B achieve near-perfect scores on linear-time tasks (e.g., *Max Degree*) but degrade significantly on NP-hard tasks (e.g., *Distance Threshold*), confirming that expanded search spaces pose greater challenges to LRMs. Second, we observe an anomaly in the *Exploration* category where models frequently outperform on the "Hard" *Diameter* task compared to the "Medium" *Distance-K* task (e.g., QWQ-32B scores 0.87 vs. 0.63 on Level-4), a discrepancy we attribute to training data distributions favoring specific classic problems. Third, reasoning taxonomy often outweighs theoretical complexity: models consistently perform better on "Hard" *Enumeration* tasks than "Easy" *Intuition* tasks (e.g., Light-R1-32B achieves 1.00 on the NP-hard *MCP* but only 0.62 on the linear-time *MKC*). This suggests that current LRMs are inherently optimized for systematic, step-by-step verification rather than the global heuristic judgments required for intuition-based problems, even when the theoretical complexity suggests the former should be more difficult.

# K  DETAILS DEMONSTRATION

## K.1  PROMPTS

---

**Prompts K.1: LLM Prompts For Categorizing Enumeration Algorithm**

SYSTEM PROMPT

You are a graph theory expert. Given a graph problem and an answer to that problem, please determine which of the following approaches is used in the answer:

   a)  Enumeration (such as brute-force methods)

   b)  Search (such as BFS, DFS)

   c)  Greedy (such as Dijkstra's algorithm)

   d)  Not mentioned (other algorithmic approaches)

Directly output your choice (a, b, c, or d) with no explanation or additional text.

USER PROMPT

Given the problem and its answer below, identify which of the following approaches was used to derive the solution:
Problem: {question}
Answer: {answer}

---

**Prompts K.2: LLM Prompts For Categorizing Exploration Algorithm**

SYSTEM PROMPT

You are a graph theory expert. Given a problem related to finding the shortest path and its corresponding answer, please determine which of the following approaches is used in the answer:

   a)  Enumeration (such as brute-force methods)

   b)  Search (such as BFS, DFS)

   c)  Greedy (such as Dijkstra's algorithm)

   d)  Not mentioned (other algorithmic approaches)

Directly output your choice (a, b, c or d) with no explanation or additional text.

USER PROMPT

Given the problem and its answer below, identify which of the following approaches was used to derive the solution:
Problem: {question}
Answer: {answer}

---

**Prompts K.3: LLM Prompts For Categorizing Intuition Algorithm**

SYSTEM PROMPT

You are a graph theory expert. Given a Maximum K-core (MKC) problem and an answer to that problem, please determine which of the following approaches is used in the answer:

   a)  Enumeration (such as brute-force methods)

   b)  Search (such as BFS, DFS)

   c)  Greedy (such as Peeling algorithm)

   d)  Not mentioned (other algorithmic approaches)

Directly output your choice (a, b, c or d) with no explanation or additional text.

USER PROMPT

Given the problem and its answer below, identify which of the following approaches was used to derive the solution:

---

Problem: {question}
Answer: {answer}

## Prompts K.4: LLM Prompts for Categorizing Errors In AI-Generated Solutions

You are an intelligent AI assistant.Given a graph problem and an LLM's response to that problem, analyze the LLM's response to identify any errors it contains. Use the following refined error categories and definitions for your analysis:

OUTPUT QUALITY

- **redundancy:** Unnecessary repetition or superfluous information in the solution.

ALGORITHM SELECTION

- **incorrect algorithm:** The chosen algorithm cannot produce the correct solution for the problem. In this task, the correct algorithms are: {correct_algorithm_dict[args.task]}.
- **suboptimal algorithm:** The algorithm used is inefficient or is implemented in a less efficient way. In this task, the efficient algorithms are: {efficient_algorithm_dict[args.task]}.

INFORMATION ERRORS

- **graph memorization error:** Misunderstanding or incorrect memory of the graph structure, such as extra or missing nodes/edges.

ALGORITHM EXECUTION

- **state update error:** During algorithm execution, state variables or data structures are updated incorrectly, causing subsequent steps to operate on erroneous states and compromising overall correctness.
- **omission:** Missing important elements during execution, e.g., failure to traverse all nodes or edges.
- **condition misjudgment:** Mistakes in condition checks, such as incorrect if-statement or loop-condition evaluations.

**Instructions:** The LLM's response will be segmented into sections labeled as "[Section 1], content...", "[Section 2], content...", etc.
Your response format should be a list of error annotations as:
"[section index, error type, detailed error analysis]"
If multiple errors exist in one section, list them separately.
Do not output anything other than these error annotations.

## Prompts K.5: LLM Prompts For Reformatting The Output

Here is the problem and the response:
You are an intelligent assistant. Given a graph problem and its response, your task is to review the response, which is divided into multiple parts with each step labeled using tags. After reading through the steps, you should group them into distinct sections, where each section represents a complete and logical problem-solving attempt or process.
Specific instructions:

1. Each section should be a standalone and complete problem-solving approach or effort.

2. For each section, include both the starting and ending tags (the ending tag should not be earlier than the starting tag). Additionally, provide a brief summary or title of the section.

3. Present your output in this format: <<start tag>> - <<end tag>> [Brief description]. The end tag should be the same as or later than the start tag of the section, and the start tag of the next section should follow directly after the end tag of the previous section.\n\n Do not output any other text or explanation.

Problem: {`question`}
Response: {`answer`}
Present your output in this format: `<<start tag>> - <<end tag>>` [Brief description]. The end tag should be the same as or later than the start tag of the section, and the start tag of the next section should follow directly after the end tag of the previous section. Do not output any other text or explanation.

## Prompts K.6: LLM Prompts For Detecting The First Correct Answer

You are a precise analysis assistant for graph theory problems. Your task is to strictly judge whether the provided reasoning segment contains an explicitly stated final answer that matches the given final answer. Given the following graph problem, a segment of reasoning process, and the final answer that was ultimately provided, please determine whether this segment already contains the final answer.

The final answer is: {`final_answer`}
Problem: {`question`}
Reasoning segment: {`segment`}

Only output YES if the final answer is explicitly presented and clearly identified in this segment (e.g., by phrases like "the answer is," "thus," "therefore," "in conclusion," etc.), not just mentioned as part of intermediate reasoning. If the final answer is not explicitly stated as a conclusion in this segment, output NO. Do not provide any explanation.

## Prompts K.7: LLM Prompts For Categorizing Response Segments

You are an expert in analyzing text relationships and functions. Your task is to strictly classify the function of Segment B in relation to Segment A, according to the provided definitions. Always output only the corresponding option without any additional explanation or commentary.

Given two text segments, A and B, determine the function of B in relation to A. There are three possible options:
1.Self-reflection: B serves to evaluate or verify the correctness of A.
2.Strategy shift: B serves to alter or adjust the strategy presented in A.
3.None: B has no clear functional relationship to A, or serves a different purpose not covered by the above categories.

Output only one of the following options without any explanation: Self-reflection, Strategy shift, or None.

Segment A: {`segment_a`}
Segment B: {`segment_b`}

## Prompts K.8: LLM Prompts For Identifying The Conclusive Answer Segment

Given two text segments, A and B, where B is labeled as "{label}", determine if B is an effective addition to A.
For a segment to be effective:

- If labeled as "Self-reflection": B should effectively evaluate or verify the correctness of A
- If labeled as "Strategy shift": B should effectively alter or adjust the strategy presented in A
- If labeled as "None": B has no clear functional relationship to A, or serves a different purpose not covered by the above categories.

Please analyze the relationship between A and B and determine:

1. Is B an effective addition to A given its label? (Yes/No)
2. What is your confidence in this judgment? (High/Medium/Low)

Output your judgment in the following format:
Effectiveness: [Yes/No] Confidence: [High/Medium/Low]
Segment A: {segment_a}
Segment B: {segment_b}

## K.2 RESPONSE REFORMATION

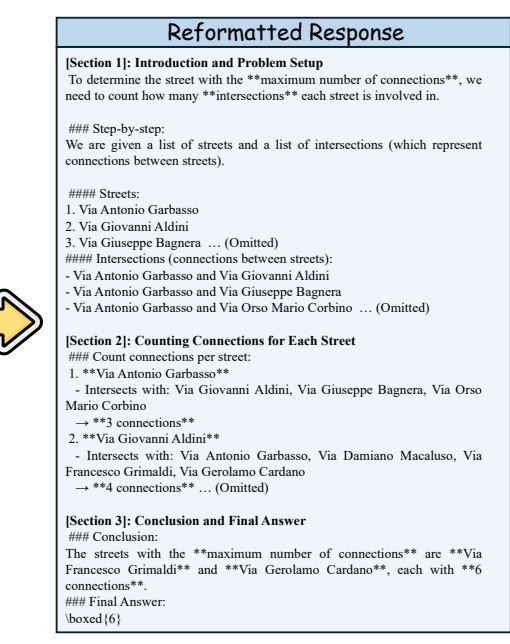

**Original Response**

To determine the street with the **maximum number of connections**, we need to count how many **intersections** each street is involved in.
### Step-by-step:
We are given a list of streets and a list of intersections (which represent connections between streets).
#### Streets:
1. Via Antonio Garbasso
2. Via Giovanni Aldini
3. Via Giuseppe Bagnera … (Omitted)
#### Intersections (connections between streets):
- Via Antonio Garbasso and Via Giovanni Aldini
- Via Antonio Garbasso and Via Giuseppe Bagnera
- Via Antonio Garbasso and Via Orso Mario Corbino … (Omitted)
### Count connections per street:
1. **Via Antonio Garbasso**
  - Intersects with: Via Giovanni Aldini, Via Giuseppe Bagnera, Via Orso Mario Corbino
  → **3 connections**
2. **Via Giovanni Aldini**
  - Intersects with: Via Antonio Garbasso, Via Damiano Macaluso, Via Francesco Grimaldi, Via Gerolamo Cardano
  → **4 connections**
3. **Via Giuseppe Bagnera**
  - Intersects with: Via Antonio Garbasso, Via Francesco Grimaldi, Via Gerolamo Cardano
  → **3 connections** … (Omitted)

The streets with the **maximum number of connections** are **Via Francesco Grimaldi** and **Via Gerolamo Cardano**, each with **6 connections**.
### Final Answer:
\boxed{6}

**Reformatted Response**

[Section 1]: Introduction and Problem Setup
 To determine the street with the **maximum number of connections**, we need to count how many **intersections** each street is involved in.

 ### Step-by-step:
We are given a list of streets and a list of intersections (which represent connections between streets).

 #### Streets:
1. Via Antonio Garbasso
2. Via Giovanni Aldini
3. Via Giuseppe Bagnera … (Omitted)
#### Intersections (connections between streets):
- Via Antonio Garbasso and Via Giovanni Aldini
- Via Antonio Garbasso and Via Giuseppe Bagnera
- Via Antonio Garbasso and Via Orso Mario Corbino … (Omitted)

[Section 2]: Counting Connections for Each Street
 ### Count connections per street:
 1. **Via Antonio Garbasso**
  - Intersects with: Via Giovanni Aldini, Via Giuseppe Bagnera, Via Orso Mario Corbino
  → **3 connections**
 2. **Via Giovanni Aldini**
  - Intersects with: Via Antonio Garbasso, Via Damiano Macaluso, Via Francesco Grimaldi, Via Gerolamo Cardano
  → **4 connections** … (Omitted)

[Section 3]: Conclusion and Final Answer
 ### Conclusion:
The streets with the **maximum number of connections** are **Via Francesco Grimaldi** and **Via Gerolamo Cardano**, each with **6 connections**.
### Final Answer:
\boxed{6}

Figure 22: Example of Response Reformation.

### K.3 CASE STUDIES

---

**Case Study K.1: Response of Qwen3-32B on a DistanceK Task**

**Problem:**
You are given an undirected graph representing the London street network, where nodes represent streets and edges represent intersections. The distance between two directly connected nodes is 1. Given a street Westwell Road Approach, find all streets that are exactly distance 4 away from the street.

Streets in the network: Cleveland Road, Colmer Road, Danbrook Road, Streatham High Road, Penistone Road, Arragon Gardens, Ellison Road, Tankerville Road, Westwell Road, Westwell Road Approach, Buckleigh Road, Greyhound Lane, Guildersfield Road, Kempshott Road, Northanger Road Intersections in the network: Cleveland Road and Ellison Road, Colmer Road and Danbrook Road, Colmer Road and Ellison Road, Colmer Road and Streatham High Road, Danbrook Road and Ellison Road, Streatham High Road and Arragon Gardens, Streatham High Road and Greyhound Lane, Streatham High Road and Guildersfield Road, Streatham High Road and Kempshott Road, Streatham High Road and Penistone Road, Streatham High Road and Westwell Road Approach, Penistone Road and Guildersfield Road, Arragon Gardens and Ellison Road, Ellison Road and Greyhound Lane, Ellison Road and Guildersfield Road, Ellison Road and Kempshott Road, Tankerville Road and Greyhound Lane, Tankerville Road and Guildersfield Road, Tankerville Road and Kempshott Road, Westwell Road and Greyhound Lane, Westwell Road and Northanger Road, Westwell Road and Westwell Road Approach, Buckleigh Road and Greyhound Lane, Buckleigh Road and Guildersfield Road, Buckleigh Road and Kempshott Road, Buckleigh Road and Northanger Road, Greyhound Lane and Northanger Road Please find the streets in distance 4 from the street Westwell Road Approach and output the number of these streets. Present your answer in the following format: `\boxed{n}`. n is the number of these streets

**Response:**
[Section 1]: Initial Understanding and Setup
The task is to find all streets exactly 4 units of distance from Westwell Road Approach in the given undirected graph, where each intersection represents an edge of distance 1. An adjacency list is a helpful way to represent the graph structure. Westwell Road Approach is the starting point (level 0). Based on the intersection list, Westwell Road Approach is directly connected to Streatham High Road and Westwell Road, making them the streets at distance 1 (level 1).

[Section 2]: BFS Traversal to Identify Level 3 Nodes
To find the nodes at distance 2, we find the neighbors of the level 1 nodes. The neighbors of Streatham High Road are Colmer Road, Penistone Road, Guildersfield Road, Greyhound Lane, Kempshott Road, and Arragon Gardens. The neighbors of Westwell Road are Greyhound Lane and Northanger Road. The unique set of these neighbors forms level 2: Colmer Road, Penistone Road, Guildersfield Road, Greyhound Lane, Kempshott Road, Arragon Gardens, and Northanger Road. To find level 3, we find the neighbors of all level 2 nodes that have not been visited. This process yields a unique set of new nodes for level 3: Danbrook Road, Ellison Road, Tankerville Road, and Buckleigh Road.

[Section 3]: Re-examination of Level 3 and 4 Nodes
Now, to find level 4, we must find the neighbors of the level 3 nodes (Danbrook Road, Ellison Road, Tankerville Road, Buckleigh Road) that have not already been assigned a level. The neighbors of Danbrook Road are Colmer Road (level 2) and Ellison Road (level 3), so no new nodes are found. The neighbors of Tankerville Road are Greyhound Lane, Guildersfield Road, and Kempshott Road, all of which are in level 2. The neighbors of Buckleigh Road are Greyhound Lane, Guildersfield Road, Kempshott Road, and Northanger Road, all of which are in level 2. For Ellison Road, we must check its intersections. Ellison Road is connected to: Colmer Road and Ellison Road, Danbrook Road and Ellison Road, Arragon Gardens and Ellison Road, Ellison Road and Greyhound Lane, Ellison Road and Guildersfield Road, Ellison Road and Kempshott Road. (This list of Ellison Road's neighbors is incomplete; it omits the intersection with Cleveland Road, which is explicitly provided in the problem description. – graph memorization error)
All of the listed neighbors for Ellison Road are in either level 2 or level 3. So no new nodes here. So indeed, no new nodes. That would mean that level 4 is empty. (This conclusion is incorrect because the algorithm failed to process the edge between Ellison

---

Road (level 3) and Cleveland Road. This failure to traverse a valid edge results in Cleveland Road being missed. – omission) [Section 4]: Final Verification and Conclusion
The analysis incorrectly concludes that since all neighbors of level 3 nodes are already in levels 2 or 3, there are no nodes at distance 4. The response attempts to manually trace paths but fails to find a path of length 4 due to the previously forgotten graph connection. Therefore, there are no nodes at distance 4. So the answer is 0.
(This final judgment is incorrect. The BFS algorithm was executed on an incomplete graph, leading to the false conclusion that level 4 is empty when it actually contains one node, Cleveland Road. – condition misjudgment)

---

**Case Study K.2: Response of Qwen3-32B on a Minimum Spanning Tree Task**

**Problem:**
You are required to solve the Minimum Spanning Tree Problem for an undirected street network. In this network, nodes represent streets (e.g., street IDs) and edges represent intersections between streets. The weight of each edge is the distance between two streets.
Streets in the network: Carbeen Road, Eucalyptus Drive, Erawar Close, Nicholson Avenue, Howes Close, Fairy Dell Close, Elouera Road, Booleroo Place, Higgins Place, Keys Close, Russell Crescent, Coppsleigh Close, Gundy Place, Boree Place, Brigalow Place, Baroona Road, Eastview Road, Mason Lane, Phillips Lane, Ulric Lane, Coral Heath Avenue, Brushtail Court, Hibbertia Place, Corang Road, Honeycup Close, Rocklily Avenue, Billarga Road, Pittwater Road, Apanie Place, Duneba Drive, Silver Crescent, Colin Place, Kimba Close, Western Crescent, Old Glenfield Road, Warrigal Drive, Coora Road, Namoi Road, Nulgarra Street, Strathallen Avenue, Settlers Way, Euroka Road, Bottle Brush Road, Gum Blossom Drive, Barkala Place, Quarter Sessions Road, Dryden Avenue, De Saxe Close, Lynrob Place, Timbarra Road Intersections between these streets: Carbeen Road and Duneba Drive (weight: 3), Carbeen Road and Elouera Road (weight: 3), Eucalyptus Drive and Billarga Road (weight: 3), Eucalyptus Drive and Boree Place (weight: 9), Eucalyptus Drive and Corang Road (weight: 3), Eucalyptus Drive and Elouera Road (weight: 9), Erawar Close and Quarter Sessions Road (weight: 4), Nicholson Avenue and Quarter Sessions Road (weight: 6), Howes Close and Quarter Sessions Road (weight: 1), Fairy Dell Close and Quarter Sessions Road (weight: 5), Elouera Road and Duneba Drive (weight: 5), Booleroo Place and Corang Road (weight: 6), Higgins Place and Keys Close (weight: 3), Higgins Place and Quarter Sessions Road (weight: 4), Higgins Place and Russell Crescent (weight: 9), Russell Crescent and Quarter Sessions Road (weight: 6), Coppsleigh Close and Corang Road (weight: 2), Gundy Place and Duneba Drive (weight: 10), Boree Place and Duneba Drive (weight: 1), Brigalow Place and Duneba Drive (weight: 1), Baroona Road and Eastview Road (weight: 2), Baroona Road and Mason Lane (weight: 2), Baroona Road and Namoi Road (weight: 8), Baroona Road and Nulgarra Street (weight: 4), Baroona Road and Phillips Lane (weight: 7), Baroona Road and Pittwater Road (weight: 9), Baroona Road and Quarter Sessions Road (weight: 9), Baroona Road and Strathallen Avenue (weight: 5), Baroona Road and Ulric Lane (weight: 1), Eastview Road and Pittwater Road (weight: 3), Eastview Road and Quarter Sessions Road (weight: 2), Mason Lane and Namoi Road (weight: 2), Coral Heath Avenue and Hibbertia Place (weight: 6), Coral Heath Avenue and Honeycup Close (weight: 7), Coral Heath Avenue and Quarter Sessions Road (weight: 1), Coral Heath Avenue and Rocklily Avenue (weight: 1), Brushtail Court and Old Glenfield Road (weight: 5), Brushtail Court and Quarter Sessions Road (weight: 3), Corang Road and Billarga Road (weight: 6), Corang Road and Kimba Close (weight: 5), Corang Road and Quarter Sessions Road (weight: 4), Pittwater Road and Quarter Sessions Road (weight: 6), Apanie Place and Duneba Drive (weight: 10), Duneba Drive and Euroka Road (weight: 6), Duneba Drive and Quarter Sessions Road (weight: 7), Silver Crescent and Coora Road (weight: 6), Silver Crescent and Quarter Sessions Road (weight: 1), Colin Place and Dryden Avenue (weight: 9), Colin Place and Quarter Sessions Road (weight: 10), Western Crescent and Quarter Sessions Road (weight: 5), Old Glenfield Road and Quarter Sessions Road (weight: 6), Warrigal Drive and Quarter Sessions Road (weight: 7), Coora Road and Barkala Place (weight: 9), Settlers Way and Quarter Sessions Road (weight: 10), Bottle Brush Road and Gum Blossom Drive (weight: 7), Bottle Brush Road and Quarter Sessions Road (weight: 4), Gum Blossom Drive and Quarter Sessions Road (weight: 10), Barkala Place and Quarter Sessions Road (weight: 7), Quarter Sessions Road and De Saxe Close (weight: 5), Quarter Sessions Road and Lynrob Place (weight: 5), Quarter Sessions Road and Timbarra Road

(weight: 6), De Saxe Close and Lynrob Place (weight: 1), De Saxe Close and Timbarra Road (weight: 4). Identify the minimum spanning tree of this network. The minimum spanning tree is a subset of edges in a connected, weighted graph that connects all the vertices together with the smallest possible total edge weight and without any cycles. Present your answer in the following format: \boxed{n}, where n is the sum of the weights of the edges in the minimum spanning tree.

**Response:**
[Section 1]: Initial Thoughts and Edge Listing
To solve this Minimum Spanning Tree (MST) problem, I will use Kruskal's algorithm, which involves sorting all edges by weight and adding them to the tree if they do not form a cycle. I will begin by listing and sorting the edges based on their weights. The edges with a weight of 1 are: Boree Place - Duneba Drive, Brigalow Place - Duneba Drive, Howes Close - Quarter Sessions Road, Baroona Road - Ulric Lane, Coral Heath Avenue - Rocklily Avenue, Coral Heath Avenue - Quarter Sessions Road, Silver Crescent - Quarter Sessions Road, and De Saxe Close - Lynrob Place. The edges with a weight of 2 are: Coppsleigh Close - Corang Road, Baroona Road - Eastview Road, Baroona Road - Mason Lane, Eastview Road - Quarter Sessions Road, and Mason Lane - Namoi Road.
[Section 2]: First Attempt at MST Calculation
I will now apply Kruskal's algorithm. First, I select all edges with weight 1, which can all be added without creating cycles. Next, I process the five edges with weight 2, all of which are safe to add. The total weight so far is 16. I then add all six edges with weight 3, bringing the total weight to 34. Next, I process edges with weight 4. So total weight 4 edges: 4 edges? (This count is incorrect. The algorithm fails to include all edges of weight 4 from the problem description, specifically missing Higgins Place - QSR and De Saxe Close - Timbarra Road. – graph memorization error) I add the four identified edges, bringing the total weight to 50. I then process the edges with weight 5. So all six weight 5 edges can be added. (This count is also incorrect. The algorithm misses the edge Fairy Dell Close - QSR (weight 5). – omission) This brings the total weight to 80.
Next, I process the edges with weight 6. So seven edges with weight 6. (This count is incorrect. The algorithm fails to identify all edges with a weight of 6, missing several connections to Quarter Sessions Road and others. – graph memorization error) I correctly determine that the edge Corang Road - Billarga Road should be skipped as it forms a cycle. However, I incorrectly determine that the edge Old Glenfield Road - QSR also creates a cycle based on a faulty state. So from the seven weight 6 edges, we can add Russell Crescent - QSR, Pittwater Road - QSR, Duneba Drive - Euroka Road, Silver Crescent - Coora Road, Warrigal Drive - QSR. (This step contains multiple errors. The list of weight 6 edges is incomplete, and the cycle detection for Old Glenfield Road - QSR is based on a flawed understanding of the current graph state. – state update error) This adds 30 to the total, making it 110. The process continues in this manner, with accumulating errors.
[Section 3]: Identifying and Correcting Mistakes in Weight 6 Edges
I now recognize that my previous analysis was flawed. I missed the edge Fairy Dell Close - QSR (weight 5). So during the weight 5 step, I should have added seven edges instead of six. (This section demonstrates the algorithm's confusion and repeated errors. While it correctly identifies one missed edge, its subsequent recalculations remain flawed, leading to more incorrect states. This repetitive and confusing correction process is a form of redundancy. – redundancy) I then re-evaluate the weight 6 edges and realize I also missed Booleroo Place - Corang Road, Quarter Sessions Road - Timbarra Road, and Coral Heath Avenue - Hibbertia Place. Ah, here's another mistake! (The repeated self-correction and recalculation indicate a persistent failure to maintain a correct state of the graph's connected components. – state update error) After several rounds of corrections, I still have an incomplete and inaccurate picture of the graph.
[Section 4]: Final MST Calculation and Verification
After realizing multiple oversights, I will restart the calculation. Weight 1 (8 edges) + Weight 2 (5 edges) + Weight 3 (6 edges) = 34. Next are the weight 4 edges. Weight 4 edges: 5*4=20 – total 54. (This is incorrect. There are six edges with weight 4 in the graph. The algorithm omits the edge De Saxe Close - Timbarra Road (weight 4). – graph memorization error) Next are the weight 5 edges. There are seven, which are all added, for a total of 35, bringing the sum to 89. Next are the weight 6 edges. Weight 6 edges: 9*6=54 – total 143. (This step contains two errors. First, the algorithm incorrectly includes the edge Warrigal Drive - QSR, which has a weight of 7, not 6. – graph memorization error) (Second, this incorrect inclusion leads to a faulty sum for this step and an incorrect running total. – state update error) Next are the weight 7 edges.

Finally, to connect the remaining components, I add edges with weights 9 and 10. Total sum: 6+10+18+20+35+54+35+9+30 = 217.

---

**Case Study K.3: Response of Qwen3-32B on Maximum Triangle Sum Task**

**Problem:**
You are required to solve the Maximum Triangle Sum Problem for an undirected wikipedia network. In this network, nodes represent wikipedia articles and edges represent hyperlinks between articles. Each node is assigned a weight. Your objective is to find the triangle with the maximum sum of weights of its three nodes.
Articles in the network: Montebelluna (weight: 1), Massimo Mascioletti (weight: 8), Gianluca Faliva (weight: 4), Eppelheim (weight: 7), United States (weight: 6), Alberto Rebecca (weight: 10), October 15 (weight: 6), Rugby union (weight: 1), Italy (weight: 9), List of football clubs in Italy (weight: 9), Manuel Dallan (weight: 8), Brad Johnstone (weight: 1) Hyperlinks between these articles: Montebelluna and List of football clubs in Italy, Montebelluna and Eppelheim, Montebelluna and Alberto Rebecca, Montebelluna and Italy, Montebelluna and Manuel Dallan, Massimo Mascioletti and Brad Johnstone, Massimo Mascioletti and Gianluca Faliva, Gianluca Faliva and Italy, Gianluca Faliva and Brad Johnstone, Gianluca Faliva and Manuel Dallan, Gianluca Faliva and Rugby union, Eppelheim and Italy, United States and Italy, United States and October 15, Alberto Rebecca and Italy, October 15 and Italy, October 15 and Manuel Dallan, Rugby union and Manuel Dallan, Italy and Manuel Dallan. Identify the triangle with the maximum sum of weights of its three nodes in this network. Present your answer in the following format: \boxed{n}. n is the maximum sum of weights of the triangle.

**Response:**
[Section 1]: Initial Setup and Problem Understanding
Okay, let's try to solve this problem. The task is to find the triangle (a set of three nodes all connected to each other) in the given undirected graph that has the maximum sum of their node weights. The nodes are Wikipedia articles with specific weights, and the edges are hyperlinks between them. First, I need to list out all the nodes and their weights to have a clear view. Let me jot them down: Montebelluna: 1, Massimo Mascioletti: 8, Gianluca Faliva: 4, Eppelheim: 7, United States: 6, Alberto Rebecca: 10, October 15: 6, Rugby union: 1, Italy: 9, List of football clubs in Italy: 9, Manuel Dallan: 8, Brad Johnstone: 1. Now, the edges (hyperlinks) are given. Let me list them out to see which nodes are connected. Each edge connects two nodes, and for a triangle, each pair in the three-node set must be connected. The edges provided are: Montebelluna - List of football clubs in Italy, Montebelluna - Eppelheim, Montebelluna - Alberto Rebecca, Montebelluna - Italy, Montebelluna - Manuel Dallan, Massimo Mascioletti - Brad Johnstone, Massimo Mascioletti - Gianluca Faliva, Gianluca Faliva - Italy, Gianluca Faliva - Brad Johnstone, Gianluca Faliva - Manuel Dallan, Gianluca Faliva - Rugby union, Eppelheim - Italy, United States - Italy, United States - October 15, Alberto Rebecca - Italy, October 15 - Italy, October 15 - Manuel Dallan, Rugby union - Manuel Dallan, Italy - Manuel Dallan.
[Section 2]: Identifying High-Weight Nodes and Initial Triangle Candidates
Now, I need to find all possible triangles. A triangle is three nodes where each pair is connected by an edge. So, for each set of three nodes, check if all three possible edges exist between them. But since there are 12 nodes, checking all combinations might be tedious, but maybe we can focus on nodes with higher weights because we want the maximum sum. High-weight nodes are more likely to be part of the maximum triangle. Looking at the weights, the highest are: Alberto Rebecca: 10, Italy: 9, List of football clubs in Italy: 9, Manuel Dallan: 8, Massimo Mascioletti: 8, United States: 6 (but lower than others), Others are 6, 4, 1, etc. So let's look for triangles involving Alberto Rebecca (10) and Italy (9) since those are high. Looking at the edges connected to Alberto Rebecca. From the edges, there's an edge between Montebelluna and Alberto Rebecca (edge 3), and edge 15: Alberto Rebecca - Italy. So if there's an edge between Montebelluna and Italy, then Montebelluna, Alberto Rebecca, Italy would form a triangle. Looking at edge 4: Montebelluna - Italy. Yes! So Montebelluna, Alberto Rebecca, Italy form a triangle. Their weights are 1 + 10 + 9 = 20. Is there a better one? Another high-weight node is Italy (9). Let's check other nodes connected to Italy. Italy is connected to: Montebelluna (edge 4), Gianluca Faliva (edge 8), Eppelheim (edge 12), Alberto Rebecca (edge 15), United States (edge 13), Manuel Dallan (edge 19). So any triangle involving Italy and two other nodes connected to Italy. Let's see. For example, if there's a node

connected to both Italy and another node connected to Italy. Let's check combinations. Take Gianluca Faliva (4) connected to Italy (edge 8). What other nodes connected to Italy are connected to Gianluca Faliva? Looking at Gianluca Faliva's edges: Edges connected to Gianluca Faliva: 7 (Massimo Mascioletti), 8 (Italy), 9 (Brad Johnstone), 10 (Manuel Dallan), 11 (Rugby union). So connected to Italy and Manuel Dallan, Brad Johnstone, etc. Now, is there a connection between Italy and Manuel Dallan? Yes, edge 19. Also, is there a connection between Gianluca Faliva and Italy (edge 8) and between Gianluca Faliva and Manuel Dallan (edge 10), and between Italy and Manuel Dallan (edge 19). So the triangle Gianluca Faliva, Italy, Manuel Dallan. Their weights: 4 + 9 + 8 = 21. That's higher than 20.

[Section 3]: Checking Other High-Weight Node Combinations for Better Triangles

Another candidate: Italy, United States, and October 15. The edges are United States - Italy (edge 13), United States - October 15 (edge 14), and Italy - October 15 (edge 16). So that's a triangle. Their weights: 6 (US) + 9 (Italy) + 6 (October 15) = 21. Same as the previous 21. So same sum. Now, what about the triangle Gianluca Faliva, Italy, Manuel Dallan (sum 21) and the triangle United States, Italy, October 15 (sum 21). Are there any triangles with higher sums? Let's check other high-weight nodes. For example, Manuel Dallan (8) and Massimo Mascioletti (8). Are they connected? Let me check. Massimo Mascioletti is connected to Brad Johnstone (edge 6) and Gianluca Faliva (edge 7). Gianluca Faliva is connected to Manuel Dallan (edge 10). But is there a direct connection between Massimo Mascioletti and Manuel Dallan? Not that I can see. So no triangle there. What about Massimo Mascioletti (8), Gianluca Faliva (4), Brad Johnstone (1). They form a triangle? Let's check edges: Massimo Mascioletti is connected to Gianluca Faliva (edge7) and Brad Johnstone (edge6). Gianluca Faliva is connected to Brad Johnstone (edge9). So yes, triangle Massimo Mascioletti, Gianluca Faliva, Brad Johnstone. Sum is 8 + 4 + 1 = 13. Not better. Another high-weight node is List of football clubs in Italy (9). Let's check if it's part of a triangle. The only edge connected to it is Montebelluna (edge1). Are there other edges? Let me check. The given edges don't show any other connections for List of football clubs in Italy. So unless Montebelluna is connected to another node that's connected to List of football clubs in Italy, but since Montebelluna is connected to List of football clubs in Italy and to Italy, but Italy isn't connected to List of football clubs in Italy? Let me check. The edges don't mention an edge between Italy and List of football clubs in Italy. So the triangle would need Montebelluna, List of football clubs in Italy, and another node connected to both. But since List of football clubs in Italy is only connected to Montebelluna, no triangle possible here. What about the triangle involving Alberto Rebecca (10), Italy (9), and someone else. For example, Alberto Rebecca is connected to Montebelluna (edge3) and Italy (edge15). If Montebelluna is connected to Italy (edge4), which it is. So that triangle is already considered (sum 20). Are there others? For example, Alberto Rebecca connected to someone else who's connected to Italy. Let's see. Suppose Alberto Rebecca is connected to another node connected to Italy. But in the edges given, Alberto Rebecca is only connected to Montebelluna and Italy. So no other connections. So no other triangles involving Alberto Rebecca except with Montebelluna and Italy. Another candidate: Manuel Dallan (8), Italy (9), and someone else. Let's see. For example, if there's a node connected to both Manuel Dallan and Italy. Let's check. October 15 is connected to Italy (edge16) and to Manuel Dallan (edge17). So the triangle Italy, Manuel Dallan, October 15. Which we already considered (sum 21). Also, Rugby union is connected to Manuel Dallan (edge18), but is Rugby union connected to Italy? No. So no triangle there. What about other combinations? Let's check if there's a triangle with higher than 21. Let's think about nodes with high weights. The highest weights are 10, 9, 9, 8, 8. Let's see if there's a triangle with two 9s and an 8. For example, Italy (9), List of football clubs in Italy (9), and someone connected to both. Are they connected? The edges don't mention a direct link between Italy and List of football clubs in Italy. So unless there's another node connected to both. Let's see. Montebelluna is connected to both List of football clubs in Italy (edge1) and to Italy (edge4). So the triangle would be Montebelluna (1), Italy (9), List of football clubs in Italy (9). (This is incorrect. The response correctly identifies that Montebelluna is connected to both Italy and List of football clubs in Italy, but it fails to verify the third required edge between Italy and List of football clubs in Italy before declaring it a triangle. This is a graph memorization error). Sum is 1 + 9 + 9 = 19, which is less than 21.

[Section 4]: Exhaustive Search for All Possible Triangles and Verification of Maximum Sum

What about the triangle between Italy (9), Eppelheim (7), and someone else? For example, Eppelheim is connected to Italy (edge12) and to Montebelluna (edge2). But Montebelluna isn't connected to Italy except through edge4. Wait, Montebelluna is connected to Italy (edge4), so

the triangle would be Montebelluna (1), Eppelheim (7), Italy (9). Sum is 1+7+9=17. Not better. Another candidate: United States (6), Italy (9), and someone else. The only other connection is October 15 (edge14 and 16). So the triangle is already considered. Is there a triangle with three high-weight nodes? For example, Italy (9), List of football clubs in Italy (9), and someone else. But they aren't connected. So no. What about Manuel Dallan (8), List of football clubs in Italy (9), and someone connected to both? List of football clubs in Italy is only connected to Montebelluna (1), and Manuel Dallan is connected to Montebelluna (edge5). So the triangle would be Montebelluna (1), List of football clubs in Italy (9), Manuel Dallan (8). (This is incorrect. The response finds that Montebelluna is a common neighbor to both 'List of football clubs in Italy' and 'Manuel Dallan' but incorrectly assumes this forms a triangle without verifying the missing third edge between 'List of football clubs in Italy' and 'Manuel Dallan' – graph memorization error). Sum is 1+9+8=18. Not better. Another possibility: Is there a triangle involving Alberto Rebecca (10), Manuel Dallan (8), and someone else? Let's see. Alberto Rebecca is connected to Montebelluna (1) and Italy (9). Manuel Dallan is connected to Italy. But there's no direct edge between Alberto Rebecca and Manuel Dallan. So no triangle there. What about the triangle between Gianluca Faliva (4), Manuel Dallan (8), and Rugby union (1)? They are connected via edges 10 and 18. But is there an edge between Gianluca Faliva and Rugby union? Yes (edge11). And between Rugby union and Manuel Dallan? Yes (edge18). And between Gianluca Faliva and Manuel Dallan (edge10). So the triangle is Gianluca Faliva (4), Manuel Dallan (8), Rugby union (1). Sum is 4+8+1=13. Not better. Another check: Are there any other triangles with high sums? Let's think. For example, United States (6), Italy (9), and someone else. The only other connection is October 15 (6). So that's the triangle we already have. Sum 21. Is there a triangle with two 8s? For example, Massimo Mascioletti (8) and Manuel Dallan (8). But they aren't connected. What about Manuel Dallan (8) and someone else? Let's see. The only other node with 8 is Massimo Mascioletti (8). But no connection. So no triangle with two 8s. What about the triangle between Manuel Dallan (8), Italy (9), and another node with 8? For example, if there's a node connected to both Manuel Dallan and Italy. But the only such node is Gianluca Faliva (4) or October 15 (6). So no. So far, the maximum sum is 21, achieved by two different triangles: (Gianluca Faliva, Italy, Manuel Dallan) and (United States, Italy, October 15). Are there any other triangles with sum 21 or higher? Let me check another angle. Let's look for all triangles and compute their sums. First, list all possible triangles. Since this is time-consuming, but maybe I can find all possible triangles by looking at each node's connections. Starting with Italy (9), since it's connected to many nodes. Let's check all pairs of nodes connected to Italy and see if they are connected to each other. Italy is connected to: Montebelluna (M), Gianluca Faliva (G), Eppelheim (E), Alberto Rebecca (A), United States (US), October 15 (O), Manuel Dallan (D). So for each pair among these, check if they are connected. Check pairs: M and G: Are Montebelluna and Gianluca Faliva connected? No. The edges connected to Montebelluna are List of football clubs in Italy, Eppelheim, Alberto Rebecca, Italy, and Manuel Dallan. So no direct edge between M and G. M and E: Montebelluna and Eppelheim are connected (edge2). So the triangle M, E, Italy. Sum is 1+7+9=17. M and A: Montebelluna and Alberto Rebecca are connected (edge3). So triangle M, A, Italy. Sum 1+10+9=20. M and US: Are Montebelluna and United States connected? No. So no triangle. M and O: No connection between Montebelluna and October 15. M and D: Montebelluna and D are connected (edge5). So triangle M, D, Italy. Sum 1+8+9=18. G and E: Are Gianluca Faliva and Eppelheim connected? No. So no triangle. G and A: Gianluca Faliva and Alberto Rebecca? No. No edge between them. G and US: Gianluca Faliva and United States? No. G and O: Gianluca Faliva and October 15? No. G and D: Gianluca Faliva and D are connected (edge10). So triangle G, D, Italy. Sum 4+8+9=21. This is one of the candidates. E and A: Eppelheim and Alberto Rebecca? No. No edge between them. E and US: Eppelheim and US? No. E and O: No. E and D: No. A and US: Alberto Rebecca and US? No. A and O: No. A and D: No. US and O: Yes, connected via edge14. Also, US and Italy (edge13), O and Italy (edge16). So triangle US, O, Italy. Sum 6+6+9=21. US and D: Are US and D connected? No. So no triangle. O and D: Yes, connected via edge17. So triangle O, D, Italy (already considered). So from Italy's connections, the triangles with sum 21 are: G (4), D (8), Italy (9): sum 21, US (6), O (6), Italy (9): sum 21. Now check other nodes not connected to Italy. For example, check if there are triangles among other nodes. For example, check Gianluca Faliva's connections. He is connected to Massimo Mascioletti (Masc), Brad Johnstone (B), Manuel Dallan (D), Rugby union (R), and Italy (I). Check pairs among these: Masc and B: connected via edge6? Yes. Masc and Gianluca (edge7), B and Gianluca (edge9). Also, Masc and B are connected (edge6). So triangle

Masc, B, Gianluca. Sum 8+1+4=13. Masc and D: Are they connected? No. Masc is connected to Gianluca and B, but not directly to D. So no. Masc and R: No. B and D: connected via edge9 (B and Gianluca) and edge10 (Gianluca and D). But B and D? No. So no triangle. B and R: No. D and R: connected via edge18 (D and R). But Gianluca and R are connected (edge11). So triangle Gianluca (4), D (8), R (1). Sum 4+8+1=13. Other combinations? No. Check Manuel Dallan's connections: Italy, October 15, Rugby union, Montebelluna. Check if any of those form triangles not involving Italy. For example, October 15 and Rugby union? No. October 15 and Rugby union are not connected. So no. Check October 15's connections: United States, Italy, Manuel Dallan. The triangle US-Italy-O15 is already considered. Check other nodes. For example, Massimo Mascioletti (8) connected to Brad Johnstone (1) and Gianluca (4). No other connections. No triangles. Alberto Rebecca (10) connected to Montebelluna (1) and Italy (9). No other connections. So no other triangles. List of football clubs in Italy is only connected to Montebelluna. So no triangles. Eppelheim is connected to Montebelluna and Italy. No other connections. So no other triangles. United States is connected to Italy and October 15. No other connections except those. Rugby union is connected to Gianluca and Manuel Dallan. No other connections. So all possible triangles have been checked. The maximum sum is 21.

