# OpenReview forum: "Exposing Weaknesses of Large Reasoning Models through Graph Algorithm Problems"
_ICLR.cc/2026/Conference — ICLR 2026 Poster_

### Official Review · Reviewer_c193 · 2025-10-25

**Soundness:** 3
**Presentation:** 3
**Contribution:** 2
**Rating:** 4
**Confidence:** 3

**Summary:**

The paper proposes a benchmark for Large Reasoning Models (LRMs) that consists of questions instantiated from various graph algorithm problems. The benchmark reveals two shortcomings of state-of-the-art LRMs: poor scalability to long contexts and overthinking.

**Strengths:**

- Despite the abundance of graph-based reasoning benchmarks in the literature, the idea of challenging LRMs with graph algorithm problems of varying complexity seems novel to me.

**Weaknesses:**

- The evaluation of the paper measures the difficulty of questions primarily by the size of their underlying graphs rather than the computational complexity of their underlying graph algorithm problems. Since the novelty of the paper lies primarily in its graph algorithm perspective, its evaluation should focus on the impact of computational complexity rather than graph size.
- Some of the graph algorithm problems, e.g. minimum spanning tree, may not resemble questions in real life.

**Questions:**

- Can the numbers in Table 3 provide additional insights if they are grouped by computational complexity rather than graph size?

---

> ### Author Response · Authors · 2025-11-19
> **Rebuttal by Authors**
>
> Thank you for your constructive feedback! Our response will address the following points: (1) new insights derived from grouping problems by their computational complexity, and (2) real‑world applications of the minimum spanning tree.
>
> ### W1 and Q1: New Insights from Grouping Problems by Computational Complexity
> > W1: ...its evaluation should focus on the impact of computational complexity rather than graph size.
>
> > Q1: Can the numbers in Table 3 provide additional insights if they are grouped by computational complexity rather than graph size?
>
> You raise a great point regarding our evaluation settings. We agree that grouping tasks by **computational complexity** rather than graph size can reveal additional insights. To further investigate this, we fixed the number of nodes (two versions: *Level‑3*, 31–50 nodes; *Level‑4*, 51–80 nodes) and grouped tasks according to their theoretical time complexity:
> - **Easy:**  O(n)
> - **Medium:** O(n²)
> - **Hard:**  O(n³) or NP‑hard
>
> We then compared model performance across these complexity levels. *(“Enum”, “Ex”, and “In” denote the **Enumeration**, **Exploration**, and **Intuition** reasoning types, respectively.)*
>
>
>
> ### Level‑3 (31–50 nodes)
>
> | Model (pass@k / cons@k) | Max Degree (En: easy) | Path Sum (Ex: easy) | MKC (In: easy) | Triangle (En: medium) | Distance‑K (Ex: medium) | MST (In: medium) | MCP (En: hard) | Diameter (Ex: hard) | Distance Threshold (In: hard) |
> |:---|:---:|:---:|:---:|:---:|:---:|:---:|:---:|:---:|:---:|
> | Distill_Qwen_32B | 1.00 / 0.80 | 1.00 / 0.74 | 0.60 / 0.28 | 0.90 / 0.66 | 0.66 / 0.34 | 0.20 / 0.00 | 0.72 / 0.36 | 0.72 / 0.32 | 0.24 / 0.24 |
> | Light‑R1‑32B | 0.90 / 0.84 | 0.96 / 0.90 | 0.62 / 0.34 | 0.94 / 0.80 | 0.84 / 0.50 | 0.33 / 0.07 | 1.00 / 0.66 | 0.92 / 0.56 | 0.56 / 0.50 |
> | Llama‑3.3‑70B | 0.78 / 0.32 | 0.56 / 0.28 | 0.06 / 0.02 | 0.12 / 0.00 | 0.34 / 0.04 | 0.00 / 0.00 | 0.08 / 0.00 | 0.20 / 0.02 | 0.00 / 0.00 |
> | OpenThinker‑32B | 0.96 / 0.78 | 0.98 / 0.80 | 0.62 / 0.36 | 0.98 / 0.80 | 0.76 / 0.44 | 0.30 / 0.10 | 0.94 / 0.68 | 0.90 / 0.40 | 0.40 / 0.30 |
> | QWQ‑32B | 0.98 / 0.96 | 1.00 / 0.98 | 0.90 / 0.64 | 1.00 / 0.94 | 0.88 / 0.64 | 0.70 / 0.50 | 0.92 / 0.76 | 0.90 / 0.48 | 0.80 / 0.60 |
> | Qwen2.5‑32B | 0.40 / 0.12 | 0.08 / 0.00 | 0.40 / 0.12 | 0.82 / 0.44 | 0.58 / 0.18 | 0.00 / 0.00 | 0.80 / 0.26 | 0.58 / 0.18 | 0.12 / 0.00 |
> | Qwen3‑32B | 0.98 / 0.76 | 1.00 / 0.98 | 0.78 / 0.46 | 1.00 / 0.80 | 0.88 / 0.66 | 0.63 / 0.30 | 0.98 / 0.76 | 0.90 / 0.48 | 0.64 / 0.56 |
> | Qwen3‑32B‑no‑thinking | 0.84 / 0.36 | 0.94 / 0.58 | 0.30 / 0.02 | 0.68 / 0.44 | 0.72 / 0.32 | 0.13 / 0.03 | 0.54 / 0.20 | 0.66 / 0.20 | 0.24 / 0.02 |
> | Skywork‑OR1‑32B | 1.00 / 0.94 | 1.00 / 1.00 | 0.90 / 0.76 | 1.00 / 0.92 | 1.00 / 0.82 | 0.67 / 0.27 | 0.98 / 0.82 | 0.94 / 0.52 | 0.54 / 0.38 |
> | gpt_oss_20b | 0.98 / 0.90 | 1.00 / 1.00 | 0.82 / 0.60 | 0.98 / 0.88 | 0.94 / 0.76 | 0.73 / 0.50 | 0.96 / 0.78 | 0.88 / 0.44 | 0.42 / 0.22 |
>
>
> ### Level‑4 (51–80 nodes)
>
> | Model (pass@k / cons@k) | Max Degree (En: easy) | Path Sum (Ex: easy) | MKC (In: easy) | Triangle (En: medium) | Distance‑K (Ex: medium) | MST (In: medium) | MCP (En: hard) | Diameter (Ex: hard) | Distance Threshold (In: hard) |
> |:---|:---:|:---:|:---:|:---:|:---:|:---:|:---:|:---:|:---:|
> | Distill_Qwen_32B | 0.93 / 0.57 | 0.67 / 0.30 | 0.27 / 0.07 | 0.80 / 0.43 | 0.40 / 0.20 | 0.16 / 0.06 | 0.70 / 0.33 | 0.60 / 0.20 | 0.03 / 0.03 |
> | Light‑R1‑32B | 0.60 / 0.50 | 0.90 / 0.53 | 0.23 / 0.03 | 0.70 / 0.37 | 0.57 / 0.33 | 0.30 / 0.12 | 0.87 / 0.50 | 0.67 / 0.23 | 0.07 / 0.07 |
> | Llama‑3.3‑70B | 0.47 / 0.13 | 0.10 / 0.00 | 0.03 / 0.00 | 0.13 / 0.00 | 0.17 / 0.00 | 0.00 / 0.00 | 0.10 / 0.00 | 0.23 / 0.00 | 0.00 / 0.00 |
> | OpenThinker‑32B | 0.90 / 0.37 | 0.80 / 0.37 | 0.13 / 0.03 | 0.80 / 0.50 | 0.50 / 0.23 | 0.36 / 0.14 | 0.80 / 0.40 | 0.87 / 0.27 | 0.13 / 0.13 |
> | QWQ‑32B | 1.00 / 0.73 | 1.00 / 0.90 | 0.23 / 0.10 | 0.90 / 0.70 | 0.63 / 0.43 | 0.46 / 0.36 | 0.90 / 0.57 | 0.87 / 0.37 | 0.70 / 0.33 |
> | Qwen2.5‑32B | 0.27 / 0.00 | 0.00 / 0.00 | 0.17 / 0.03 | 0.57 / 0.20 | 0.30 / 0.07 | 0.02 / 0.00 | 0.70 / 0.20 | 0.57 / 0.07 | 0.20 / 0.10 |
> | Qwen3‑32B | 0.93 / 0.40 | 0.90 / 0.77 | 0.30 / 0.03 | 0.87 / 0.63 | 0.77 / 0.47 | 0.48 / 0.14 | 0.83 / 0.53 | 0.90 / 0.23 | 0.50 / 0.33 |
> | Qwen3‑32B‑no‑thinking | 0.60 / 0.23 | 0.50 / 0.07 | 0.13 / 0.00 | 0.60 / 0.17 | 0.53 / 0.27 | 0.04 / 0.00 | 0.53 / 0.03 | 0.60 / 0.23 | 0.13 / 0.03 |
> | Skywork‑OR1‑32B | 0.93 / 0.53 | 0.93 / 0.83 | 0.37 / 0.17 | 0.80 / 0.37 | 0.67 / 0.37 | 0.52 / 0.40 | 0.80 / 0.50 | 0.93 / 0.30 | 0.00 / 0.00 |
> | gpt_oss_20b | 0.97 / 0.83 | 0.80 / 0.67 | 0.37 / 0.17 | 0.70 / 0.50 | 0.57 / 0.47 | 0.34 / 0.20 | 0.67 / 0.43 | 0.63 / 0.37 | 0.63 / 0.50 |

---

> ### Author Response · Authors · 2025-11-19
> **Rebuttal by Authors**
>
> ### Insghts
>
> 1.  **Performance generally declines with increasing computational complexity within the same algorithmic category.**
>     For instance, models such as **QWQ‑32B** and **Qwen3‑32B** achieve near‑perfect performance on simple *O(n)* tasks like *Max Degree* or *Path Sum*, yet show a marked drop on *O(n²)* and especially NP‑hard problems such as *Distance Threshold*. This trend indicates that higher algorithmic complexity expands the search space, posing greater reasoning challenges for current large reasoning models.
>
> 2.  **However, we observe a notable anomaly within the *Exploration* category: models often perform better on the "Hard" *Diameter* task than on the "Medium" *Distance-K* task.**
>     Despite *Diameter* having a higher theoretical complexity ($O(n^3)$) than *Distance-K* ($O(n^2)$), several top models score significantly higher on the former. For example, on Level-4 graphs, **QWQ-32B** achieves a pass rate of **0.87** on *Diameter* compared to **0.63** on *Distance-K*, and **OpenThinker-32B** scores **0.87** versus **0.50**. We hypothesize that this discrepancy stems primarily from differences in **training data distribution**.
>
> 3.  **Reasoning paradigm plays a critical role, often outweighing theoretical complexity: models excel at *Enumeration* but struggle with *Intuition*.**
>     Our data reveals that "Hard" Enumeration tasks often yield better performance than "Easy" Intuition tasks. For instance, on Level-3 graphs, **Qwen3-32B** achieves a perfect **0.98** on *MCP* (Enumeration: Hard) but only **0.78** on *MKC* (Intuition: Easy). Similarly, **Qwen2.5-32B** scores **0.80** on *MCP* compared to just **0.40** on *MKC*.
>     This suggests that Large Reasoning Models are inherently better suited for systematic reasoning processes (characteristic of Enumeration) rather than tasks requiring global heuristic judgments or pattern recognition (characteristic of Intuition), even when the search space for the former is significantly larger.
>
> We have incorporated these updated experimental results and findings into the revised version of our paper (Appendix J). Thank you again for your valuable suggestions!
>
> ---
>
> ### W2: Real-World Applications of Minimum Spanning Tree
> > W2: Some of the graph algorithm problems, e.g. minimum spanning tree, may not resemble questions in real life.
>
> We appreciate this comment, as it helps us better clarify the practical relevance of our benchmark. Although the *Minimum Spanning Tree (MST)* problem may appear to be an abstract graph‑theoretic task, it has broad real‑world applications [1][2][3][4][5]. In **network design**, for instance, MST is widely used to minimize the total cost of connecting multiple nodes—such as cities, routers, or base stations—in **fiber‑optic**, **local area**, or **telecommunication backbone networks**. A typical example is when a telecom company aims to connect 20 cities so that all are reachable with minimal wiring cost, where the MST model directly provides the optimal configuration. Similarly, in **transportation and logistics planning**, MST helps design cost‑efficient road or railway systems by identifying the minimum‑cost way to connect multiple locations or hubs. Therefore, including MST in our benchmark not only evaluates model performance on classical algorithmic reasoning but also reflects their ability to handle optimization problems closely aligned with real‑world infrastructure design.
>
> [1] Bakar et al. **PRIM-MST: AN Algorithm for Designing and Optimizing Local Area Network Planning**. International Journal of Computer Science & Network Security, 2024.
>
> [2] Sinaga et al. **Prim’s algorithm to model the pipe network at the water supply company**. Journal of Physics: Conference Series. IOP Publishing, 2019.
>
> [3] Son et al. **On the design and optimization of a free space optical access network**. Optical Switching and Networking, 2014.
>
> [4] Felzenszwalb et al. **Efficient graph-based image segmentation**. International journal of computer vision, 2004.
>
> [5] Huang et al. **Multicast routing for multimedia communications in the Internet of Things**. IEEE Internet of Things Journal, 2016.

---

> ### Author Response · Authors · 2025-11-27
> **Kind Reminder for Rebuttal**
>
> Dear Reviewer c193,
>
> As the rebuttal period draws to a close, we are very interested in hearing any further feedback you might have and would be pleased to clarify or address any remaining concerns. Please do not hesitate to let us know if you have additional comments or questions.
>
> Thank you again for your time and thoughtful review.
>
> Best regards,
> Authors of Paper  8673

---

### Official Review · Reviewer_dfQZ · 2025-10-30

**Soundness:** 3
**Presentation:** 3
**Contribution:** 3
**Rating:** 6
**Confidence:** 4

**Summary:**

The authors propose a new benchmark named GRALGOBENCH to evaluate the long-context reasoning ability of Large Reasoning Models (LRMs) through graph algorithm problems. Specifically, this benchmark includes three major categories of problems: Enumeration, Exploration, and Intuition. Each category is further classified into three difficulty levels: easy, medium, and hard (3*3=9 tasks), and each task is set with 6 levels based on the number of nodes and edges. Based on this benchmark, the authors conducted systematic experiments to explore the long-text reasoning ability of the current LRMs. In summary, the accuracy of LRMs in long contexts has dropped sharply. Secondly, LRMs are plagued by the phenomenon of overthinking, which is mainly caused by frequent and ineffective self-verification.

**Strengths:**

1.	Well-motivated benchmark & taxonomy. The bridge from CLRS-style algorithmic families to the Enumeration / Exploration / Intuition taxonomy is clear and useful for reasoning analysis.

2.	Real-world graph sources & scaling. Instances are derived from DBLP, street networks (multiple cities), OpenFlights, Wikipedia, and DBpedia; each task is generated across six scales to modulate difficulty, supporting long-context evaluation without synthetic toy artifacts.

3.	Comprehensive evaluation & clear trends. Results across both reasoning and non-reasoning models show consistent degradation with context length and with graph size, and reveal a robust hierarchy of task difficulty.

4.	Thoughtful error analysis. The paper proposes a structured error taxonomy (algorithm selection vs. execution vs. output quality vs. information/memorization) and demonstrates where LRMs fail.

5.	Over-thinking analysis. The segmentation of traces with high-entropy tokens and labeling of self-reflection/strategy-shift provides evidence that “self-verification” often inflates tokens with little gain; the “outcome efficiency” plots are compelling.

**Weaknesses:**

1.	LLM-as-judge dependence. Error categorization and over-thinking judgments rely on LLM pipelines. The paper would benefit from reporting inter-annotator agreement with humans on a stratified subset to quantify labeling reliability and possible bias.

2.	On line 72 you note that GraphWalks is intended to benchmark LRMs’ long-context capabilities. Please clarify how your benchmark compares to GraphWalks in scope, task design, evaluation protocol, and key findings. In particular, explain what is novel or complementary (e.g., task families, graph sources, scaling, automatic judging, contamination checks, tool-use assumptions) and, where feasible, include a side-by-side quantitative comparison to justify the need for a new benchmark.

3.	Tool-use / code execution. Many LRMs now include tool-use or code-execution modes. The benchmark fixes models to text-only. That is fair for isolating intrinsic reasoning, but the paper should discuss how results might differ under code-execution assistance and whether GRALGOBENCH can evaluate that regime.

4.	External validity and scope. The benchmark focuses exclusively on deterministic graph-algorithm tasks. While this is a valuable slice of reasoning, it omits other long-context regimes. Without evidence of cross-task transfer, it’s unclear whether conclusions about long-context brittleness and over-thinking generalize beyond graph-structured problems, other data modalities (tables, trees, sequences), or multilingual settings.

**Questions:**

1.	How consistent are the error labels across judges? It might be good to  report a human-checked agreement rate on subset samples, stratified by taxonomy/level.

2.	In the verbosity experiments, how are node names expanded and how is the added text distributed across the prompt (e.g., per-node descriptors vs. preamble)? Can you share a few full prompts at each length?

3.	Have you tried a token-budget cap (e.g., limiting self-verification tokens) to quantify over-thinking’s cost-benefit? The efficiency plots suggest diminishing returns; an intervention study would be valuable.

4.	In Figure 3, several curves appear to run at or below zero. Maybe tightening the y-axis limits and/or adding clipping can avoid visual ambiguity.

5.	On transfer/generalization beyond graphs. Can you provide transfer evidence that performance and observed phenomena on this benchmark relate to broader “general reasoning” tasks? For example:

(i) model-wise correlations between GRALGOBENCH scores and external benchmarks targeting long-context/general reasoning (e.g., LongBench, Needle-in-a-Haystack variants, BIG-bench Hard, GPQA, MMLU-Pro, long-form GSM8K),

(ii) an intervention study showing that a method which improves GRALGOBENCH (prompting, memory limits, self-verification caps) also improves one or more non-graph long-context tasks, and/or

(iii) an ablation that rephrases graph instances into equivalent textual narratives or table/tree forms to test whether the observed failures stem from graph structure per se versus generic context-length effects. If these are out of scope, please temper the generality of your conclusions accordingly.

---

> ### Author Response · Authors · 2025-11-19
> **Rebuttal by Authors**
>
> We are grateful for the reviewer’s thorough and constructive feedback. Our responses will address the following key points: (1) inter‑judge consistency and human‑verified reliability; (2) the relation to GraphWalks and what aspects are novel or complementary; (3) tool use and code execution—specifically, whether GRALGOBENCH can evaluate code‑generation abilities and how the results might differ; (4) generalization of our conclusions beyond graph‑related tasks; (5) clarification on how node names are expanded and where the additional text is inserted within the prompt; (6) analysis of token‑budget constraints and the cost–benefit trade‑off of over‑thinking; and (7) adjustments to the figures.
>
>
>
> ### **W1/Q1: On inter‑judge consistency and human‑checked reliability**
>
> > W1: ..The paper would benefit from reporting inter-annotator agreement with humans on a stratified subset to quantify labeling reliability and possible bias.
>
> > Q1: How consistent are the error labels across judges? It might be good to report a human-checked agreement..
>
> We thank the reviewer for raising this important question on the **dependence of our error‑analysis pipeline on LLM‑as‑judge models** and for suggesting to include **human‑checked agreement metrics**.
> Following the recommendation, we incorporated human annotation to quantitatively evaluate labeling consistency and possible biases.
>
>
>
> #### **(1) Verification strategy**
>
> Because exhaustively verifying every reasoning trace is impractical, we adopted a **two‑stage evaluation**:
>
> 1. **Full LLM‑based labeling** –
>    We applied **three independent judges** (**GPT‑5**, **DeepSeek‑R1**, and **O3‑mini**) to assign four‑way error categories — *Redundancy / Graph Memory / Execution Error / Others* — to all reasoning traces in the benchmark.
>
> 2. **Human cross‑check on a stratified subset** –
>    We then sampled **50 traces from each reasoning paradigm** (Enumeration, Exploration, Intuition) to cover different task levels and both thinking‑ and no‑thinking‑mode traces.
>    These **150 traces** were independently labeled by **three postgraduate annotators** with competitive‑programming experience, using the same taxonomy. In cases of labeling disagreement, the annotators discussed collectively to reach a consensus on the final label.
>
>    The aggregated human results are directly compared with the three LLM judges below.
>
>
>
> #### **(2) Inter‑judge results and consistency**
>
> #### **Qwen3‑32B‑thinking**
>
> #### *O3‑mini*
>
> | Paradigm | Redundancy | Graph Memory | Execution Error | Others |
> |-----------|-------------|---------------|----------------|--------|
> | Enumeration | 53.6 % | 17.1 % | 10.5 % | 18.8 % |
> | Exploration | 37.7 % | 21.6 % | 35.3 % | 5.4 % |
> | Intuition   | 24.6 % | 21.0 % | 48.6 % | 5.8 % |
>
> #### *DeepSeek‑R1*
>
> | Paradigm | Redundancy | Graph Memory | Execution Error | Others |
> |-----------|-------------|---------------|----------------|--------|
> | Enumeration | 56.1 % | 16.5 % | 12.1 % | 15.3 % |
> | Exploration | 40.4 % | 19.5 % | 33.9 % | 6.2 % |
> | Intuition   | 28.2 % | 18.9 % | 45.7 % | 7.2 % |
>
> #### *GPT‑5*
>
> | Paradigm | Redundancy | Graph Memory | Execution Error | Others |
> |-----------|-------------|---------------|----------------|--------|
> | Enumeration | 49.1 % | 15.8 % | 13.9 % | 21.2 % |
> | Exploration | 38.9 % | 19.6 % | 34.2 % | 7.3 % |
> | Intuition   | 29.4 % | 20.3 % | 45.1 % | 5.2 % |
>
> #### *Human*
>
> | Paradigm     | Redundancy | Graph Memory | Execution Error | Others |
> |---------------|-------------|---------------|----------------|---------|
> | Enumeration | 52.0 % | 16.0 % | 12.0 % | 20.0 % |
> | Exploration | 38.0 % | 20.0 % | 36.0 % | 6.0 % |
> | Intuition  | 26.0 % | 18.0 % | 46.0 % | 10.0 % |

---

> > ### Author Response · Authors · 2025-11-19
> > **Rebuttal by Authors**
> >
> > #### **Qwen3‑32B‑no‑thinking**
> >
> > #### *O3‑mini*
> >
> > | Paradigm | Redundancy | Graph Memory | Execution Error | Others |
> > |-----------|-------------|---------------|----------------|--------|
> > | Enumeration | 16.1 % | 42.6 % | 28.9 % | 12.4 % |
> > | Exploration | 5.4 %  | 38.7 % | 54.8 % | 1.1 % |
> > | Intuition   | 11.3 % | 27.2 % | 58.4 % | 3.1 % |
> >
> > #### *GPT‑5*
> >
> > | Paradigm | Redundancy | Graph Memory | Execution Error | Others |
> > |-----------|-------------|---------------|----------------|--------|
> > | Enumeration | 14.9 % | 40.4 % | 30.1 % | 14.6 % |
> > | Exploration | 6.6 %  | 36.9 % | 53.1 % | 3.4 % |
> > | Intuition   | 10.8 % | 29.6 % | 55.5 % | 4.1 % |
> >
> >
> > #### *DeepSeek‑R1*
> >
> > | Paradigm | Redundancy | Graph Memory | Execution Error | Others |
> > |-----------|-------------|---------------|----------------|--------|
> > | Enumeration | 17.1 % | 39.4 % | 29.2 % | 14.3 % |
> > | Exploration | 6.5 %  | 39.8 % | 51.2 % | 2.5 % |
> > | Intuition   | 12.3 % | 28.5 % | 55.4 % | 3.8 % |
> >
> >
> > #### *Human*
> >
> > | Paradigm     | Redundancy | Graph Memory | Execution Error | Others |
> > |---------------|-------------|---------------|----------------|---------|
> > | Enumeration | 18.0 % | 36.0 % | 32.0 % | 14.0 % |
> > | Exploration | 8.0 %  | 42.0 % | 48.0 % | 2.0 % |
> > | Intuition   | 8.0 % | 28.0 % | 60.0 % | 4.0 % |
> >
> >
> >
> > Across judge models and the human annotators, the **qualitative trends are highly consistent**:
> >
> > - For **thinking** traces, the dominant issues are **redundancy**, **execution inaccuracies**, and moderate **graph‑memory errors**.
> > - For **no‑thinking** traces, **execution errors** and **graph‑memory failures** dominate, while redundancy decreases.
> >
> > Despite minor quantitative differences, human assessments align closely with the LLM ensemble, confirming that the observed error taxonomy and its relative proportions are stable across judging sources.
> >
> >
> >
> > #### **(3) Reliability justification**
> >
> > Moreover, our workflow—first using **Qwen‑2.5‑72B** for segmentation, then using **O3‑mini** to assign error types—is directly inspired by **ProcessBench** [1]. Their experiments show that **O1‑mini** can reach an error‑classification accuracy of **93.2%** on GSM8K process supervision, and **O3‑mini** is a stronger, more recent model from the same family. Besies, [2] provides a detailed discussion on the feasibility of LLM‑as‑a‑judge. These studies give us strong prior evidence that such judge models are capable of reliably performing this kind of textual error analysis.
> >
> >
> >
> > [1] Zheng et al. **ProcessBench: Identifying process errors in mathematical reasoning**, in ACL 2025.
> >
> > [2] Gu et al. **A Survey on LLM-as-a-Judge**. ArXiv Preprint 2024.

---

> > ### Author Response · Authors · 2025-11-19
> > **Rebuttal by Authors**
> >
> > ### W2: Relation to GraphWalks and what is novel/complementary
> > > W2: ...Please clarify how your benchmark compares to GraphWalks in scope, task design, evaluation protocol, and key findings...
> >
> > We thank the reviewer for pointing out GraphWalks, which shares a similar high‑level goal of probing long‑context capabilities with graph‑structured inputs. We view GraAlgoBench as complementary rather than redundant, and clarify the main differences below.
> >
> > - **Scope & difficulty.**
> >   GraphWalks focuses on a **single** synthetic graph‑walk task whose hardest cases involve roughly **O(n)**‑style walk reasoning. GraAlgoBench contains **9 distinct graph problems**, ranging from linear‑time algorithms to **NP‑hard** graph optimization problems, allowing us to probe much harder global reasoning behaviors.
> >
> > - **Task families & reasoning types.**
> >   We explicitly organize tasks into three **reasoning paradigms**—**Enumeration, Exploration, Intuition**—to analyze how models behave under different styles of graph reasoning. GraphWalks does not make such a taxonomy and centers on one family of walk tasks.
> >
> > - **Graph sources & real‑world grounding.**
> >   GraphWalks uses synthetic graphs with hex‑like node IDs. GraAlgoBench combines **real‑world graphs** (DBLP, Wikipedia, street networks, etc.) with synthetic graphs, which both reduces contamination risk and tests models’ ability to combine **structure + semantics**.
> >
> > - **Model coverage & evaluation protocol.**
> >   GraphWalks primarily evaluates GPT‑series. We benchmark **multiple model families and sizes** (Qwen, GPT, LLaMA, Gemini; from ≈8B to >100B), and for each instance we sample **k** solutions and report **pass@k, cons@k, z‑score** rather than only single‑shot accuracy. This lets us study robustness and self‑consistency, not just raw success.
> >
> > - **Key findings & analysis.**
> >   While GraphWalks shows that performance drops with longer contexts, our broader task set and process‑level analysis further reveal **three concrete bottlenecks** in LRMs’ long‑context graph reasoning: **coarse step‑by‑step execution, weak memory, and excessive redundancy (over‑thinking).**
> >
> > A short side‑by‑side summary is:
> >
> > | Aspect              | GraphWalks                         | GraAlgoBench                                             |
> > |---------------------|------------------------------------|----------------------------------------------------------|
> > | Tasks             | 1                 | 9                                        |
> > | Complexity          | Up to O(n) walk reasoning         | Linear‑time → NP‑hard                                |
> > | Task taxonomy       | None                              | Enumeration / Exploration / Intuition                |
> > | Graph source        | Synthetic                          | Real‑world                              |
> > | Models              | Mainly GPT‑5                      | Qwen, GPT, LLaMA, Gemini, 8B–100B+                   |
> > | Metrics             | Accuracy                          | pass@k, cons@k, z‑score                              |
> > | Main finding        | Accuracy drops with length        | Length drop + over-thinking        |

---

> > > ### Author Response · Authors · 2025-11-19
> > > **Rebuttal by Authors**
> > >
> > > ### W3. Tool-use / code execution: can GRALGOBENCH evaluate code-writing, and how would results differ?
> > >
> > > > W3: ...the paper should discuss how results might differ under code-execution assistance and whether GRALGOBENCH can evaluate that regime.
> > >
> > > You raise a good point about an effective way for models to solve graph problems. We address your two concrete questions below.
> > >
> > > **(1) Can GRALGOBENCH evaluate LRMs’ code-writing ability? — Yes.**
> > >
> > > GRALGOBENCH can be used directly in a **code-generation + execution** setting by slightly changing the prompt. For example, we can instruct the model:
> > >
> > > > Solve this problem using Python code only. At the end of the code, present your final answer in the following format: `\boxed{n}`, where `n` is the answer to the problem. Wrap your Python code in the following format:
> > > ```python
> > > ### Your code here
> > > print("\\boxed{n}")
> > > ```
> > >
> > > Under this instruction, the model writes Python that parses the graph from text, computes the answer, and prints it in a standardized format. We then execute the code and evaluate accuracy exactly as in the text‑only setting. Thus, GRALGOBENCH can **also** serve as a benchmark for code‑writing + tool‑use.
> > >
> > > **(2) How do results differ under code-execution assistance, and why focus on text-only?**
> > >
> > > In an additional experiment on three tasks (Triangle, DistanceK, MST) and three models (Qwen3‑32B, Gemini‑2.5‑pro, GPT5‑mini), allowing code execution **consistently and noticeably improves** p@k / c@k, especially for difficult tasks. Results are shown as follows:
> > >
> > > **Level6**
> > >
> > > | Task       | Qwen3‑32B      | Qwen3‑32B‑Code | Gemini‑2.5‑pro  | Gemini‑2.5‑pro‑Code | GPT5‑mini      | GPT5‑mini‑Code   |
> > > |-----------|----------------|----------------|-----------------|---------------------|----------------|------------------|
> > > |           | p@k / c@k      | p@k / c@k      | p@k / c@k       | p@k / c@k           | p@k / c@k      | p@k / c@k        |
> > > | Triangle  | 60.0/42.2    | 92.0 / 88.0 | 86.7/83.3    | 90.0 / 88.0     | 86.7/70.0    | 99.0 / 99.0  |
> > > | DistanceK |36.0/10.0     | 96.7 / 96.7 |    43.3/33.3   | 100 / 100           | 26.7/13.3      | 100 / 100        |
> > > | MST       |   0/0 | 88.0 / 84.0 |  0/0   | 92.0 / 90.0    |   0/0   | 100 / 100        |
> > >
> > > **Level5**
> > >
> > > | Task       | Qwen3‑32B      | Qwen3‑32B‑Code | Gemini‑2.5‑pro  | Gemini‑2.5‑pro‑Code | GPT5‑mini      | GPT5‑mini‑Code   |
> > > |-----------|----------------|----------------|-----------------|---------------------|----------------|------------------|
> > > |           | p@k / c@k      | p@k / c@k      | p@k / c@k       | p@k / c@k           | p@k / c@k      | p@k / c@k        |
> > > | Triangle  |  76.0/66.6   | 96.0 / 93.3 | 90.0 / 80.0     | 94.0 / 92.0     | 56.7 / 50.0    | 99.0 / 99.0  |
> > > | DistanceK |  40.0/20.0  | 98.0 / 96.0 | 43.3 / 30.0       | 100 / 100           | 60.0 / 53.3      | 100 / 100        |
> > > | MST       |  0/0  | 97.3 / 94.7 | 16.6 / 3.3     | 98.7 / 98.0     | 3.3 / 0      | 100 / 100        |
> > >
> > >
> > >
> > > This confirms that many of our tasks become much easier once models can offload graph traversal and bookkeeping to Python.
> > >
> > > However, the main goal of this paper is to isolate **intrinsic long‑context graph reasoning**, rather than the ability to call an external executor. Tool‑use essentially shifts the core difficulty (multi‑step traversal and state maintenance) into the tool, so performance no longer directly reflects the model’s own reasoning. For this reason, our primary results are reported in the **text‑only** setting, while we treat the code‑execution mode as a complementary usage of GRALGOBENCH.

---

> ### Author Response · Authors · 2025-11-19
> **Rebuttal by Authors**
>
> ### W4 and Q5: Conclusion Generalization beyond Graphs
> > W4: ...it’s unclear whether conclusions about long-context brittleness and over-thinking generalize beyond graph-structured problems...
>
> > Q5: ...Can you provide transfer evidence that performance and observed phenomena on this benchmark relate to broader “general reasoning” tasks?...
>
> We thank the reviewer for this insightful question, which directly addresses the core motivation of our work: exposing the limitations of current LRMs. We agree that for our conclusions to be impactful, they must generalize beyond graph-specific tasks. To demonstrate the broader relevance of our findings, we have conducted the following analyses as suggested.
>
> #### **1. Model-wise Correlation with External Benchmarks**
>
> To validate that performance on **GraAlgobench** correlates with broader reasoning capabilities, we analyzed the performance of six models against their scores on established benchmarks for general and long-context reasoning: **LongBench-v2**, **GPQA**, and **MMLU-Pro**.
>
> **Model Performance Comparison:**
>
> | Model | GraAlgobench | LongBench-v2 | GPQA | MMLU-Pro |
> | :--- | :--- | :--- | :--- | :--- |
> | Qwen3-235B | 63.0 | 60.6 | 79.0 | 84.3 |
> | Gemini-2.5-pro | 55.5 | 63.3 | 84.4 | 86.2 |
> | QWQ-32B | 55.0 | 48.9 | 59.3 | 76.4 |
> | Deepseek-R1 | 50.0 | 58.3 | 70.8 | 84.4 |
> | GPT-OSS-120B | 41.5 | - | 67.8 | 77.5 |
> | Distill-Qwen-32B | 39.0 | - | 48.4 | 73.9 |
>
> **Correlation Analysis:**
>
> We calculated the Spearman's rank correlation coefficient (ρ) between GraAlgobench scores and the scores on other benchmarks.
>
> | | LongBench-v2 (N=4) | GPQA (N=6) | MMLU-Pro (N=6) |
> | :--- | :--- | :--- | :--- |
> | **GraAlgobench (Spearman's ρ)** | 0.60 | 0.77 | 0.60 |
>
> **Conclusion:** According to standard empirical guidelines for interpreting correlation coefficients [3], values in the range of 0.50–0.79 are typically regarded as indicating a moderate to strong association. Accordingly, the observed correlations between GraAlgobench and GPQA (ρ = 0.77) and between GraAlgobench and LongBench‑v2 (ρ = 0.60) suggest that the **complex reasoning skills required to succeed on our benchmark are closely aligned with those needed for high‑level general reasoning and long‑context understanding**.
>
> [3] Mukaka et al. **A Guide to Aappropriate Use of Correlation Coefficient in Medical Research**, in Malawi medical journal, 2012.
>
>
> #### **2. Ablation Study on Graph Representation**
>
> To test whether the observed failures stem from our specific choice of graph representation (natural language edge list) versus more general context-length effects, we conducted an ablation study. We evaluated the performance of **Qwen3-32B** on the **DistanceK** and **MST** tasks using four different input formats for the graph structure:
>
> *   **edge_list:** The format used in our paper (natural language description of edges with semantic node names).
> *   **adj_list:** An adjacency list representation.
> *   **markdown:** A Markdown table to represent the graph's edges.
> *   **Numeric_ID:** Same as `edge_list`, but with node names replaced by abstract numeric IDs.
>
> The results for `pass@k` / `cons@k` are as follows:
>
> **Level-1 (8-15 nodes)**
> | Task | edge_list | adj_list | markdown | Numeric_ID |
> | :--- | :--- | :--- | :--- | :--- |
> | | (p@k/c@k) | (p@k/c@k) | (p@k/c@k) | (p@k/c@k) |
> | **DistanceK** | 100/100 | 100/100 | 100/100 | 100/100 |
> | **MST** | 100/100 | 100/100 | 100/100 | 100/100 |
>
> **Level-2 (16-30 nodes)**
> | Task | edge_list | adj_list | markdown | Numeric_ID |
> | :--- | :--- | :--- | :--- | :--- |
> | | (p@k/c@k) | (p@k/c@k) | (p@k/c@k) | (p@k/c@k) |
> | **DistanceK** | 100/83.33 | 100/96.67 | 96.67/86.67 | 100/90.00 |
> | **MST** | 96.67/83.33 | 100/86.67 | 93.33/83.33 | 96.67/83.33 |
>
> **Level-3 (31-50 nodes)**
> | Task | edge_list | adj_list | markdown | Numeric_ID |
> | :--- | :--- | :--- | :--- | :--- |
> | | (p@k/c@k) | (p@k/c@k) | (p@k/c@k) | (p@k/c@k) |
> | **DistanceK** | 93.33/66.67 | 100/100 | 86.67/76.67 | 96.67/83.33 |
> | **MST** | 23.33/13.33 | 23.33/3.33 | 53.33/13.33 | 53.33/26.67 |
>
> **Conclusion:** The results show that across different graph sizes, the model exhibits **comparable performance under all four representation formats**. This consistency indicates that the choice of graph representation (edge list, adjacency list, Markdown table, or numeric IDs) has little effect on performance, suggesting that the observed behaviors are **independent of input format** and mainly reflect the model’s intrinsic reasoning capability.
>
> We have updated these experiments in the **Discussion** section of our paper. Thanks for your valuable suggestion again!

---

> > ### Author Response · Authors · 2025-11-19
> > **Rebuttal by Authors**
> >
> > ### Q2. How are node names expanded and where is the extra text placed in the prompt
> > > Q2: ... how are node names expanded and how is the added text distributed across the prompt...
> >
> > This question touches on an important implementation detail, and clarifying it is helpful for reproducibility. In our experiments, we increase context length by **expanding node names with longer per‑node descriptors**, rather than adding a long preamble.
> >
> > Concretely:
> >
> > - The **graph structure, task description, and query format are kept fixed** across verbosity conditions.
> > - We only change the **length of the node identifiers** (e.g., `Street_xxx...`), so that the extra tokens are distributed **uniformly over all node mentions** in the graph list and in any subsequent references.
> > - There is **no added explanatory preamble** or additional natural language; the added text consists purely of extra characters in the node names.
> >
> > Below are illustrative snippets at different target context lengths (for the same MKC-style street-network task):
> >
> > **Length 4K**
> >
> > ```text
> > You are required to find a maximum k-core (k = 3) in an undirected street network. In this network, nodes represent streets and edges represent intersections. Maximum 3-core is the largest subgraph that every node has at least 3 neighbors.
> > Streets in the network: Street_kuuCYID, Street_nIUgJCk, Street_WYWNpuk, ...
> > ```
> >
> > **Length 8K**
> >
> > ```text
> > ...
> > Streets in the network: Street_vKmcXtFBRGTKpbOEoDGXjw, Street_LQSZNHIvOvSqeEvwvChQVo, Street_AKQfNdzNItnVSIBCtinGXy, ...
> > ```
> >
> > **Length 16K**
> >
> > ```text
> > ...
> > Streets in the network: Street_mcqGEwVwthpOriGswTQGlkjCznbILBvbRgQxSReqjw, Street_zILUoNDnQiLQyLtyCsSQcDkphywPVdnqXQbzewxgyx, Street_DcLFOJlOaaxdKciafmAAlIqiWAVdGidEKWOVODrehD, ...
> > ```
> >
> > **Length 32K**
> >
> > ```text
> > ...
> > Streets in the network: Street_jgKPEMWtjePMmZYsauAdfBomnrRPjhIrmUukYdpsjpAFMxtkAodnwkijTmOqfCzNsOSIlvPbPLQNXFWJSOYYxnfpmHyjxV, ...
> > ```
> >
> > **Length 64K**
> >
> > ```text
> > ...
> > Streets in the network: Street_flcpWZBkimekEujAbMxzmaltkgjmaisNonhCeIhjeEJFgekhUkRMPYypbMHrcatnNmatRMFUrEFDzFckrdBmsaTAyGZRhomcLCqtwTKdvjAuzvgMZRTIbKXjfqELvollXQSPKxwPOUEnPDJMdQpgnAJGxURynziuEIvHCPjuOIOcBzpUqAgRMCGHIrmTZlEIPXk, ...
> > ```

---

> ### Author Response · Authors · 2025-11-19
> **Rebuttal by Authors**
>
> ### Q3:  On token-budget caps and over-thinking’s cost–benefit
> > Q3: Have you tried a token-budget cap (e.g., limiting self-verification tokens) to quantify over-thinking’s cost-benefit?...
>
> You raise a promising angle for intervening on over‑thinking: explicitly limiting the token budget for self‑verification to quantify its cost–benefit. Due to time and resource constraints, we did not explore training‑time interventions, but we did run a small **prompt‑ and decoding‑level** intervention study along the lines you suggest.
>
> **(1) Intervention design**
>
> We consider two kinds of test‑time controls:
>
> 1. **Prompting away self‑verification (Confident).**
>    We add a global instruction discouraging self‑doubt and redundant checks:
>    > “You are an efficient graph algorithm expert. … Please state your core reasoning steps directly and concisely, then provide the final answer. Avoid unnecessary repetition, self‑doubt, or redundant checks of your previous steps.”
>
> 2. **Explicit token caps for self‑reflection (reflect\_N).**
>    We keep the above instruction and additionally bound the allowed self‑reflection budget:
>    > “You can use at most {token limitation} tokens to conduct self‑reflection behaviour (e.g., checking your previous steps, verifying your answer, etc.) when you solve the problem.”
>
>    where `{token limitation} ∈ {1024, 2048, 4096}`.
>
> 3. **Decoding‑level output caps.**
>    We directly cap the model’s **max output tokens** at {8k, 16k, 32k}.
>
> We apply these interventions to **Qwen3‑32B** on two tasks that naturally exhibit high self‑verification rates in our main experiments: **MKC** and **MaxDegree**, at **level‑3** and **level‑4** difficulty, and report **pass@k / cons@k**.
>
> **(2) Results**
>
> _Level‑3_
>
> | Setting        | MKC p@k | MKC c@k | MaxDegree p@k | MaxDegree c@k |
> |---------------|---------|---------|----------------|----------------|
> | Confident     | **70.00** | 36.67   | **100.00**       | **83.00**          |
> | reflect\_1024 | 46.67   | 20.00 | 90.00          | 23.33          |
> | reflect\_2048 | 53.33   | 33.33   | **100.00**       | 66.67          |
> | reflect\_4096 | 60.00   | 26.67   | 96.67          | 76.67          |
> | 8k tokens     | 3.33    | 0.00    | 66.67          | 3.33           |
> | 16k tokens    | 66.67   | 26.67   | 96.67          | 66.67          |
> | 32k tokens (Original Setting) | 63.33   | **40.00**   | **100.00**       | 73.00          |
>
> _Level‑4_
>
> | Setting        | MKC p@k | MKC c@k | MaxDegree p@k | MaxDegree c@k |
> |---------------|---------|---------|----------------|----------------|
> | Confident     | 6.67    | 3.33    | **90.00**       | **50.00**          |
> | reflect\_1024 | 10.00   | 6.67    | 76.67          | 40.00          |
> | reflect\_2048 | 26.67 | **10.00** | 73.33          | 40.00          |
> | reflect\_4096 | 16.67   | 3.33    | 76.67          | 3.33           |
> | 8k tokens     | 0.00    | 0.00    | 80.00          | 33.33          |
> | 16k tokens    | **30.00** | 6.67    | 76.67          | 43.33          |
> | 32k tokens (Original Setting) | **30.00** | **10.00** | 83.00       | 40.00          |
>
>
>
>
> **(3) Takeaways**
>
> Overall, simple prompt‑level or decoding‑level token caps on self‑verification **do not reliably improve performance** in our setting: they sometimes help slightly, sometimes hurt. This suggests that **controlling over‑thinking purely via prompts or output‑length limits is not sufficient on its own to yield consistent accuracy gains on GraAlgoBench**.
>
> ---
> ### Q4: Figure Adjustment
> > Q4: >In Figure 3, several curves appear to run at or below zero. Maybe tightening the y-axis limits and/or adding clipping can avoid visual ambiguity.
>
> We appreciate the suggestion regarding Figure 3. In the revised version, we have tightened the y‑axis limits and applied clipping where appropriate so that no curve appears to run at or below zero, eliminating the visual ambiguity you pointed out.

---

> ### Author Response · Authors · 2025-11-27
> **Kind Reminder for Rebuttal**
>
> Dear Reviewer dfQZ,
>
> As the rebuttal period draws to a close, we are very interested in hearing any further feedback you might have and would be pleased to clarify or address any remaining concerns. Please do not hesitate to let us know if you have additional comments or questions.
>
> Thank you again for your time and thoughtful review.
>
> Best regards,
> Authors of Paper  8673

---

### Official Review · Reviewer_8MVa · 2025-11-01

**Soundness:** 3
**Presentation:** 4
**Contribution:** 3
**Rating:** 8
**Confidence:** 4

**Summary:**

This work introduces a new benchmark for evaluating LLM reasoning capabilities under long-context scenarios and conducts careful error analysis to study the underlying mechanism of reasoning failure in this setting. The paper's methodology involves building prompts with controllable length that each contain a problem as an instance of one of 9 algorithmic graph tasks (maximum degree node, maximum clique problem, path sum, etc.) plus a graph instance from real-world graph datasets, alongside an interesting taxonomy to categorize the required reasoning into enumeration, exploration, and intuition labels. The length is controlled either by graph size (number of nodes) or by using fixed graphs with varying node names as an alternative way of controlling the input prompt length.

By careful evaluation of their introduced dataset, alongside error analysis on generated responses, this work reveals some key findings. Both increasing the number of nodes and node name token length resulted in degraded performance consistently across a range of frontier open- and closed-source reasoning models. The error analysis contains a taxonomy of common issues found in incorrect model responses, and it identifies algorithmic execution error, graph memorization error, and excessive redundancies as the key failure modes. Moreover, their error analysis shows that intuition-driven reasoning (the greedy subset of the prompts) degrades the most as input context length grows.

To fully understand the excessive redundancies issue, the paper designs an experiment to segment the output response based on “aha moment” signals (slicing the response after finding high-entropy tokens such as “wait”, “but”, “so”). Then, it classifies these segments, concluding that most of these segments are unsuccessful attempts at self-verification that fail to detect the error in prior segments. Hence, the paper argues that unsuccessful self-verification is the main driver of the "overthinking" (the observed redundancy in reasoning traces).

**Strengths:**

The paper is well-written and clear, and the experiments are designed carefully (e.g., a diverse selection of open- and closed-source frontier models, a strong qualitative study, and error analysis). Moreover, the code is available, which aids in the reproducibility of the work.

* **Task design:** The reasoning taxonomy is well-designed and well-motivated, categorizing reasoning types into enumeration, exploration, and intuition, which appears to be a strong differentiator compared to prior work like [1]. More importantly, the usage of an LLM-as-judge to classify the reasoning traces of models for an initial 100 graph samples is a clever design decision to assign each task to one of the reasoning labels.
* **The evaluation error analysis reveals some impactful findings:**
    1.  Quantifying the reasoning failure modes under long-context settings, and attributing them mainly to Algorithm Execution Errors (AEE), Graph Memorization Errors (GME), and excessive redundancy is a strong contribution.
    2.  Identifying high-entropy tokens and using them to segment reasoning traces for detecting reasoning shifts is an interesting methodology. The key findings of this experiment are that self-verification fails most of the time to progress smoothly past errors and is highly redundant. This finding is impactful as it reveals one possible underlying failure mechanism of reasoning models, especially under highly long-context inputs.
* While the data collection and controlled-length problem synthesis share similarities with GraphArena[1] (e.g., sharing tasks and the methodology to sample real-world graphs), and while a key finding (that increasing input length leads to performance degradation) has also been studied in [1, 2], this paper provides a key differentiation. GraphArena[1] has a very limited focus on dedicated reasoning models and only considers input graph node counts up to 50. This paper correctly indicates this is a setting where good reasoning models have already reached near-saturated performance; hence, its usage of **up to 160 nodes** is well-designed to study the limits of frontier reasoning models.
* Using such graph problems as evaluation benchmarks is an impactful direction and is aligned with frontier labs' evaluations on long-context reasoning (e.g., OpenAI GraphWalks).

**Weaknesses:**

* One minor weakness is that the **fixed assignments of Enumeration, Exploration, and Intuition reasoning labels** to tasks might be slightly misleading, as models might sometimes use different reasoning to solve a task (e.g., doing Exploration instead of Intuition). While Appendix H.1 does a great job showing this issue would be rare, it also shows such an issue exists. Furthermore, these ablations are done by using an LLM-as-judge on 100 problem instances that seem to differ from the wide range of task complexity used in the final proposed dataset. Hence, the algorithm ratio as defined in Appendix H.1 might show a more uniform distribution in the actual dataset, especially for weaker and smaller open-source models.
* While evaluating the long-context input scenario for reasoning problems is an important benchmark, the proposed evaluation seems to **slightly diverge from the realistic use case of frontier models**. It seems that all tasks in this evaluation could be handled much better with tool calls (writing code and receiving code execution results). Having extremely sophisticated short-context inputs that require a large amount of reasoning token generation (like olympiad competitions) is an alternative way of measuring long-context reasoning while keeping the input tokens fixed. Thus, I would suggest a discussion on this matter and perhaps a refinement of the introductory claim (lines 47-49) that "First, they lack long-context evaluation: existing benchmarks predominantly use short problem texts, which cannot be easily scaled to assess LRMs’ reasoning capabilities over extended contexts."
* As mentioned in this paper, as input context length grows, the performance of models degrades noticeably in correctly solving the graph problems. However, in the evaluation workflow, there are multiple LLM-as-judge calls using Qwen-2.5-72B (such as prompt I.4 for error analysis and prompt I.5 for reformatting) that seem to need to operate and reason over a potentially much larger input context compared to the original graph-solving task. Hence, **the quality of the output responses from Qwen-2.5-72B when using prompts I.4 and I.5 on an already large, long-context input problem is questionable.**
* More discussion and illustration are needed on **how high-entropy tokens are being used for segmentation and what their failure modes are**. How likely is it that the main finding—that “ineffective self-verification is a primary driver of over-thinking in LRMs”—is the result of incorrect segmentation?

**Questions:**

1.  How much can we trust the validity of Qwen-2.5-72B's responses for prompts I.4 and I.5, especially since they are also given a very long context input (possibly even larger than the input for the graph-solving problems)?
2. Could you provide an argument for the real-world use cases of this benchmark, especially in light of the fact that these graph problems could be solved with tool calls?
3.  Could you elaborate more on the segmentation using high-entropy tokens? How is the threshold for detecting high-entropy tokens tuned?




**References:**

[1] Tang, Jianheng, et al. "Grapharena: Evaluating and exploring large language models on graph computation." arXiv preprint arXiv:2407.00379 (2024).

[2] Xu, Hao, et al. "GraphOmni: A Comprehensive and Extendable Benchmark Framework for Large Language Models on Graph-theoretic Tasks." arXiv preprint arXiv:2504.12764 (2025).

---

> ### Author Response · Authors · 2025-11-19
> **Rebuttal by Authors**
>
> We sincerely appreciate the reviewer’s valuable suggestions! Our responses will focus on the following aspects:（1）Robustness of Algorithm Ratios and Validation of Dominant Reasoning Paradigms.（2）On tool-based evaluation vs. long-context reasoning without tools. (3) On the reliability of LLM‑as‑judge under long‑context inputs. (4) On segmentation using high‑entropy tokens and its reliability. (5) Without‑Tool LRMs: The Real‑World Significance of Long CoT Reasoning on Graph Problems.
>
> ### **W1: Robustness of Algorithm Ratios and Validation of Dominant Reasoning Paradigms**
>
> > W1: ...While Appendix H.1 does a great job showing this issue would be rare, it also shows such an issue exists. Furthermore, these ablations are done by using an LLM-as-judge on 100 problem instances that seem to differ from the wide range of task complexity used in the final proposed dataset...
>
> We thank the reviewer for emphasizing this methodological aspect. We acknowledge that using only 100 judged samples could potentially introduce sampling bias or model‑specific influence. To evaluate the robustness of our **algorithm ratios** and the validity of the **dominant reasoning paradigm per task**, we performed a more comprehensive analysis that (1) expands data coverage, (2) includes multiple judge and responder models, and (3) integrates human validation.
>
>
>
> #### **1. Expanded algorithm‑ratio evaluation on the full benchmark**
>
> To reduce sampling variance and capture the true distribution of reasoning strategies, we recomputed algorithm ratios on the **entire test split** of our benchmark:
>
> - **9 tasks**, each containing **300 problems**.
> - **4 reasoning traces per problem** generated by each responder LRM.
> - **Total per model: 9 × 300 × 4 = 10,800 traces.**
>
> This design ensures that the computed ratios comprehensively reflect the full benchmark scale and problem diversity.
>
>
>
> #### **2. Multiple judge models and responder LRMs**
>
> To control for bias from a single judge and to assess consistency across model capabilities, we used:
>
> - **Two extra judge models:** *DeepSeek‑V3* and *GPT‑4.1*.
> - **Three responder LRMs:** *Qwen3‑8B* (small), *QWQ‑32B* (medium), and *Gemini‑2.5‑pro* (strong).
>
> This combination allows us to test whether the reasoning‑type taxonomy remains stable under changes in both judging and responding models.
>
>
> #### **3. Human validation**
>
> To align with human interpretation and assess the clarity of reasoning‑type distinctions, we recruited **three postgraduate judges** with competitive‑programming backgrounds. Each human judge independently annotated **100 randomly sampled traces per task**, using the same four‑way scheme: **Enumeration (en)** / **Exploration (ex)** / **Intuition (in)** / **Not mentioned (nm)**. All three judges labeled each trace independently, and any disagreements were resolved through discussion to determine the final label.
>
> The aggregated distributions from human and model judges are shown below:

---

> > ### Author Response · Authors · 2025-11-19
> > **Rebuttal by Authors**
> >
> > #### **Qwen3‑8B**
> >
> > | Model / Judge | PathSum | Distancek | Diameter | Degree | Triangle | MCP | MKC | MST | DistanceThreshold |
> > |----------------|-----------|------------|-----------|-----------|-----------|-----------|-----------|-----------|----------------|
> > |  | en/ex/in/nm | en/ex/in/nm | en/ex/in/nm | en/ex/in/nm | en/ex/in/nm | en/ex/in/nm | en/ex/in/nm | en/ex/in/nm | en/ex/in/nm |
> > | **DeepSeek‑V3** | 0.0/100.0/0.0/0.0 | 1.3/89.6/4.6/4.5 | 0.0/98.8/0.0/1.2 | 90.8/0.0/0.0/9.2 | 83.8/8.8/1.7/5.7 | 90.8/0.0/0.0/9.2 | 0.0/0.0/98.3/1.7 | 0.0/0.0/100.0/0.0 | 14.6/0.0/85.0/0.4 |
> > | **GPT‑4.1** | 0.0/100.0/0.0/0.0 | 0.3/99.7/0.0/0.0 | 14.0/81.0/0.0/5.0 | 100.0/0.0/0.0/0.0 | 88.8/0.0/11.2/0.0 | 87.3/0.0/10.7/2.0 | 0.0/0.0/100.0/0.0 | 0.0/0.0/99.0/1.0 | 11.8/0.0/81.7/6.5 |
> > | **Human** | 0.0/100.0/0.0/0.0 | 7.0/93.0/0.0/0.0 |11.0/89.0/0.0/0.0 | 100.0/0.0/0.0/0.0 | 84.0/9.0/7.0/0.0 | 88.0/0.0/12.0/0.0 | 0.0/0.0/100.0/0.0 | 0.0/0.0/100.0/0.0 | 18.0/0.0/82.0/0.0 |
> >
> >
> >
> > #### **QWQ‑32B**
> >
> > | Model / Judge | PathSum | Distancek | Diameter | Degree | Triangle | MCP | MKC | MST | DistanceThreshold |
> > |----------------|-----------|------------|-----------|-----------|-----------|-----------|-----------|-----------|----------------|
> > | | en/ex/in/nm | en/ex/in/nm | en/ex/in/nm | en/ex/in/nm | en/ex/in/nm | en/ex/in/nm | en/ex/in/nm | en/ex/in/nm | en/ex/in/nm |
> > | **DeepSeek‑V3** | 0.0/100.0/0.0/0.0 | 0.0/96.3/2.5/1.2 | 0.0/99.6/0.0/0.4 | 100.0/0.0/0.0/0.0 | 73.8/14.7/0.8/10.7 | 86.7/0.0/0.8/12.5 | 0.0/0.0/99.2/0.8 | 0.0/0.0/100.0/0.0 | 16.2/5.8/83.8/0.0 |
> > | **GPT‑4.1** | 0.0/100.0/0.0/0.0 | 0.0/100.0/0.0/0.0 | 12.0/88.0/0.0/0.0 | 100.0/0.0/0.0/0.0 | 76.5/0.0/23.5/0.0 | 86.8/0.0/2.0/11.2 | 0.0/0.0/100.0/0.0 | 0.0/0.0/100.0/0.0 | 10.8/0.0/88.5/0.7 |
> > | **Human** | 0.0/100.0/0.0/0.0 | 0.0/100.0/0.0/0.0 | 7.0/93.0/0.0/0.0 | 100.0/0.0/0.0/0.0 | 77.0/10.0/13.0/0.0 | 87.0/0.0/13.0/0.0 | 0.0/0.0/100.0/0.0 | 0.0/0.0/100.0/0.0 | 12.0/0.0/88.0/0.0 |
> >
> >
> >
> > #### **Gemini‑2.5‑Pro**
> >
> > | Model / Judge | PathSum | Distancek | Diameter | Degree | Triangle | MCP | MKC | MST | DistanceThreshold |
> > |----------------|-----------|------------|-----------|-----------|-----------|-----------|-----------|-----------|----------------|
> > | | en/ex/in/nm | en/ex/in/nm | en/ex/in/nm | en/ex/in/nm | en/ex/in/nm | en/ex/in/nm | en/ex/in/nm | en/ex/in/nm | en/ex/in/nm |
> > | **DeepSeek‑V3** | 0.0/100.0/0.0/0.0 | 5.3/92.4/2.2/0.1 | 2.0/97.8/0.0/0.2 | 100.0/0.0/0.0/0.0 | 86.7/13.3/0.0/0.0 | 91.3/0.0/8.4/0.3 | 0.0/0.0/100.0/0.0 | 0.0/0.0/100.0/0.0 | 4.7/0.0/91.0/4.3 |
> > | **GPT‑4.1** | 0.0/100.0/0.0/0.0 | 0.0/100.0/0.0/0.0 | 0.0/100.0/0.0/0.0 | 100.0/0.0/0.0/0.0 | 82.3/8.7/9.0/0.0 | 90.4/0.0/5.0/4.6 | 0.0/0.0/100.0/0.0 | 0.0/0.0/100.0/0.0 | 2.0/0.0/93.7/4.3 |
> > | **Human** | 0.0/100.0/0.0/0.0 | 5.0/95.0/0.0/0.0 | 0.0/100.0/0.0/0.0 | 100.0/0.0/0.0/0.0 | 84.0/10.0/6.0/0.0 | 88.0/0.0/12.0/0.0 | 0.0/0.0/100.0/0.0 | 0.0/0.0/100.0/0.0 | 3.0/0.0/97.0/0.0 |
> >
> > #### **Summary of Findings**
> >
> > Across **judge models**, **human evaluators**, and **responder LRMs**, the dominant reasoning paradigm for each task is consistently preserved:
> >
> > - **PathSum / Distancek / Diameter → Exploration**
> > - **Degree / Triangle / MCP → Enumeration**
> > - **MKC / MST / DistanceThreshold → Intuition**
> >
> > Minor deviations (e.g., mixed strategies in boundary cases) do not alter the dominant category for any task.  Hence, the **algorithm ratios and task‑level taxonomy remain stable** across data scales, model capacities, and judgment sources.

---

> ### Author Response · Authors · 2025-11-19
> **Rebuttal by Authors**
>
> ### W2. On tool-based evaluation vs. long-context reasoning without tools
> > ...It seems that all tasks in this evaluation could be handled much better with **tool calls** (writing code and receiving code execution results). Having extremely sophisticated **short-context inputs** that require a large amount of reasoning token generation (like olympiad competitions) is an alternative way of measuring long-context reasoning while keeping the input tokens fixed...
>
>
> We thank the reviewer for highlighting this important perspective and for suggesting tool-based evaluation (e.g., code writing plus execution) as a more realistic use case for frontier models.
>
> **(1) Empirical comparison: with-code vs. without-code on our benchmark**
>
> To better understand the impact of tool usage on our benchmark, we conducted an additional experiment where models are explicitly instructed to solve problems by writing code and then using a code-execution tool. Concretely, we compare three LRMs of different strengths (Qwen3‑32B, Gemini‑2.5‑pro, GPT5‑mini) on three representative tasks (Triangle, DistanceK, MST), at graph size level‑5 and level‑6. We report pass@k and cons@k (p@k / c@k) with and without tool calls.
>
> For brevity, we summarize the key results below (level‑5 & level‑6 combined):
>
> **Level6 (121-160 nodes)**
>
> | Task       | Qwen3‑32B      | Qwen3‑32B‑Code | Gemini‑2.5‑pro  | Gemini‑2.5‑pro‑Code | GPT5‑mini      | GPT5‑mini‑Code   |
> |-----------|----------------|----------------|-----------------|---------------------|----------------|------------------|
> |           | p@k / c@k      | p@k / c@k      | p@k / c@k       | p@k / c@k           | p@k / c@k      | p@k / c@k        |
> | Triangle  | 60.0/42.2    | 92.0 / 88.0 | 86.7/83.3    | 90.0 / 88.0     | 86.7/70.0    | 99.0 / 99.0  |
> | DistanceK |36.0/10.0     | 96.7 / 96.7 |    43.3/33.3   | 100 / 100           | 26.7/13.3      | 100 / 100        |
> | MST       |   0/0 | 88.0 / 84.0 |  0/0   | 92.0 / 90.0    |   0/0   | 100 / 100        |
>
> **Level5 (81-120 nodes)**
>
> | Task       | Qwen3‑32B      | Qwen3‑32B‑Code | Gemini‑2.5‑pro  | Gemini‑2.5‑pro‑Code | GPT5‑mini      | GPT5‑mini‑Code   |
> |-----------|----------------|----------------|-----------------|---------------------|----------------|------------------|
> |           | p@k / c@k      | p@k / c@k      | p@k / c@k       | p@k / c@k           | p@k / c@k      | p@k / c@k        |
> | Triangle  |  76.0/66.6   | 96.0 / 93.3 | 90.0 / 80.0     | 94.0 / 92.0     | 56.7 / 50.0    | 99.0 / 99.0  |
> | DistanceK |  40.0/20.0  | 98.0 / 96.0 | 43.3 / 30.0       | 100 / 100           | 60.0 / 53.3      | 100 / 100        |
> | MST       |  0/0  | 97.3 / 94.7 | 16.6 / 3.3     | 98.7 / 98.0     | 3.3 / 0      | 100 / 100        |
>
>
> Across models and tasks, enabling code+tool usage **consistently and substantially improves performance**, especially for difficult tasks. This directly supports the reviewer’s intuition that our tasks “could be handled much better with tool calls.”
>
> **(2) Why we focus on reasoning *without* delegating the core difficulty to tools**
>
> That said, our primary motivation is to measure the **intrinsic reasoning ability of LRMs on graph‑structured problems**, rather than their ability to offload the hardest parts to external tools. In virtually all tasks considered here, the main difficulty lies in:
>
> - iteratively exploring or enumerating graph neighborhoods,
> - maintaining a long‑range “mental state” over the graph (graph memory), and
> - performing step‑by‑step execution of an algorithm over many hops.
>
> When a model is allowed to simply produce a short program and let an external executor handle traversal, state maintenance, and exact computation, it effectively **bypasses** these challenges. The resulting metric becomes a mixture of (i) high‑level problem understanding and (ii) basic code synthesis, but no longer reflects whether the model itself can carry out multi‑step, long‑horizon graph reasoning in its **own activations**.
>
> For this work, we therefore deliberately **disable tool calls** and require models to reason directly over long graph descriptions and multi‑step algorithmic procedures in natural language. This design isolates the reasoning capability we aim to study, and complements (rather than replaces) tool‑augmented evaluations.

---

> > ### Author Response · Authors · 2025-11-19
> > **Rebuttal by Authors**
> >
> > **(3) Clarifying what we mean by “long‑context reasoning”**
> >
> > We thank the reviewer for prompting us to refine our introductory claim on long‑context evaluation. The term *long‑context reasoning* can be interpreted in two ways: it may refer to tasks with either **long inputs** or **long outputs**.
> > In our work—and consistent with prior studies [1, 2, 3]—we adopt the **long‑input definition**, focusing on scenarios where models must reason over extended and information‑rich inputs such as large graphs or long chains of constraints. Although benchmarks like *OlympiadBench* involve short but highly challenging problems that can trigger long reasoning outputs, they represent the *long‑output* form of long‑context reasoning, which differs from our target setting.
> >
> > We have updated the introduction to explicitly refer to *“long‑context input”* (Line 45) and added a new **Discussion Section (Section 3.3)** to analyze the performance gains from allowing LRMs to write and execute code on graph problems, as well as to explain why we adopt *long‑CoT reasoning without tools* as our core experimental setting.
> >
> > [1] Liu et al. **A Comprehensive Survey on Long Context Language Modeling.** Arxiv Preprint 2025.
> >
> > [2] Das et al. **Needle in the Haystack for Memory Based Large Language Models**, in ICML2024.
> >
> > [3] Wu et al. **Longgenbench: Benchmarking Long-Form Generation in Long Context LLMs**. Arxiv Preprint 2024.

---

> ### Author Response · Authors · 2025-11-19
> **Rebuttal by Authors**
>
> ### W3/Q1: On the reliability of LLM‑as‑judge under long‑context inputs
> > W3: the quality of the output responses from Qwen-2.5-72B when using prompts I.4 and I.5 on an already large, long-context input problem is questionable.
>
> > Q1: How much can we trust the validity of Qwen-2.5-72B's responses for prompts I.4 and I.5,
>
>
> We thank the reviewer for raising this important concern about the robustness of our LLM‑as‑judge setup, especially given that the judging prompts (I.4 for segmentation and I.5 for error categorization) also operate on long contexts.
>
>
> #### **(a) Task difficulty: summarization / error tagging vs. solving graph problems**
>
> First, we emphasize that the tasks in prompts I.4 and I.5 are **text summarization and error tagging** over a *given* reasoning trace, not solving the underlying graph problem itself. Concretely:
>
> - Prompt I.4 asks the judge to segment a trace into coherent steps and briefly summarize each step.
> - Prompt I.5 asks the judge to assign high‑level error categories (e.g., redundancy, graph memory failure, execution error) to each segment.
>
> These are qualitatively **simpler** than the original graph‑reasoning tasks that require multi‑step algorithmic reasoning over large graphs. The judge does not need to execute or simulate the full graph algorithm; it only needs to (i) detect repetition, (ii) notice when the model loses track of prior statements (graph memory), and (iii) identify local execution mistakes. This type of meta‑analysis over text is precisely what LLM‑as‑judge setups are commonly used for, even on long inputs.
>
> Moreover, our workflow—first using **Qwen‑2.5‑72B** for segmentation, then using **O3‑mini** to assign error types—is directly inspired by **ProcessBench** [4]. Their experiments show that **O1‑mini** can reach an error‑classification accuracy of **93.2%** on GSM8K process supervision, and **O3‑mini** is a stronger, more recent model from the same family. Besies, [5] provides a detailed discussion on the feasibility of LLM‑as‑a‑judge. These studies give us strong prior evidence that such judge models are capable of reliably performing this kind of textual error analysis.
>
>
>
> #### **(b) Cross‑model and human validation of error analysis**
>
> To further test the robustness of our conclusions beyond a single judge, we re‑ran the error analysis on the reasoning traces of **Qwen3‑32B‑thinking** and **Qwen3‑32B‑no‑thinking**, using **three independent LLM judges** (*DeepSeek‑R1*, *GPT‑5*, *O3‑mini*) and an additional **human evaluation** set.
>
> For the human evaluation, three postgraduates with competitive‑programming experience annotated **150 reasoning segments** (50 per paradigm) with the same four labels: **Redundancy**, **Graph Memory**, **Execution Error**, **Others**.  In cases of labeling disagreement, the annotators discussed collectively to reach a consensus on the final label.
> The aggregated human results are shown alongside the model‑judge distributions below.
>
>
>
> ### **Qwen3‑32B‑thinking**
>
> #### *O3‑mini*
>
> | Paradigm | Redundancy | Graph Memory | Execution Error | Others |
> |-----------|-------------|---------------|----------------|--------|
> | Enumeration | 53.6 % | 17.1 % | 10.5 % | 18.8 % |
> | Exploration | 37.7 % | 21.6 % | 35.3 % | 5.4 % |
> | Intuition   | 24.6 % | 21.0 % | 48.6 % | 5.8 % |
>
> #### *DeepSeek‑R1*
>
> | Paradigm | Redundancy | Graph Memory | Execution Error | Others |
> |-----------|-------------|---------------|----------------|--------|
> | Enumeration | 56.1 % | 16.5 % | 12.1 % | 15.3 % |
> | Exploration | 40.4 % | 19.5 % | 33.9 % | 6.2 % |
> | Intuition   | 28.2 % | 18.9 % | 45.7 % | 7.2 % |
>
> #### *GPT‑5*
>
> | Paradigm | Redundancy | Graph Memory | Execution Error | Others |
> |-----------|-------------|---------------|----------------|--------|
> | Enumeration | 49.1 % | 15.8 % | 13.9 % | 21.2 % |
> | Exploration | 38.9 % | 19.6 % | 34.2 % | 7.3 % |
> | Intuition   | 29.4 % | 20.3 % | 45.1 % | 5.2 % |
>
> #### *Human*
>
> | Paradigm     | Redundancy | Graph Memory | Execution Error | Others |
> |---------------|-------------|---------------|----------------|---------|
> | Enumeration | 52.0 % | 16.0 % | 12.0 % | 20.0 % |
> | Exploration | 38.0 % | 20.0 % | 36.0 % | 6.0 % |
> | Intuition   | 26.0 % | 18.0 % | 46.0 % | 10.0 % |

---

> ### Author Response · Authors · 2025-11-19
> **Rebuttal by Authors**
>
> ### **Qwen3‑32B‑no‑thinking**
>
> #### *O3‑mini*
>
> | Paradigm | Redundancy | Graph Memory | Execution Error | Others |
> |-----------|-------------|---------------|----------------|--------|
> | Enumeration | 16.1 % | 42.6 % | 28.9 % | 12.4 % |
> | Exploration | 5.4 %  | 38.7 % | 54.8 % | 1.1 % |
> | Intuition   | 11.3 % | 27.2 % | 58.4 % | 3.1 % |
>
> #### *DeepSeek‑R1*
>
> | Paradigm | Redundancy | Graph Memory | Execution Error | Others |
> |-----------|-------------|---------------|----------------|--------|
> | Enumeration | 17.1 % | 39.4 % | 29.2 % | 14.3 % |
> | Exploration | 6.5 %  | 39.8 % | 51.2 % | 2.5 % |
> | Intuition   | 12.3 % | 28.5 % | 55.4 % | 3.8 % |
>
> #### *GPT‑5*
>
> | Paradigm | Redundancy | Graph Memory | Execution Error | Others |
> |-----------|-------------|---------------|----------------|--------|
> | Enumeration | 14.9 % | 40.4 % | 30.1 % | 14.6 % |
> | Exploration | 6.6 %  | 36.9 % | 53.1 % | 3.4 % |
> | Intuition   | 10.8 % | 29.6 % | 55.5 % | 4.1 % |
>
> #### *Human*
>
> | Paradigm     | Redundancy | Graph Memory | Execution Error | Others |
> |---------------|-------------|---------------|----------------|---------|
> | Enumeration | 18.0 % | 36.0 % | 32.0 % | 14.0 % |
> | Exploration | 8.0 %  | 42.0 % | 48.0 % | 2.0 % |
> | Intuition  | 8.0 % | 28.0 % | 60.0 % | 4.0 % |
>
>
> Across all judges—including human evaluation—the **qualitative patterns remain consistent**:
>
> - For **thinking** traces, the main issues are **redundancy**, **execution inaccuracies**, and moderate **graph‑memory slips**.
> - For **no‑thinking** traces, **execution** and **graph‑memory errors** dominate, while redundancy decreases.
>
> These jointly reinforce that the LLM‑as‑judge process captures valid and reproducible error trends across evaluation sources.
>
>
>
> [4] Zheng et al. **ProcessBench: Identifying process errors in mathematical reasoning**, in ACL 2025.
>
> [5] Gu et al. **A Survey on LLM-as-a-Judge**. ArXiv Preprint 2024.

---

> ### Author Response · Authors · 2025-11-19
> **Rebuttal by Authors**
>
> ### **W3/Q3: On segmentation using high‑entropy tokens and its reliability**
>
> >More discussion and illustration are needed on how high-entropy tokens are being used for segmentation and what their failure modes are. How likely is it that the main finding— that “ineffective self-verification is a primary driver of over-thinking in LRMs”— is the result of incorrect segmentation?
>
> >Could you elaborate more on the segmentation using high-entropy tokens? How is the threshold for detecting high-entropy tokens tuned?
>
>
>
>
> We thank the reviewer for this constructive question regarding the **high‑entropy‑token‑based segmentation** and the possibility that our findings on “ineffective self‑verification” might be affected by segmentation errors. Below, we provide additional detail and clarification.
>
>
>
> #### **(1) High‑entropy token computation and segmentation design**
>
> During model inference, we record the model’s **top‑20 predicted next‑token probabilities** at each token generation step and compute the corresponding **token‑level entropy** (see Appendix F.3),  which measures the dispersion of these probabilities—higher entropy means the model’s confidence is spread across many candidate tokens, while lower entropy indicates concentration on a single likely token.
>
> By averaging entropy values across the test set, we find that **“wait”**, **“but”**, and **“so”** consistently occur in **high‑uncertainty contexts**. These words typically serve as discourse markers of **self‑checking** or **reasoning transitions**.
>
> Once identified, this **set of high‑entropy tokens remains fixed** for all analyses.
> Segmentation is thus **rule‑based and static**, with **no threshold tuning**—the tokens (“wait”, “but”, “so”) are used to divide reasoning traces throughout all experiments.
>
>
>
>
> #### **(2) Behavior annotation per segment**
>
> Although segments are obtained through these fixed high‑entropy tokens, we do **not directly interpret every segmentation** as a self‑verification step.
> After segmentation, each segment is further analyzed using our **LLM‑as‑judge** process (Prompt K.7), which classifies its **reasoning behavior type** into one of three categories: **Self‑verification**, **Strategy shift**, or **None**.
>
> This two‑stage design (rule‑based segmentation → LLM‑based classification) prevents misinterpreting purely connective tokens as genuine verification events. In effect, high‑entropy tokens serve only as *anchors* for dividing reasoning streams into discourse units, while the final labeling of the cognitive function of each unit is independently validated.
>
>
> #### **(3) Failure Modes**
>
> In some cases, a segment created by a high‑entropy token does not exhibit any identifiable reasoning behavior.  However, such cases are relatively rare—**as shown in Figure 6b, only about 10 % of all segments** fall into the *“no‑behavior”* category—and therefore have **limited impact** on our analysis.

---

> ### Author Response · Authors · 2025-11-19
> **Rebuttal by Authors**
>
> ### Q2: **Without‑Tool LRMs: The Real‑World Significance of Long CoT Reasoning on Graph Problems**
> > Q2:  Could you provide an argument for the real-world use cases of this benchmark, especially in light of the fact that these graph problems could be solved with tool calls?
>
> We appreciate this insightful question. Indeed, when tool use can precisely solve graph problems, it is natural to wonder whether having LRMs reason about graphs through long chains of thought (CoT) still has real‑world value.  However, we argue that evaluating LRMs via long CoT reasoning on graph problems remains scientifically and practically vital for three key reasons:
>
>
> **1. Handling "Semantic Graphs" with Soft Constraints (Beyond Hard Logic)**
>
> While tools (e.g., NetworkX) excel at executing rigid algorithms on structured data (e.g., adjacency matrices), real-world graphs are often unstructured and noisy.
> *   **Example:** In Anti-Money Laundering (AML), a standard connected-component algorithm might flag an entire community because a suspect bought an ice cream from a shop owner. This creates a "false positive" edge based on rigid topology.
> *   **LRM's Role:** An LRM acts as a semantic filter. It can reason about the *nature* of the relationship (e.g., "a $10 transaction is likely incidental, not complicit") *before* or *during* the traversal. This ability to dynamically define what constitutes a meaningful "edge" based on context is something hard-coded tools cannot easily replicate without complex, pre-defined heuristics.
>
> **2. Internalization Is Necessary for Planning in Dynamic‑Graph Settings**
> Even with tool access, once the model writes code, that code assumes a fixed graph. But many real‑world graphs change during the reasoning process, with nodes or edges updating on the fly. In such cases, static code becomes outdated immediately, so the model must maintain an internal, continuously updated representation of the graph to adjust its plan, revise tool calls, and keep reasoning consistent. A model that cannot internally track even a static graph will inevitably fail in these dynamic‑graph scenarios.
>
> **3. Benchmarking General Reasoning Under Algorithmic Constraints**
>
> Finally, the goal of our benchmark is not limited to solving graph problems per se, but to use them as structured, well‑defined testbeds to **evaluate the reasoning intelligence of LRMs**. Their real‑world relevance is only one aspect of our contribution; the broader purpose is to assess systematic general‑reasoning capabilities under algorithmic constraints.

---

> ### Author Response · Authors · 2025-11-27
> **Kind Reminder for Rebuttal**
>
> Dear Reviewer 8MVa,
>
> As the rebuttal period draws to a close, we are very interested in hearing any further feedback you might have and would be pleased to clarify or address any remaining concerns. Please do not hesitate to let us know if you have additional comments or questions.
>
> Thank you again for your time and thoughtful review.
>
> Best regards,
> Authors of Paper  8673

---

### Official Review · Reviewer_vGuw · 2025-11-03

**Soundness:** 3
**Presentation:** 3
**Contribution:** 2
**Rating:** 4
**Confidence:** 3

**Summary:**

The paper introduces **GRALGOBENCH**, a benchmark of nine graph-algorithm tasks spanning three “reasoning” categories Enumeration, Exploration, and Intuition and scales them across six graph sizes to stress LRMs’ long-context abilities. The headline claims are: (i) accuracy collapses as graphs grow (below ~50% beyond 120 nodes), and (ii) LRMs “over-think,” producing long, redundant traces; the authors analyze this by partitioning traces via high-entropy tokens like *wait*, *but*, *so*.

**Strengths:**

The nine tasks are precisely specified (with optimal algorithms/complexities in the appendix), and the dataset scales graph sizes systematically (Level-1…Level-6), which is valuable for probing context-length effects in a controlled way.

**Weaknesses:**

1. The core taxonomy (Enumeration / Exploration / Intuition) is *empirically* assigned by first generating 100 ER instances per problem, collecting LRM responses, and then asking **Qwen-2.5-72B** to classify which algorithmic family the *responses* reflect; the team then picks 9 tasks whose “algorithms are relatively unambiguous.” This risks circularity (taxonomy depends on current LRM behaviors) and injects judge-model bias into the ground truth of the benchmark’s conceptual framing. It also undermines generality, since task selection is contingent on that filter.

2. In the MST case study, nodes are *streets* and edges are *intersections* then the text states “the weight of each edge is the distance between two streets,” which is a confusing physical interpretation (intersections don’t have lengths, and “distance between streets” is ill-defined if nodes are streets). This muddles the mapping from the real network to its graph abstraction and invites ambiguity about what the algorithm is actually optimizing.

3. The paper enforces a `\boxed{}` final answer format, but some tasks (e.g., Distance Threshold) ask for a *node identifier* rather than a numeric value; exact-string matching on natural-language street names is brittle without strict canonicalization rules. The paper specifies decoding hyperparameters but does not detail robustness checks against case/spacing variants, which weakens the claim that graph tasks “enable standardized programmatic evaluation.” (Suggestion: emit canonical IDs alongside text and evaluate on IDs.)

**Questions:**

* Line 241: **“Minimum Spinning Tree” → “Minimum *Spanning* Tree”**
* **Acronym not introduced**: Figures use **MCP**; earlier text spells out *Maximum Clique Problem* without defining “(MCP)”. Define on first use.
* **Awkward phrasing**: “We enforce LRMs to output their final answers in the \boxed{} format.” → “We enforce that LRMs output…” (stylistic).

---

> ### Author Response · Authors · 2025-11-19
> **Rebuttal by authors**
>
> We sincerely thank the reviewer for the thoughtful questions and constructive comments. Our response addresses the following aspects: (1) Robustness of Algorithm Ratios and Validation of Dominant Reasoning Paradigms (2) Clarification and Prompt Revision for the MST Case Study (3) Canonical ID Emission and Evaluation (4) Editorial and Stylistic Modification.
>
> ### W1: Robustness of Algorithm Ratios and Validation of Dominant Reasoning Paradigms
>
> > W1: ...This risks circularity (taxonomy depends on current LRM behaviors) and injects judge-model bias into the ground truth of the benchmark’s conceptual framing. It also undermines generality, since task selection is contingent on that filter.
>
> We thank the reviewer for emphasizing this methodological aspect. We acknowledge that if each task were labeled using single‑judge classifications with only 100 samples, the resulting taxonomy might be influenced by model‑specific bias. To evaluate robustness, we performed additional experiments that **expand sample coverage**, **employ multiple judge and responder models**, and **include human alignment** analysis.
>
>
> #### 1. Expanded algorithm‑ratio evaluation on the full benchmark
>
> To mitigate sampling variance, we recomputed algorithm ratios on the **entire test split**:
>
> - **9 tasks**, each containing **300 problems**.
> - **4 reasoning traces per problem** from one responder LRM.
> - **Total per model: 9 × 300 × 4 = 10,800 traces.**
>
> This ensures that ratio estimates capture the complete benchmark distribution.
>
>
> #### 2. Multiple judge models and responder LRMs
>
> To reduce potential bias from a single judge, we used two extra LLM‑as‑judge models: **DeepSeek‑V3** and **GPT‑4.1.**
>
> To verify consistency across capability tiers, we evaluated three responder LRMs: **Qwen3‑8B**, **QWQ‑32B**, and **Gemini‑2.5‑pro.**
>
>
> #### 3. Human validation
>
> To align with human interpretation and assess the clarity of reasoning‑type distinctions, we recruited **three postgraduate judges** with competitive‑programming backgrounds. Each human judge independently annotated **100 randomly sampled traces per task**, using the same four‑way scheme: **Enumeration (en)** / **Exploration (ex)** / **Intuition (in)** / **Not mentioned (nm)**. All three judges labeled each trace independently, and any disagreements were resolved through discussion to determine the final label.
>
> The experiment results are shown as follows:

---

> > ### Author Response · Authors · 2025-11-19
> > **Rebuttal by Authors**
> >
> > #### **Qwen3‑8B**
> >
> > | Model / Judge | PathSum | Distancek | Diameter | Degree | Triangle | MCP | MKC | MST | DistanceThreshold |
> > |----------------|-----------|------------|-----------|-----------|-----------|-----------|-----------|-----------|----------------|
> > |  | en/ex/in/nm | en/ex/in/nm | en/ex/in/nm | en/ex/in/nm | en/ex/in/nm | en/ex/in/nm | en/ex/in/nm | en/ex/in/nm | en/ex/in/nm |
> > | **DeepSeek‑V3** | 0.0/100.0/0.0/0.0 | 1.3/89.6/4.6/4.5 | 0.0/98.8/0.0/1.2 | 90.8/0.0/0.0/9.2 | 83.8/8.8/1.7/5.7 | 90.8/0.0/0.0/9.2 | 0.0/0.0/98.3/1.7 | 0.0/0.0/100.0/0.0 | 14.6/0.0/85.0/0.4 |
> > | **GPT‑4.1** | 0.0/100.0/0.0/0.0 | 0.3/99.7/0.0/0.0 | 14.0/81.0/0.0/5.0 | 100.0/0.0/0.0/0.0 | 88.8/0.0/11.2/0.0 | 87.3/0.0/10.7/2.0 | 0.0/0.0/100.0/0.0 | 0.0/0.0/99.0/1.0 | 11.8/0.0/81.7/6.5 |
> > | **Human** | 0.0/100.0/0.0/0.0 | 7.0/93.0/0.0/0.0 |11.0/89.0/0.0/0.0 | 100.0/0.0/0.0/0.0 | 84.0/9.0/7.0/0.0 | 88.0/0.0/12.0/0.0 | 0.0/0.0/100.0/0.0 | 0.0/0.0/100.0/0.0 | 18.0/0.0/82.0/0.0 |
> >
> >
> >
> > #### **QWQ‑32B**
> >
> > | Model / Judge | PathSum | Distancek | Diameter | Degree | Triangle | MCP | MKC | MST | DistanceThreshold |
> > |----------------|-----------|------------|-----------|-----------|-----------|-----------|-----------|-----------|----------------|
> > | | en/ex/in/nm | en/ex/in/nm | en/ex/in/nm | en/ex/in/nm | en/ex/in/nm | en/ex/in/nm | en/ex/in/nm | en/ex/in/nm | en/ex/in/nm |
> > | **DeepSeek‑V3** | 0.0/100.0/0.0/0.0 | 0.0/96.3/2.5/1.2 | 0.0/99.6/0.0/0.4 | 100.0/0.0/0.0/0.0 | 73.8/14.7/0.8/10.7 | 86.7/0.0/0.8/12.5 | 0.0/0.0/99.2/0.8 | 0.0/0.0/100.0/0.0 | 16.2/5.8/83.8/0.0 |
> > | **GPT‑4.1** | 0.0/100.0/0.0/0.0 | 0.0/100.0/0.0/0.0 | 12.0/88.0/0.0/0.0 | 100.0/0.0/0.0/0.0 | 76.5/0.0/23.5/0.0 | 86.8/0.0/2.0/11.2 | 0.0/0.0/100.0/0.0 | 0.0/0.0/100.0/0.0 | 10.8/0.0/88.5/0.7 |
> > | **Human** | 0.0/100.0/0.0/0.0 | 0.0/100.0/0.0/0.0 | 7.0/93.0/0.0/0.0 | 100.0/0.0/0.0/0.0 | 77.0/10.0/13.0/0.0 | 87.0/0.0/13.0/0.0 | 0.0/0.0/100.0/0.0 | 0.0/0.0/100.0/0.0 | 12.0/0.0/88.0/0.0 |
> >
> >
> >
> > #### **Gemini‑2.5‑Pro**
> >
> > | Model / Judge | PathSum | Distancek | Diameter | Degree | Triangle | MCP | MKC | MST | DistanceThreshold |
> > |----------------|-----------|------------|-----------|-----------|-----------|-----------|-----------|-----------|----------------|
> > | | en/ex/in/nm | en/ex/in/nm | en/ex/in/nm | en/ex/in/nm | en/ex/in/nm | en/ex/in/nm | en/ex/in/nm | en/ex/in/nm | en/ex/in/nm |
> > | **DeepSeek‑V3** | 0.0/100.0/0.0/0.0 | 5.3/92.4/2.2/0.1 | 2.0/97.8/0.0/0.2 | 100.0/0.0/0.0/0.0 | 86.7/13.3/0.0/0.0 | 91.3/0.0/8.4/0.3 | 0.0/0.0/100.0/0.0 | 0.0/0.0/100.0/0.0 | 4.7/0.0/91.0/4.3 |
> > | **GPT‑4.1** | 0.0/100.0/0.0/0.0 | 0.0/100.0/0.0/0.0 | 0.0/100.0/0.0/0.0 | 100.0/0.0/0.0/0.0 | 82.3/8.7/9.0/0.0 | 90.4/0.0/5.0/4.6 | 0.0/0.0/100.0/0.0 | 0.0/0.0/100.0/0.0 | 2.0/0.0/93.7/4.3 |
> > | **Human** | 0.0/100.0/0.0/0.0 | 5.0/95.0/0.0/0.0 | 0.0/100.0/0.0/0.0 | 100.0/0.0/0.0/0.0 | 84.0/10.0/6.0/0.0 | 88.0/0.0/12.0/0.0 | 0.0/0.0/100.0/0.0 | 0.0/0.0/100.0/0.0 | 3.0/0.0/97.0/0.0 |
> >
> >
> > Across **judges models** and **humans**, the dominant reasoning paradigm per task is consistently preserved:
> >
> > - **PathSum / Distancek / Diameter → Exploration**
> > - **Degree / Triangle / MCP → Enumeration**
> > - **MKC / MST / DistanceThreshold → Intuition**
> >
> >
> > Overall, the algorithm ratios and task‑level taxonomy are **stable** across sampling, model type, and judgment source, confirming that the benchmark’s conceptual structure is **not an artifact of a specific judge model**, but a robust reflection of LRM reasoning behavior.

---

> ### Author Response · Authors · 2025-11-19
> **Rebuttal by Authors**
>
> ### W2: Clarification and Prompt Revision for the MST Case Study
> > W2: In the MST case study, nodes are streets and edges are intersections then the text states “the weight of each edge is the distance between two streets,” which is a confusing physical interpretation (intersections don’t have lengths)...
>
> We appreciate the reviewer’s insightful comment regarding the confusion in our MST case study. We have revised the prompt as follows:
>
> ```
> You are required to solve the Minimum Spanning Tree Problem over an undirected weighted network. Each node denotes a place (location). Each undirected edge connects two places, and its weight denotes the distance between the two places.
> - Places in the network: Alison Road, Abbotford Street, Anzac Parade, Doncaster Avenue, Brushwood Street, Barker Street, Blackwood Street, Borrodale Road, Bowral Street
> - Pairs of places and their distances: Alison Road and Anzac Parade (distance: 5), Alison Road and Doncaster Avenue (distance: 7), Abbotford Street and Anzac Parade (distance: 6), Abbotford Street and Doncaster Avenue (distance: 8), Anzac Parade and Barker Street (distance: 2), Anzac Parade and Borrodale Road (distance: 4), Anzac Parade and Bowral Street (distance: 1), Anzac Parade and Doncaster Avenue (distance: 3), Doncaster Avenue and Barker Street (distance: 4), Doncaster Avenue and Blackwood Street (distance: 9), Doncaster Avenue and Borrodale Road (distance: 10), Doncaster Avenue and Bowral Street (distance: 7), Doncaster Avenue and Brushwood Street (distance: 9).
> Identify the minimum spanning tree of this network (minimizing the total distance) without any cycles.
> Present your answer in the following format: '\boxed{n}', where n is the sum of the distances of the edges in the minimum spanning tree.
> ```
>
> We evaluated **Qwen3‑32B**, **GPT‑5-mini**, and **DeepSeek‑R1** on both the original and the revised prompts. For each prompt version, we generated 30 problems across level‑1, level‑2, and level‑3 categories. Each problem produced four answer samples per model, from which we computed **pass@4** and **cons@4** metrics. The detailed results are summarized as follows.
>
>
>
>
>
> #### **Qwen3‑32B (MST)**
>
> | Difficulty | old Cons@K | new Cons@K | old Pass@K | new Pass@K |
> |:-----------:|:-----------:|:-----------:|:-----------:|:-----------:|
> | Level-1   | 1.00 | 1.00 | 1.00 | 1.00 |
> | Level-2  | 0.90 | 0.73 | 1.00 | 0.97 |
> | Level-3    | 0.13 | 0.10 | 0.47 | 0.43 |
>
>
>
> #### **DeepSeek‑R1 (MST)**
>
> | Difficulty | old Cons@K | new Cons@K | old Pass@K | new Pass@K |
> |:-----------:|:-----------:|:-----------:|:-----------:|:-----------:|
> | Level-1        | 1.00 | 1.00 | 1.00 | 1.00 |
> | Level-2     | 0.93 | 0.93 | 1.00 | 1.00 |
> | Level-3      | 0.80 | 0.90 | 0.87 | 0.97 |
>
>
>
>
> #### **GPT‑5-mini (MST)**
>
> | Difficulty | old Cons@K | new Cons@K | old Pass@K | new Pass@K |
> |:-----------:|:-----------:|:-----------:|:-----------:|:-----------:|
> | Level-1       | 1.00 | 1.00 | 1.00 | 1.00 |
> | Level-2    | 0.73 | 0.77 | 0.83 | 0.90 |
> | Level-3      | 0.63 | 0.57 | 0.70 | 0.70 |
>
>
>
>
> The results indicate that the performance difference before and after the prompt modification is minimal and does not affect the main conclusions of our paper. This may be because the models are highly robust to variations in textual descriptions. We will unify the prompts across all related data when releasing the dataset to further enhance its reliability.
>
> Again, we sincerely thank the reviewer for the helpful suggestion.

---

> ### Author Response · Authors · 2025-11-19
> **Rebuttal by Authors**
>
> ### W3: Canonical ID Emission and Evaluation
> > W3: ...The paper specifies decoding hyperparameters but does not detail robustness checks against case/spacing variants, which weakens the claim that graph tasks “enable standardized programmatic evaluation...
>
> You raised an excellent point concerning the rigor of our evaluation protocol. Following your suggestion, we modified the prompt to **emit canonical node IDs alongside text outputs** and to **evaluate models based on these IDs**. The revised prompt is as follows:
>
> ```
> You are given an undirected weighted graph representing the airport network, where nodes represent airports (codes: DHM, BDQ, HEA, BKK, BBI, KBL, AMD, BLR, DEL) and edges represent direct flights between airports. The weight of each edge is the distance between two airports.
> - The list of airports: DHM, BDQ, HEA, BKK, BBI, KBL, AMD, BLR, DEL
> - The direct flights (edges) are: [DHM, DEL, 408], [BDQ, DEL, 796], [HEA, DEL, 1544], [HEA, KBL, 642], [BKK, DEL, 2947], [BKK, BLR, 2498], [BBI, BLR, 1188], [BBI, DEL, 1277], [KBL, DEL, 1001], [AMD, BLR, 1244], [AMD, DEL, 758], [BLR, DEL, 1737]
> - The distance threshold is 1449.
> - Canonical node IDs: DHM:0, BDQ:1, HEA:2, BKK:3, BBI:4, KBL:5, AMD:6, BLR:7, DEL:8
> For each airport, the distance to another airport is defined as the sum of the weights (distances) along the shortest path connecting them.
> Your task is: Return the airport code with the smallest number of other airports that can be reached with a shortest path distance no more than the threshold. If there are multiple such airports, return the one with the lexicographically largest code.
> Present your answer in the following format: '\boxed{airport_code|id}'. We will grade on the id.
> ```
>
> We evaluated **Qwen3‑32B**, **GPT‑5-mini**, and **DeepSeek‑R1** on both the original and the revised prompts. For each prompt version, we generated 30 problems across level‑1, level‑2, and level‑3 categories. Each problem produced four answer samples per model, from which we computed **pass@4** and **cons@4** metrics. The detailed results are summarized as follows.
>
>
>
> #### **Qwen3‑32B (DistanceThreshold)**
>
> | Difficulty | old Cons@K | new Cons@K | old Pass@K | new Pass@K |
> |:-----------:|:-----------:|:-----------:|:-----------:|:-----------:|
> | Level-1    | 1.00 | 1.00 | 1.00 | 1.00 |
> | Level-2         | 0.83 | 0.57 | 0.90 | 0.70 |
> | Level-3       | 0.57 | 0.47 | 0.63 | 0.87 |
>
>
> #### **DeepSeek‑R1 (DistanceThreshold)**
>
> | Difficulty | old Cons@K | new Cons@K | old Pass@K | new Pass@K |
> |:-----------:|:-----------:|:-----------:|:-----------:|:-----------:|
> | Level-1      | 1.00 | 1.00 | 1.00 | 1.00 |
> | Level-2     | 0.97 | 1.00 | 1.00 | 1.00 |
> | Level-3       | 0.90 | 1.00 | 1.00 | 1.00 |
>
>
>
> #### **GPT‑5-mini (DistanceThreshold)**
>
> | Difficulty | old Cons@K | new Cons@K | old Pass@K | new Pass@K |
> |:-----------:|:-----------:|:-----------:|:-----------:|:-----------:|
> | Level-1       | 1.00 | 1.00 | 1.00 | 1.00 |
> | Level-2    | 0.87 | 0.73 | 0.93 | 0.83 |
> | Level-3       | 0.60 | 0.57 | 0.83 | 0.87 |
>
>
>
>
>
>
> The experimental results show that the prompt modification has **minimal impact on model performance**, and our main conclusions remain unchanged. This also suggests that even before the change, model outputs for the `airport_code` were already consistent with the codes described in the prompt, indicating robust alignment between model predictions and graph canonicalization.
>
>
>
> ### Q1, Q2, and Q3: Editorial and Stylistic Modification
>
> >Line 241: “Minimum Spinning Tree” → “Minimum Spanning Tree”
>
> >Acronym not introduced: Figures use MCP; earlier text spells out Maximum Clique Problem without defining “(MCP)”. Define on first use.
>
> >Awkward phrasing: “We enforce LRMs to output their final answers in the \boxed{} format.” → “We enforce that LRMs output…” (stylistic)..
>
>
> We sincerely thank the reviewer for the helpful writing and style suggestions. In the revised version, we have corrected *“Minimum Spinning Tree”* to *“Minimum Spanning Tree”*, introduced the acronym *(MCP)* upon first mention of *Maximum Clique Problem*, and revised the sentence to *“We enforce that LRMs output their final answers in the \boxed{} format.”*
>
> All corresponding edits have been incorporated into the revised manuscript.

---

> ### Author Response · Authors · 2025-11-27
> **Kind Reminder for Rebuttal**
>
> Dear Reviewer vGuw,
>
> As the rebuttal period draws to a close, we are very interested in hearing any further feedback you might have and would be pleased to clarify or address any remaining concerns. Please do not hesitate to let us know if you have additional comments or questions.
>
> Thank you again for your time and thoughtful review.
>
> Best regards,
> Authors of Paper  8673

---

### Author Response · Authors · 2025-12-01
**Revision Summary**

Dear Reviewers,

Thank you for your valuable feedback! We have revised the manuscript as follows (all related changes are highlighted in BLUE in the PDF):

- In the Introduction, we replaced “long-context evaluation” with “long-context input evaluation” to more clearly emphasize that one of the key functions of GRALGOBENCH is to test LRMs’ ability to process and reason over long textual inputs rather than long textual outputs.
- In all components that rely on an LLM-as-judge (e.g., reasoning taxonomy labeling, error categorization), we explicitly added human verification on sampled instances to increase the reliability and credibility of the annotations.
- We added a Discussion section, where we: (1) analyze the potential benefits of letting LRMs generate code and invoke external tools for solving graph algorithm problems, and (2) examine how far our insights generalize beyond the specific tasks and settings in this work, supported by additional experiments.
- In Appendix J, we added an experiment that groups the test set by problem complexity (Easy / Medium / Hard) and reports the corresponding additional findings.
- Following the reviewers’ suggestions, we also fixed several grammar issues and figure formatting problems to improve clarity and presentation.

---

### Meta-Review · Area_Chair_6A7n · 2026-01-07

**Summary:**

All reviewers acknowledged strengths of the paper, with praise for the careful design of the experiments, the thorough error analysis and the thoughtful and well-motivated construction of the reasoning taxonomy.
Several concerns were raised during the review process.

C1. In the originally submitted paper, the taxonomy is determined empirically by asking an LRM to determine which algorithmic family the responses reflect. This poses the danger of circularity.

C2. Some experimental decisions concerning formatting of prompts are questionable.

C3. Some of the problems would likely be much better handled by allowing the LRM to use tool calls. While there is value in isolating intrinsic reasoning, disallowing them raises questions about whether the assessment pertains to a more realistic use of LRMs (and whether the benchmark is useful for the setting where LRMs can use tools).

C4. The distinction between the presented benchmark and GraphWalks is unclear.

C5. There is no evidence that any conclusions about long-context brittleness and over-thinking generalize beyond graph-structured problems. It is questionable whether some of the graph problems align with general reasoning tasks.

C6. The difficulty of a question would be better measured by the computational complexity of the algorithm required to solve the underlying graph problem rather than graph size.

**Reviewer Concerns:**

The response to the reviews was very thorough with many additional experiments for clarification and validation. The authors also exhibited an openness to constructive criticism by making changes to the methodology and paper where appropriate.

C1. The authors addressed this by a considerably more thorough procedure for taxonomy construction that involved human alignment checks and consistency across multiple judges.

C2. The identified prompts were modified to remove confusion in graph construction and enhance the robustness of the testing process.

C3. The response provided experimental results demonstrating that the use of tools does lead to dramatically improved performance. The authors argue that it is important to evaluate intrinsic reasoning capability without the use of tools. An alternative viewpoint is that the correct reasoning approach to solve the introduced problems is to write an algorithm (rather than trying to identify a minimum spanning tree in my head, I would write and apply an MST algorithm). From this perspective, preventing the use of tools can be seen as unreasonably and unrealistically handicapping an LRM, making any conclusions derived from the benchmark less relevant.

C4. The authors provide a thorough comparison that clear identifies the need for and advantages of the introduced benchmark.

C5. The response provides a study that analyzes the correlation between performance on the introduced benchmark and other non-graph reasoning benchmarks. This provides convincing evidence that the graph-based benchmark does provide a good indication of reasoning capability for non-graph problems.

C6. The response provides a thorough study of how performance varies when using this alternative measure of problem difficulty.

**Reviewer Scores:**

Reviewer vGuw. LIKELY TO CHANGE (4 to 6). The reviewer raised three concrete criticisms (summarized in C1 and C2). The response directly addressed these issues, with a more carefully checked taxonomy construction, and strategies to address the identified prompt issues. It is likely that these modifications would have satisfied the reviewer.

Reviewer 8MVa. UNLIKELY TO CHANGE. The original score was already an 8.

Reviewer dfQZ. UNLIKELY TO CHANGE. The reviewer provided a score of 6 but identified multiple weaknesses and posed several questions. Although the response was very detailed, the concerns regarding tool-use and external validity and scope are likely to result in a retention of the positive score.

Reviewer c193. POSSIBLE/PROBABLE INCREASE (4 to 6). This was a very brief review with only two weaknesses and one question. The response clearly addresses the first weakness. The response also provides an argument addressing the second criticism. For this second point, which questions whether the graph algorithm problems such as MST “resemble questions in real life”, my interpretation is that the reviewer is questioning whether these problems represent real-world reasoning tasks. The response instead focuses on the real-world applicability of calculating a minimum spanning tree, which seems to miss the point of whether this adequately mirrors a reasoning exercise.

---

### Decision · Program_Chairs · 2026-01-26

Accept (Poster)